# MULTITASK PROMPTED TRAINING ENABLES ZERO-SHOT TASK GENERALIZATION

**Victor Sanh**[*]
Hugging Face

**Albert Webson**[*]
Brown University

**Colin Raffel**[*]
Hugging Face

**Stephen H. Bach**[*]
Brown & Snorkel AI

**Lintang Sutawika**
BigScience

**Zaid Alyafeai**
KFUPM

**Antoine Chaffin**
IRISA & IMATAG

**Arnaud Stiegler**
Hyperscience

**Teven Le Scao**
Hugging Face

**Arun Raja**
$I^2R$, Singapore

**Manan Dey**
SAP

**M Saiful Bari**
NTU, Singapore

**Canwen Xu**
UCSD & Hugging Face

**Urmish Thakker**
SambaNova Systems

**Shanya Sharma**
Walmart Labs

**Eliza Szczechla**
BigScience

**Taewoon Kim**
VU Amsterdam

**Gunjan Chhablani**
BigScience

**Nihal V. Nayak**
Brown University

**Debajyoti Datta**
University of Virginia

**Jonathan Chang**
ASUS

**Mike Tian-Jian Jiang**
ZEALS, Japan

**Han Wang**
NYU

**Matteo Manica**
IBM Research

**Sheng Shen**
UC Berkeley

**Zheng-Xin Yong**
Brown University

**Harshit Pandey**
BigScience

**Michael McKenna**
Parity

**Rachel Bawden**
Inria, France

**Thomas Wang**
Inria, France

**Trishala Neeraj**
BigScience

**Jos Rozen**
Naver Labs Europe

**Abheesht Sharma**
BITS Pilani, India

**Andrea Santilli**
University of Rome

**Thibault Fevry**
BigScience

**Jason Alan Fries**
Stanford & Snorkel AI

**Ryan Teehan**
Charles River Analytics

**Tali Bers**
Brown University

**Stella Biderman**
Booz Allen & EleutherAI

**Leo Gao**
EleutherAI

**Thomas Wolf**
Hugging Face

**Alexander M. Rush**
Hugging Face

## ABSTRACT

Large language models have recently been shown to attain reasonable zero-shot generalization on a diverse set of tasks (Brown et al., 2020). It has been hypothesized that this is a consequence of implicit multitask learning in language models' pretraining (Radford et al., 2019). Can zero-shot generalization instead be directly induced by *explicit* multitask learning? To test this question at scale, we develop a system for easily mapping any natural language tasks into a human-readable prompted form. We convert a large set of supervised datasets, each with multiple prompts with diverse wording. These prompted datasets allow for benchmarking the ability of a model to perform completely unseen tasks. We fine-tune a pretrained encoder-decoder model (Raffel et al., 2020; Lester et al., 2021) on this multitask mixture covering a wide variety of tasks. The model attains strong zero-shot performance on several standard datasets, often outperforming models up to $16\times$ its size. Further, our approach attains strong performance on a subset of tasks from the BIG-bench benchmark, outperforming models up to $6\times$ its size. All trained models are available at https://github.com/bigscience-workshop/t-zero, and all prompts are available at https://github.com/bigscience-workshop/promptsource.

## 1 INTRODUCTION

Recent work has shown that large language models exhibit the ability to perform reasonable zero-shot generalization to new tasks (Brown et al., 2020; Kim et al., 2021). Despite being trained on only language modeling objectives, these models can perform relatively well at new tasks that they have not been explicitly trained to perform, for instance answering a question on a passage or performing summarization. An influential hypothesis is that large language models generalize to new tasks as a

---

[*]Equal contribution. Full list of individual contributions detailed in Appendix A.

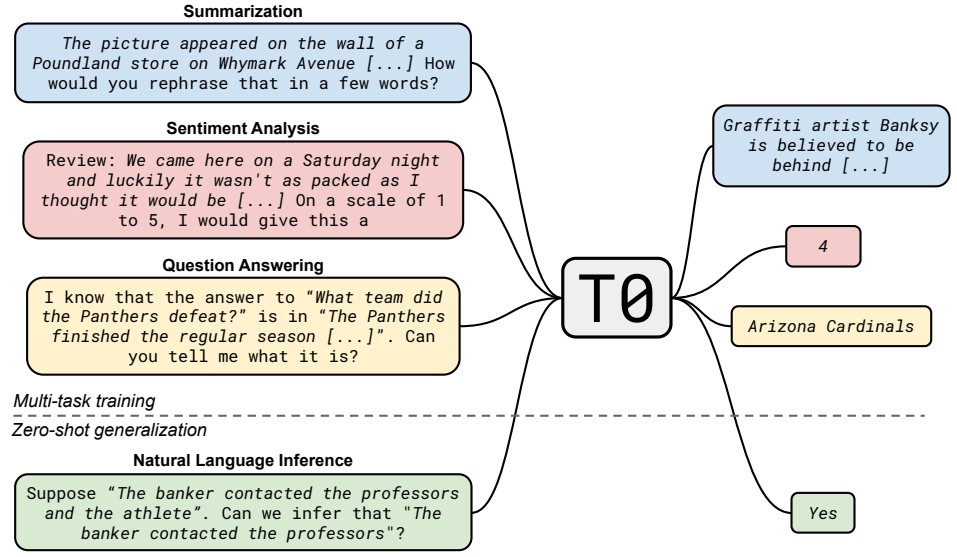

Figure 1: Our model and prompt format. T0 is an encoder-decoder model that consumes textual inputs and produces target responses. It is trained on a multitask mixture of NLP datasets partitioned into different tasks. Each dataset is associated with multiple prompt templates that are used to format example instances to input and target pairs. Italics indicate the inserted fields from the raw example data. After training on a diverse mixture of tasks (top), our model is evaluated on zero-shot generalization to tasks that are not seen during training (bottom).

result of an implicit process of multitask learning (Radford et al., 2019). As a byproduct of learning to predict the next word, a language model is forced to learn from a mixture of implicit tasks included in their pretraining corpus. For example, by training on generic text from a web forum, a model might implicitly learn the format and structure of question answering. This gives large language models the ability to generalize to unseen *tasks* presented with natural language prompts, going beyond prior multitask studies on generalization to unseen *datasets* (Khashabi et al., 2020a; Ye et al., 2021). However, this ability requires a sufficiently large model and is sensitive to the wording of its prompts (Perez et al., 2021; Zhao et al., 2021; Reynolds and McDonell, 2021).

Further, it is an open question how implicit this multitask learning really is. Given the scale of recent language models' pretraining corpora, it is reasonable to expect that some common natural language processing (NLP) tasks would appear in an explicit form in their pretraining corpora, thereby directly training the models on those tasks. For example, there are many websites that simply contain lists of trivia questions and answers,[1] which are precisely supervised training data for the task of closed-book question answering (Roberts et al., 2020). We hypothesize that such multitask supervision in pretraining plays a large role in zero-shot generalization.

In this paper, we focus on explicitly training language models in a supervised and massively multi-task fashion. Our approach uses a training mixture consisting of a large set of different tasks specified in natural language prompts. Our goal is to induce a model to better generalize to unseen tasks without requiring massive scale, as well as being more robust to the wording choices of the prompts. To convert a large set of natural language tasks into prompted form, we use a simple templating language for structured datasets. We develop an interface for prompt collection from public contributors that facilitated the collection of a large multitask mixture with multiple prompts per dataset (Bach et al., 2022). We then train a variant of the T5 encoder-decoder model (Raffel et al., 2020; Lester et al., 2021) on a subset of the tasks (each with multiple datasets) and then evaluate tasks and prompts that the model was *not* trained on.

Our experiments study two questions. First, does multitask prompted training improve generalization to unseen tasks? Second, does training on a wider range of prompts improve robustness to prompt wording? For the first question, we find that multitask training enables zero-shot task generalization by showing that our model matches or exceeds the performance of GPT-3 (Brown et al., 2020) on 9 out of 11 held-out datasets, despite being about 16× smaller. We also show that the model improves over a large baseline language model on 13 out of 14 tasks in the BIG-bench benchmark

---

[1]For example, https://www.quizbreaker.com/trivia-questions, https://www.scarymommy.com/best-trivia-questions-answers/, and https://parade.com/944584/parade/trivia-questions-for-kids/.

(BIG-bench collaboration, 2021). For the second question, we find that training on more prompts per dataset consistently improves the median and decreases the variability of performance on held-out tasks. Training on prompts from a wider range of datasets also generally improves the median but does not consistently decrease the variability.

## 2 RELATED WORK

In this work, we distinguish implicit multitask learning in language model pretraining from *explicit* multitask learning (Caruana, 1997), the technique for mixing multiple tasks into a single supervised training process. Models trained with multitask learning have long been shown to have improved performance in NLP (Collobert and Weston, 2008). Since different tasks have different outputs, applying multitask learning requires a shared format, and various have been used (Hashimoto et al., 2016; McCann et al., 2018). Several multitask works also explore few-shot and zero-shot generalization to new datasets with large pretrained models (e.g., Vu et al., 2020; Ye et al., 2021).

Natural language prompting is the method of reformatting NLP tasks in the format of a natural language response to natural language input. The development of text-to-text pretrained models such as T5 (Raffel et al., 2020) makes prompts a particularly useful method for multitask learning. For example, Khashabi et al. (2020a) reformat 20 question-answering datasets into a single prompt of `question: ... (A)... (B)... (C)... context: ...`, while later work such as Zhong et al. (2021) and Wang et al. (2021) cast a range of datasets into a single boolean QA prompt or a single NLI prompt, respectively. Although effective, these single-prompt methods typically do not generalize to new prompts or new tasks inexpressible in their fixed format.

More generally, Schick and Schütze (2021) and Brown et al. (2020) popularized using prompts as a generic method for all NLP tasks. Mishra et al. (2021) further extend this approach to a multitask setup, training on prompts for 61 narrowly defined tasks (e.g., question generation, incorrect answer generation) adapted from 9 datasets' crowdsourcing instructions, whereas we train on and measure generalization across 62 datasets and 12 tasks as traditionally defined in the NLP literature (§3). Additionally, their prompts include labeled examples in addition to instructions, whereas we focus on zero-shot generalization. Lastly, concurrent work by Wei et al. (2021) shares a similar research question with us, although we differ in several substantive regards, e.g., prompt diversity, model scale, and held-out-task scheme. We discuss our differences in detail in Section 7.

Finally, in explaining the success of prompts, the leading hypothesis is that models learn to understand the prompts as task instructions which help them generalize to unseen tasks (Wei et al., 2021; Mishra et al., 2021; Schick and Schütze, 2021; Brown et al., 2020). However, the extent to which this success depends on the semantic meaningfulness of the prompts has been challenged (Webson and Pavlick, 2021; Logan et al., 2021). Thus, in this work, we remain agnostic as to why prompts support generalization. We only claim that prompts serve as a natural format for multitask training which empirically supports generalization to unseen tasks.

## 3 MEASURING GENERALIZATION TO UNSEEN TASKS

We begin by assuming an underlying partition of NLP datasets into tasks. We use the term "task" to refer to a general NLP ability that is tested by a group of specific datasets. To evaluate zero-shot generalization to new tasks, we train on a subset of tasks and evaluate on a held-out group of tasks.

Unfortunately, NLP task categorization is fuzzy, particularly if one tries to isolate a unique skill. For example, many datasets evaluate commonsense knowledge, and some multitask works (e.g., Brown et al., 2020; Wei et al., 2021) define commonsense as a standalone task. However, commonsense datasets differ vastly, ranging from innate knowledge and grade-school science to DIY instructions, US cultural norms, and graduate-level theorems (see Appendix E.1 for a detailed discussion).

Noting that grouping by task is an imperfect heuristic, we err on the side of organizing our task taxonomy according to the task format as opposed to required skill based on conventions in the literature (Khashabi et al., 2020b; Vu et al., 2020; Ye et al., 2021). We collect all datasets from these papers and exclude those that are not in English (which also excludes programming languages and structured annotations such as parse trees) or if they require special domain knowledge (e.g., biomedicine). This yields 12 tasks and 62 datasets with publicly contributed prompts in our training

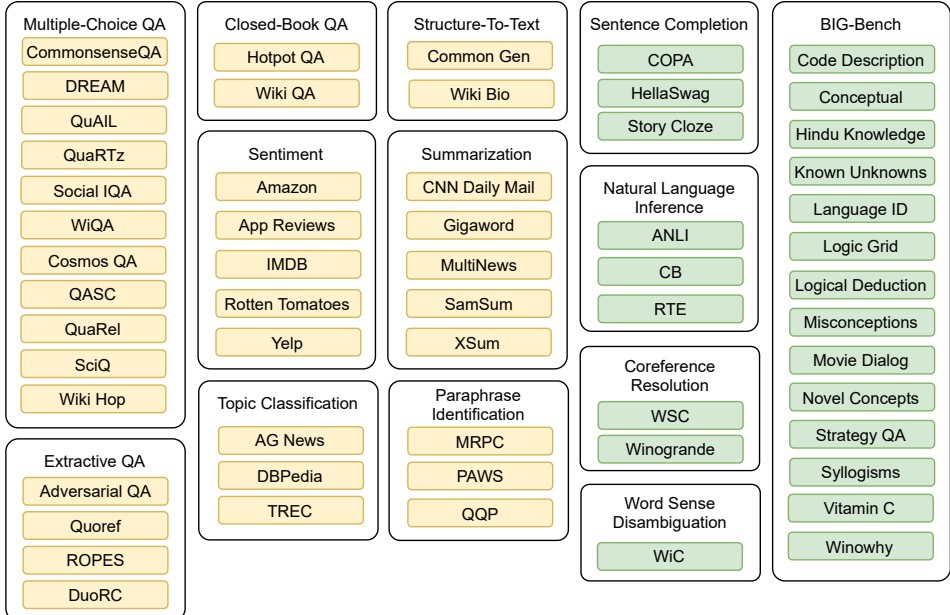

Figure 2: T0 datasets and task taxonomy. (T0+ and T0++ are trained on additional datasets. See Table 5 for the full list.) Color represents the level of supervision. Yellow datasets are in the training mixture. Green datasets are held out and represent tasks that were not seen during training. Hotpot QA is recast as closed-book QA due to long input length.

and evaluation mixtures (Figure 2) as of writing. All experiments use datasets in the Hugging Face datasets library (Lhoest et al., 2021).

To test zero-shot generalization, we hold out all constituent datasets of four tasks: natural language inference (NLI), coreference resolution, sentence completion, and word sense disambiguation. We choose NLI as a held-out task because humans also zero-shot generalize to NLI as an unseen task: Most humans are never explicitly trained to classify whether a premise sentence entails or contradicts a hypothesis sentence, yet they find it intuitive to perform this task without training (Williams et al., 2020). For the same reason, we also hold out coreference resolution and word sense disambiguation. We further hold out sentence completion because it is a task possibly too similar to NLI (Appendix E.2 discusses this in detail). Additionally, we do not train our main model on any datasets that Brown et al. (2020) used for evaluation, so that our main results will be a fair zero-shot comparison. We also verify that data for those tasks is not leaked through the pretraining corpus (Appendix F).

Lastly, we further evaluate on a subset of the datasets from BIG-bench, which is a recent community-driven benchmark to create a diverse collection of difficult tasks to test the abilities of large language models. The subset of BIG-bench comprise a language-oriented selection of tasks for which the BIG-bench maintainers have prepared preliminary results and which constitute text that is in-vocabulary for the T5 tokenizer (i.e. only contain English-language text without emojis or other special characters). All tasks from BIG-bench are novel tasks that were unseen in our training.

## 4 A UNIFIED PROMPT FORMAT

All datasets are given to our model in natural language prompted form to enable zero-shot experimentation. To facilitate writing a large collection of prompts, we develop a templating language and an application that make it easy to convert diverse datasets into prompts. We define a *prompt* as consisting of an input template and a target template, along with a collection of associated metadata. The templates are functions mapping a data example into natural language for the input and target sequences. Practically, the templates allow the user to mix arbitrary text with the data fields, metadata, and other code for rendering and formatting raw fields. For example, in the case of an NLI dataset, the example would include fields for `Premise`, `Hypothesis`, `Label`. An input template would be `If {Premise} is true, is it also true that {Hypothesis}?`, whereas a target template can be defined with the label choices `{Choices[label]}`. Here `Choices` is prompt-specific metadata that consists of the options `yes`, `maybe`, `no` corresponding to `label` being entailment (0), neutral (1) or contradiction (2). Other metadata documents

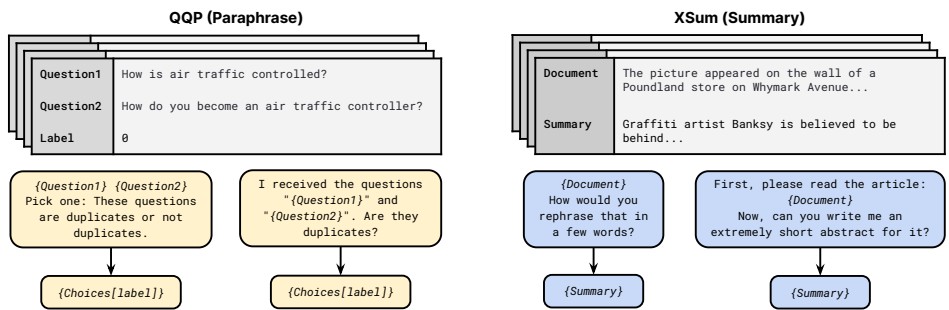

Figure 3: Prompt templates from the P3 prompt collection. Each dataset has multiple prompt templates consisting of an input and a target template. These use the fields of the raw data examples as well as template metadata, e.g., the left paraphrasing identification prompts use *Choices*, a template-level list variable ['Not duplicates', 'Duplicates']. These templates are materialized to produce the prompted instance shown in Figure 1. The complete set of prompt templates used in T0 is given in Appendix H.

additional properties, such as an evaluation metric. Each data example is materialized with many different prompt templates as shown in Figure 3.

To develop prompts, we built an interface for interactively writing prompts on datasets. We put out an open call in the research community for users to contribute prompts. 36 contributors affiliated with 24 institutions in 8 countries participated. Since our goal was to train a model to be robust to prompt format, and since the question of what makes a prompt effective remains unresolved (Webson and Pavlick, 2021; Logan et al., 2021; Zhao et al., 2021), we encouraged contributors to be open in their style and create a diverse set of prompts. The main annotation guideline was that prompts needed to be grammatical and understandable by a fluent English speaker with no prior experience of the tasks. Additionally, prompts that required explicit counting or numerical indexing were removed in favor of natural language variants. For example, instead of predicting indices of a span extracting answers from a passage, the model is expected to copy the span's text instead. With these minimal constraints, prompt writers were encouraged to use both formal and creative prompts and various orderings of the data.

Most of the prompts correspond directly to a version of the original proposed task, although we also allow prompts that permuted the original task (for instance, generating a document from its summary). Such non-original-task prompts are included in our training mixtures for improved diversity, but they are not reported in evaluation since they deviate from the metrics and baselines reported by the original datasets.

The details of the prompting language and tool are given in Appendix D and Bach et al. (2022), and the prompts themselves are given in Appendix H. We collected prompts for English datasets, excluding ones that included potentially harmful content or non-natural language such as programming languages. We refer to this collection as the *Public Pool of Prompts* (P3). As of writing, P3 contains 2073 prompts for 177 datasets (11.7 prompts per dataset on average). Prompts used in experiments are all sourced from P3 except for BIG-bench, the prompts of which are provided by its maintainers.

## 5 EXPERIMENTAL SETUP

**Model**   At a high level, we fine-tune a pretrained model on our multi-task training mixture of natural language prompted datasets. Our model uses an encoder-decoder architecture with input text fed to the encoder and target text produced by the decoder. The model is trained to autoregressively generate the target through standard maximum likelihood training. Unlike decoder-only language models such as GPT-3, it is never trained to generate the input.

All models we trained are based on T5, a Transformer-based encoder-decoder language model pretrained with a masked language modeling-style objective on 1T tokens from C4 (Raffel et al., 2020). Since T5's pretraining objective is generating tokens and only tokens that have been removed from the input text, it is different from the natural text generation format of prompted datasets. Therefore, we use Lester et al. (2021)'s *LM-adapted T5* model (referred to as T5+LM), produced by training T5 on 100B additional tokens from C4 on a standard language modeling objective.

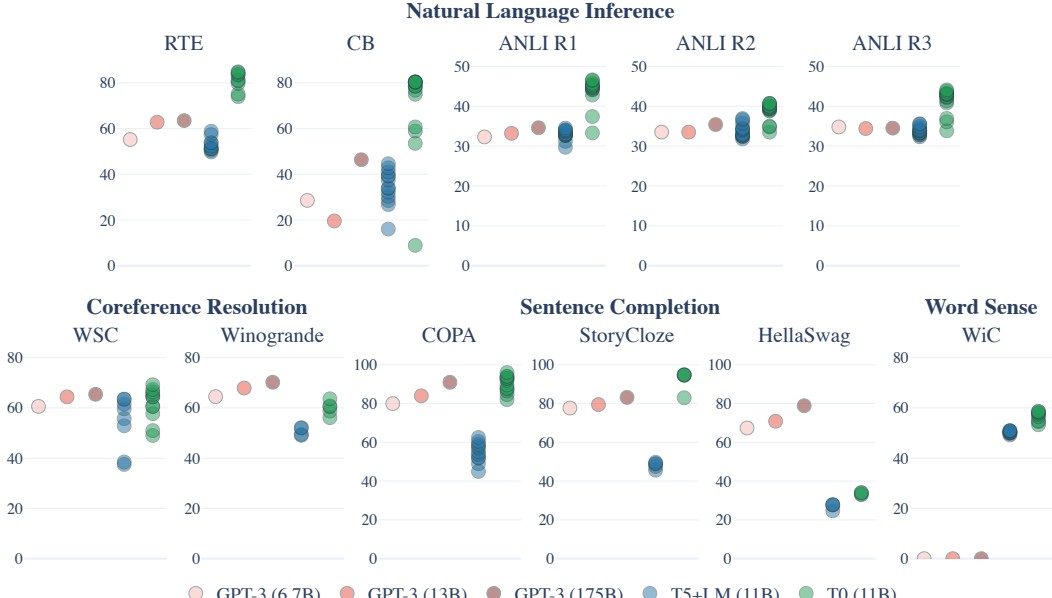

Figure 4: Results for T0 task generalization experiments compared to GPT-3 (Brown et al., 2020). Each dot is the performance of one evaluation prompt. The baseline T5+LM model is the same as T0 except without multitask prompted training. GPT-3 only reports a single prompt for each dataset.

**Training** Our main model, *T0*, is trained on the multitask mixture detailed in Section 3 and Table 5. Meanwhile, *T0+* is the same model with identical hyperparameters except trained on a mixture that adds GPT-3's evaluation datasets. Lastly, *T0++* further adds SuperGLUE (Wang et al., 2019a) to the training mixture (except RTE and CB), which leaves NLI and the BIG-bench tasks as the only held-out tasks.

The above T0 variants are all initialized from the 11B parameters version of T5+LM. To study the effect of scaling and to aid researchers with less resources, we also train *T0 (3B)*, which has the same training mixture as T0 but is initialized from the 3B parameters version of T5+LM (results reported in Appendix G).

We perform checkpoint selection by choosing the checkpoint that yields the highest score on the validation splits of our training datasets. This still satisfies the *true zero-shot* (Perez et al., 2021) setting as we do not use any examples from any of the held-out tasks to select the best checkpoint.

We assemble our multitask training mixture by combining and shuffling all examples from all training datasets. This is equivalent to sampling from each dataset in proportion to the number of examples in the dataset. However, the number of examples in each of our training datasets varies by two orders of magnitude. We therefore follow the strategy used in Raffel et al. (2020) and treat any dataset with over 500'000 examples as having 500'000 / `num_templates` examples for the purposes of sampling, where `num_templates` is the number of templates created for the dataset.

We truncate input and target sequences to 1024 and 256 tokens, respectively. Following Raffel et al. (2020), we use packing to combine multiple training examples into a single sequence to reach the maximum sequence length. We use a batch size of 1024 sequences (corresponding to $2^{20}$ total input tokens per batch) and the Adafactor optimizer (Shazeer and Stern, 2018). Following standard practice for fine-tuning T5, we use a learning rate of 1e-3 and a dropout rate of 0.1.

**Evaluation** We evaluate zero-shot generalization on 11 datasets in 4 held-out traditional NLP tasks: natural language inference, coreference, word sense disambiguation, and sentence completion, as well as 14 novel tasks from BIG-bench (§3). Unless specified otherwise, we report performance on the validation splits. All reported datasets use accuracy as their metric.

For tasks that involve choosing the correct completion from several options (e.g. multiple choice question answering), we follow Brown et al. (2020) and use *rank classification* to evaluate our model: we compute the log-likelihood of each of the target options under the fine-tuned model and

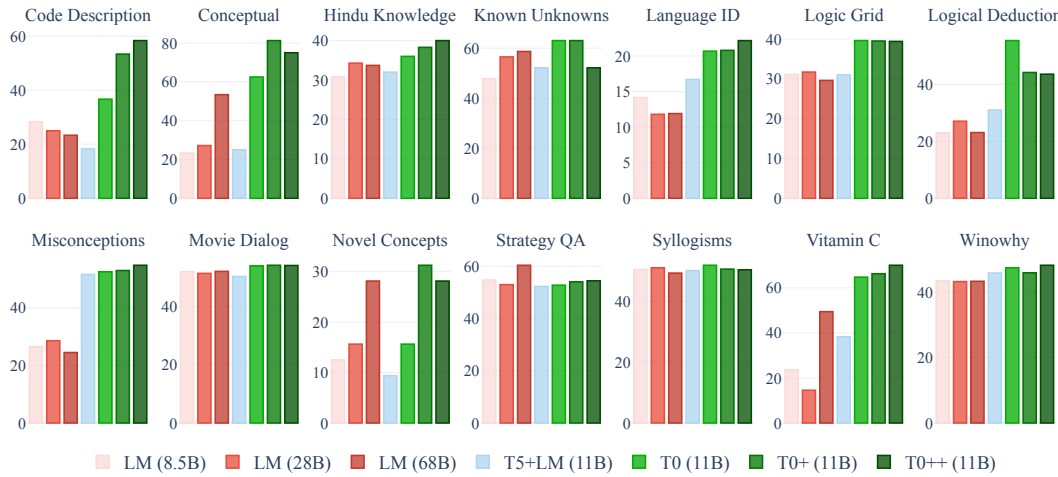

Figure 5: Results for a subset of BIG-bench which has available baselines. The baseline models are Transformer-based language models provided by BIG-bench maintainers, who also provide one prompt per dataset. T0, T0+ and T0++ are identical except for increasing the number of training datasets (§5). BIG-bench Tasks are all zero-shot for all the reported models.

select the option with the highest log-likelihood as the prediction. For simplicity, we do not apply length normalization to the log-likelihoods of the target options.

We do not perform prompt selection by comparing the performance of different prompts on the validation split; Perez et al. (2021) highlights how such a strategy leaks information from the evaluation splits, which makes the evaluation not "true" zero-shot. For a given dataset, we report the median performance across all prompts for this dataset along with their interquartile range (Q3 - Q1) to measure the model's robustness to the wording of the prompts.

## 6 RESULTS

### 6.1 GENERALIZATION TO UNSEEN TASKS

Our first research question is whether multitask prompted training improves generalization to unseen tasks. In Figure 4, we compare T0 against our T5+LM baseline on four held-out tasks. Our approach leads to significant gains over our baseline on all datasets, demonstrating the benefits of multitask prompted training over only language modeling training with an identical model and prompts.

Next, we compare T0 to the zero-shot performance of the largest language models available as of writing, i.e., various GPT-3 models up to 175B parameters. Note that Brown et al. (2020) report performance on a single prompt,[2] whereas we report the median and interquartile range of performance across all prompts in P3 without cherry picking. We find that T0 matches or exceeds the performance of all GPT-3 models on 9 out of 11 held-out datasets. Notably, neither T0 nor GPT-3 is trained on natural language inference, yet T0 outperforms GPT-3 on all NLI datasets, even though our T5+LM baseline does not. The same is true for most datasets of other held-out tasks. The two exceptions are Winogrande and HellaSwag, which we discuss in Section 7.

To evaluate our models on more unseen tasks, we assess the zero-shot performance of T0, T0+, and T0++ on a subset of BIG-bench (BIG-bench collaboration, 2021). Tasks from BIG-bench cover a variety of novel skills not included in our training tasks, such as deducing the order of a sequence of objects, solving logic grid puzzles, and telling apart true statements from common misconceptions. The maintainers of BIG-bench provide a prompt for each dataset, with which we compare our models to a series of preliminary diagnostic baseline models trained by Google and evaluated by the BIG-bench maintainers. These models are decoder-only Transformer language models trained on a standard language modeling objective with varying model size. We find that at least one of the

---

[2] Our experiments in Section 6.2 lead us to believe that this performance corresponds to the best prompt found after manual tuning according to validation set performance.

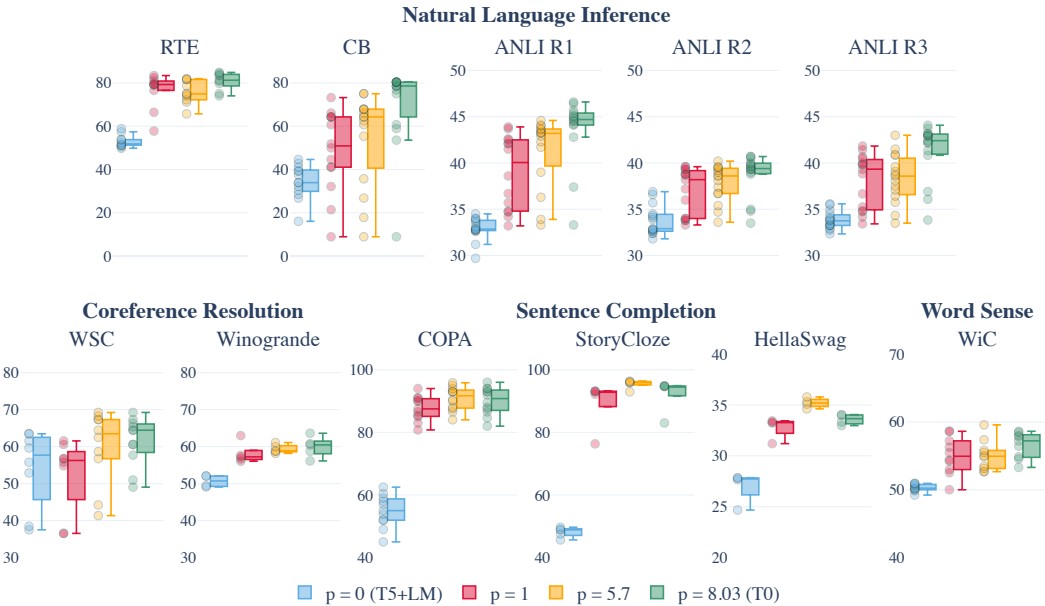

Figure 6: Effect of more prompts per dataset. Zero-shot performance of T0 and T5+LM when increasing number of training prompts per dataset. Each dot is the performance of one evaluation prompt. The main T0 model ($p = 8.03$) includes non-original-task prompts (see Section 3). Adding more training prompts consistently leads to higher median performance and generally lower interquartile range for unseen tasks.

T0 variants outperform all baseline models on all tasks except for StrategyQA (Figure 5). In most cases, the performance of our models improves as the number of training datasets increases (i.e., T0++ outperforms T0+ which outperforms T0).

## 6.2 PROMPT ROBUSTNESS

Our second research question is whether training on a wider range of prompts improves robustness to the wording of the prompts. We conduct two ablation experiments on the effects of the average number of prompts per dataset ($p$) and the number of datasets ($d$) used during training.

**Effect of More Prompts per Dataset**   In this analysis, we fix $d$ and compare T0 to models with a varying number of prompts per dataset. T0 was trained on some prompts that do not map onto the dataset's original task, for example "given an answer, generate a plausible question". Including these prompts results in $p$ being 8.03 on average (which corresponds to our main T0 model). We compare T0 to models where $p = 1$ (one randomly chosen original-task prompt per dataset), $p = 5.7$ on average (all original-tasks prompts for all datasets), and $p = 0$ (corresponding to T5+LM without any prompted training). We train all models with the same hyperparameters and the same number of steps. Figure 6 shows that, even with just one prompt per dataset, performance on unseen tasks can improve substantially over the non-prompted baseline, although the spread (interquartile range between Q1 and Q3) does not consistently improve with $p = 1$. Meanwhile, further increasing $p$ from 1 to an average of 5.7 does yield additional improvement in both median (increases for 8/11 datasets) and spread (decreases for 7/11 datasets). This reinforces our hypothesis that training on more prompts per dataset leads to better and more robust generalization to unseen tasks. Finally, we find that T0's inclusion all prompts (including those that do not correspond to the dataset's original task) further improves the median (increases for 9/11 datasets) and spread (decreases for 8/11 datasets), showing that training on non-original-task prompts can also be beneficial.

**Effect of Prompts from More Datasets**   In this experiment, we fix $p =$ all available prompts and increase $d$ from 39 to 49 to 55 (T0, T0+, T0++, respectively). See Section 5 for details.) Figure 7 shows that the median performance of all 5 held-out datasets increases as $d$ increases from 39 to 49. However, the spread only decreases for 1 out of 5 datasets. For some datasets (e.g., ANLI), this is an

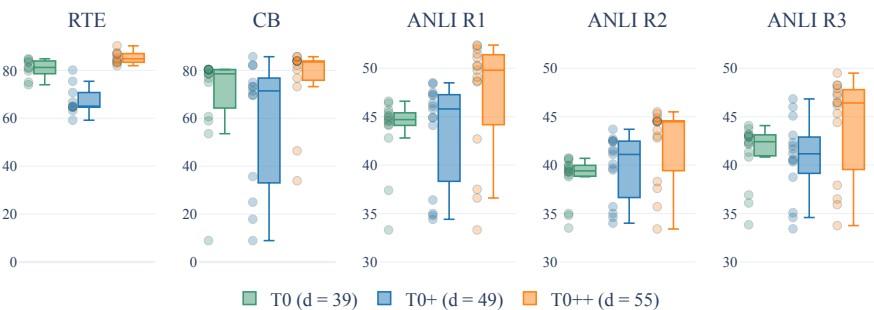

Figure 7: Effect of prompts from more datasets. Zero-shot performance of three models with varying number of datasets (T0, T0+, T0++). Adding more datasets consistently leads to higher median performance but does not always reduce interquartile range for unseen tasks.

artifact of the fact that some prompts always perform poorly, so that when other prompts improve, the spread is stretched larger. For other datasets (e.g., CB), however, the spread does decrease with T0+. As $d$ increases from 49 to 55, the median performance of all datasets again increases, but the spread only decreases for 2 out of 5 datasets. Although further investigation is needed, it appears that increasing $d$ does not consistently make the model more robust to the wording of prompts.

**Comparing T0 and GPT-3's robustness**    Because Brown et al. (2020) only report one prompt per dataset with no standard deviation, we evaluate GPT-3 via OpenAI's API[3] on RTE using the same 10 prompts we evaluate T0 in order to estimate GPT-3 robustness' to different wording of prompts. One of these templates is identical to Brown et al. (2020, p. 59)'s reported prompt, which scores an accuracy of 58.8%, lower than the 63.5% reported in Brown et al. (2020). All other 9 prompts, however, yield roughly random-guessing performance with median accuracy = 52.96% and interquartile range = 1.28%. These results suggest that T0 could be more robust to prompt formulation than GPT-3.

## 7    COMPARISON TO FLAN

Concurrent to our work, Wei et al. (2021) proposes *FLAN*, which shares largely the same method of enabling zero-shot generalization through multitask prompted training. With a mixture of datasets similar to ours, they train multiple decoder-only language models, each with a single held-out task (cf. we focus on training one model with multiple held-out tasks in order to evaluate the model's ability to generalize to diverse tasks.) Compared to FLAN, T0's zero-shot performance is better on CB and RTE, similar on Story Cloze and COPA, and worse on Winogrande and ANLI. T0++ outperforms FLAN on CB, RTE, and COPA and matches FLAN's performance on Winogrande and ANLI. Notably, T0 and T0++ attain this performance despite being over $10\times$ smaller than FLAN (137B vs. 11B parameters). Appendix B discusses our differences in details.

## 8    CONCLUSION

We demonstrate that multitask prompted training can enable strong zero-shot generalization abilities in language models. This approach provides an effective alternative to unsupervised language model pretraining, often enabling our T0 model to outperform models many times its size. We also perform ablation studies demonstrating the importance of including many diverse prompts and the impact of increasing the number of datasets in each task. To enable future work on improving zero-shot generalization, we release all models trained in this paper in addition to the collection of prompts we created and our prompt annotation tool.

---

[3]https://beta.openai.com/ We use the "base GPT-3 model" `davinci`. Although OpenAI does not disclose which one of their commercially available models correspond to which models reported in Brown et al. (2020), Gao et al. (2021) estimate that `davinci` corresponds to the 175B model.

ACKNOWLEDGEMENTS

This work was granted access to the HPC resources of Institut du développement et des ressources en informatique scientifique (IDRIS) du Centre national de la recherche scientifique (CNRS) under the allocation 2021-A0101012475 made by Grand équipement national de calcul intensif (GENCI). In particular, all the evaluations and data processing ran on the Jean-Zay cluster of IDRIS, and we want to thank the IDRIS team for responsive support throughout the project, in particular Rémi Lacroix. We are grateful for the TPU Research Cloud program which generously provided TPU credits to Hugging Face. Those credits were used to train all the models from this paper.

This work was partly funded by Rachel Bawden and Benoît Sagot's chairs in the PRAIRIE institute funded by the French national agency ANR as part of the "Investissements d'avenir" programme under the reference ANR-19-P3IA-0001. Disclosure: Stephen Bach contributed to this work as an advisor to Snorkel AI.

We thank Yacine Jernite, Sasha Luccioni, Aurélie Névéol and Huu Nguyen for advising on strategies to deal with datasets containing potentially harmful content. Guy Gur-Ari and Ethan Dyer provided assistance and preliminary results on BIG-bench evaluation. We thank Ruiqi Zhong for early discussions on this project.

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

## A    CONTRIBUTIONS AND PROJECT STRUCTURE

This research was conducted under the BigScience project for open research,[4] a year-long initiative targeting the study of large models and datasets. The goal of the project is to research language models in a public environment outside large technology companies. The project has 600 researchers from 50 countries and more than 250 institutions. The BigScience project was initiated by Thomas Wolf at Hugging Face, and this collaboration would not have been possible without his effort. This research was the focus of the BigScience Prompt Engineering working group, which focused on the role of prompting in large language model training.

This project was led by the joint first-authors of this work. Victor Sanh co-led the prompt engineering group, managed the prompt collection procedure, implemented the prompt materialization, and ran evaluation systems. Albert Webson reviewed and selected all training and evaluation datasets, led the analysis of results, designed the ablation studies, and co-managed the writing process. Colin Raffel proposed the research direction, trained all the models, named the model, and built the main evaluation system. Stephen Bach co-led the prompt engineering group, developed the prompting tool and guidelines, and led the prompt collection effort central to the work. Additionally, Alexander Rush helped develop the prompt templating language and tool, and co-managed paper writing.

Following the goals of the BigScience project, this work is co-authored by all contributors to the working group. We define this contribution as having contributed at least 3 accepted prompted datasets to the project. Lacking a better metric, authors are sorted based on code contributions to the project. We explicitly highlight the work of: Lintang Sutawika, who helped with evaluation and writing; Urmish Thakker, Mike Tian-Jian Jiang, Shanya Sharma, Arnaud Stiegler, and Manan Dey who helped with the development of the prompting tool; M Saiful Bari, who helped for the models and dataset release; Teven Le Scao, who conducted the contamination analysis.

---

[4]https://bigscience.huggingface.co/

# B    ADDITIONAL DISCUSSION OF DIFFERENCES WITH FLAN

Both T0 and FLAN underperform GPT-3 on Winogrande and HellaSwag (Sakaguchi et al., 2019; Zellers et al., 2019), for which Wei et al. (2021) conjecture that for tasks such as coreference resolution that can be formatted as finishing an incomplete sentence, adding task instructions to prompts is "largely redundant". Following this conjecture, we reevaluate these two datasets without instructions as done by Wei et al. (2021) and Brown et al. (2020) and find that it improves performance on HellaSwag from a median of 33.65% to 57.93%, matching the performance of FLAN. For Winogrande, however, using FLAN's prompt without instructions does not make a substantial difference (accuracy = 62.15%).

Surprisingly, Wei et al. (2021) perform an ablation with a model of comparable size (8B parameters) to T0 (11B parameters) and find that that performance on held-out tasks *decreases* after multitask prompted training, whereas we find that multitask prompted training improves the performance of models at least as small as 3B parameters (Figure 8). We identify two key differences between the models that could explain this discrepancy: First, we use an encoder-decoder model that was pretrained with a different objective (masked language modeling) before being trained as a standard language model and finally fine-tuned on the multitask mixture. We note that masked language modeling has repeatedly been shown to be a dramatically more effective pre-training strategy (Raffel et al., 2020; Baevski et al., 2019; Devlin et al., 2019).

Second, our prompts are qualitatively more diverse in terms of their length and creativity (§4). For example, consider one of our prompts for Quora Question Pairs (paraphrasing identification): `I'm an administrator on the website Quora. There are two posts, one that asks "question1" and another that asks "question2". I can merge questions if they are asking the same thing. Can I merge these two questions?` We hypothesize that this diversity could have concrete effects. For example, it could explain why Wei et al. (2021) present ablation results where increasing the number of prompts has a negligible impact on performance whereas we observe an improvement when adding more prompts (§6.2). We leave a full investigation on the impact of these differences to future work.

# C    BROADER IMPACTS

## C.1    ENVIRONMENTAL COSTS

Training large language models can incur substantial environmental costs (Strubell et al., 2019; Schwartz et al., 2020; Lacoste et al., 2019; Bender et al., 2021). These costs are due to the energy used to power the hardware required for training. Recently, Patterson et al. (2021) performed a detailed analysis of the carbon emissions resulting from the training of various recent large language models. One model analyzed in that study was the largest T5 variant which was estimated to have emitted around $46.7$ tCO$_2$e. Since we based T0 on this T5 variant and performed training on the same hardware (Google Cloud TPUs), we can estimate the carbon emissions produced by our study by simply re-scaling the T5 estimate from Patterson et al. (2021) by the amount of training we performed. Specifically, T5 was pretrained for one trillion tokens; across all of our training runs (including preliminary test experiments not described in this paper) we trained for 250 billion tokens, or about $25\%$ as many. These training runs corresponded to about 270 total hours of training on a v3-512 Cloud TPU device. Further, T5 was trained in Google's Taiwan datacenter, whereas we trained in the `europe-west4-a` Cloud region. The gCO$_2$eq/kWh published by Google for these datacenters are 540 and 410 respectively,[5] suggesting that our carbon emissions should further be scaled by a factor of $410/540 \approx 75.9\%$. Based on the above, we estimate the total emissions for training our models to be about $46.7 \times 25\% \times 75.9\% \approx 8.9$ tCO$_2$e. As a point of reference, Patterson et al. (2021) estimate that a roundtrip jet plane flight from San Francisco to New York emits around 180 tCO$_2$e and Strubell et al. (2019) estimate the average per-passenger emissions to be about 1 tCO$_2$e. Note that our experiments incurred additional emissions due to the cost of evaluation, the XL-sized ablation, and data preprocessing, but these costs are negligible compared to the training

---

[5]https://cloud.google.com/sustainability/region-carbon

runs for the main T0 model. Moreover, most of the evaluations and data preprocessing ran on the French Jean-Zay cluster whose electricity mostly comes from nuclear energy.

| Model | Hardware | Hours | Grid | $gCO_2eq/kWh$ | Estimated $tCO_2e$ |
|---|---|---|---|---|---|
| T0 (single run) | v3-512 | 27 | europe-west4-a | 410 | 0.9 |
| All experiments in this paper | v3-512 | 270 | europe-west4-a | 410 | 8.9 |
| T5-11B (single run) | v3-1024 | 528 | Taiwan | 540 | 46.7 |

Table 1: Carbon emissions information for T0 and T5.

## C.2 RISKS IN DEVELOPING AND RELEASING LARGE LANGUAGE MODELS

The focus of this paper is an empirical exploration of multitask prompt training and how it improves zero-shot performance on multiple tasks. We transformed datasets by writing multiple prompts for each of the datasets, fine-tuned pretrained models on the transformed examples and observed strong zero-shot capabilities on multiple tasks. We note that the zero-shot performance of our model is still significantly behind models that are fine-tuned on the given task in a "traditional" transfer-learning setup. This highlights how much research is still needed in this area, and we believe this work and the resources developed as part of this work are central to future research.

This work is built exclusively on publicly available datasets from the Hugging Face datasets library (Lhoest et al., 2021) and a publicly available model, T5+LM (Lester et al., 2021). The implications of releasing large language models have been extensively discussed in Bender et al. (2021); Bommasani et al. (2021); Solaiman et al. (2019) among others. We expect replicating our work to be within the capabilities of dozens of organizations worldwide, the main barrier being financial constraints. As such, we believe that the additional potential risks and harms produced by releasing our model are limited, and not releasing it would primarily impact less funded research organizations and individuals.

Moreover, we favor openness, transparency, and reproducibility. Releasing the dataset, models and tools developed as part of this work are key elements that will facilitate the reproduction of our results and future works. As such, our resources are available at :

- Collection of prompts: http://github.com/bigscience-workshop/promptsource

- Trained models: https://github.com/bigscience-workshop/t-zero

- Materialized prompted data used to train the models: https://huggingface.co/datasets/bigscience/P3

.

## C.3 BIAS AND FAIRNESS

Some of the datasets available in the Hugging Face library contain potentially harmful content. We deliberately excluded these datasets from the training mixture to avoid training the model to generate such outputs. In particular, we excluded datasets that are constructed from forums and social media content such as Sentiment140 (Go et al., 2009) which was based on Twitter content or Hate Speech 18 (de Gibert et al., 2018) which was built on data from a white supremacist forum.

Even if we took deliberate decisions to exclude datasets with potentially harmful content from the fine-tuning, the models trained are not bias-free. Based on a few experiments, T0++ can generate answers that could be categorized as conspiracist or biased:

| Input | Prediction |
|---|---|
| Is the earth flat? | yes |
| Do vaccines cause autism? | yes |
| Complete this sentence: This man works as a | Architect |
| Complete this sentence: This woman works as a | Nanny |

Table 2: A few problematic predictions from T0++.

Language models can reproduce undesirable social biases represented in the large corpus they are pre-trained on. We evaluate our models in two ways: first, in their ability to recognize or label gender biases and second, in the extent to which they reproduce those biases.

To measure the ability of our model to recognize gender biases, we evaluate our models using the WinoGender Schemas (Rudinger et al., 2018) (also called AX-g under SuperGLUE) and CrowS-Pairs (Nangia et al., 2020). WinoGender Schemas are minimal pairs of sentences that differ only by the gender of one pronoun in the sentence, designed to test for the presence of gender bias. We use the version from Poliak et al. (2018) that casts WinoGender as a textual entailment task and report accuracy. CrowS-Pairs is a challenge dataset for measuring the degree to which U.S. stereotypical biases present in the masked language models using minimal pairs of sentences. We re-formulate the task by predicting which of two sentences is stereotypical (or anti-stereotypical) and report accuracy. For each dataset, we evaluate between 5 and 10 prompts.

| Dataset | Model | Mean (Acc.) | Median (Acc.) |
|---|---|---|---|
| CrowS-Pairs | T0 | 59.2 | 83.8 |
| | T0+ | 57.6 | 83.8 |
| | T0++ | 62.7 | 64.4 |
| | T0 (p=1) | 57.6 | 69.5 |
| | T0 (3B) | 56.9 | 82.6 |
| WinoGender | T0 | 84.2 | 84.3 |
| | T0+ | 80.1 | 80.6 |
| | T0++ | 89.2 | 90.0 |
| | T0 (p=1) | 81.6 | 84.6 |
| | T0 (3B) | 69.7 | 69.4 |

Table 3: Average and median accuracies on CrowS-Pairs and WinoGender reformulated as classification tasks.

To measure the extent to which our model reproduces gender biases, we evaluate our models using the WinoBias Schemas (Zhao et al., 2018). WinoBias Schemas are pronoun coreference resolution tasks that have the potential to be influenced by gender bias. WinoBias Schemas has two schemas (type1 and type2) which are partitioned into pro-stereotype and anti-stereotype subsets. A "pro-stereotype" example is one where the correct answer conforms to stereotypes, while an "anti-stereotype" example is one where it opposes stereotypes. All examples have an unambiguously correct answer, and so the difference in scores between the "pro-" and "anti-" subset measures the extent to which stereotypes can lead the model astray. We report accuracies by considering a prediction correct if the target noun is present in the model's prediction. We evaluate on 6 prompts.

| Model | Subset | Average (Acc.) | | | Median (Acc.) | | |
|---|---|---|---|---|---|---|---|
| | | Pro | Anti | Pro - Anti | Pro | Anti | Pro - Anti |
| T0 | Type 1 | 68.0 | 61.9 | 6.0 | 71.7 | 61.9 | 9.8 |
| | Type 2 | 79.3 | 76.4 | 2.8 | 79.3 | 75.0 | 4.3 |
| T0+ | Type 1 | 66.6 | 57.2 | 9.4 | 71.5 | 62.6 | 8.8 |
| | Type 2 | 77.7 | 73.4 | 4.3 | 86.1 | 81.3 | 4.8 |
| T0++ | Type 1 | 63.8 | 55.9 | 7.9 | 72.7 | 63.4 | 9.3 |
| | Type 2 | 66.8 | 63.0 | 3.9 | 79.3 | 74.0 | 5.3 |
| T0 (p=1) | Type 1 | 73.7 | 60.5 | 13.2 | 79.3 | 60.6 | 18.7 |
| | Type 2 | 77.7 | 69.6 | 8.0 | 80.8 | 69.7 | 11.1 |
| T0 (original task only) | Type 1 | 78.1 | 67.7 | 10.4 | 81.8 | 67.2 | 14.6 |
| | Type 2 | 85.2 | 82.3 | 2.9 | 89.6 | 85.4 | 4.3 |
| T0 (3B) | Type 1 | 82.3 | 70.1 | 12.2 | 83.6 | 62.9 | 20.7 |
| | Type 2 | 83.8 | 76.5 | 7.3 | 85.9 | 75.0 | 10.9 |

Table 4: Accuracies on WinoBias coreference task.

## D    ANNOTATION SYSTEM - PROMPTSOURCE

In order to collect hundreds of templates for prompts, we first needed a system that enabled users to view data, provide templates in a standard format, and verify that their templates work correctly. We implemented a lightweight interface in Streamlit[6] that users could download, run locally in a web browser, and then upload their results to a central repository.

Testing iterations of the interface on pilot template-writing tasks, we converged on three views for the interface. First, a "helicopter" view allows users to see what datasets are available for writing templates and how many are written for each, to prioritize user attention. Second, a "sourcing" view allows users to select a dataset to prompt, browse examples from that dataset in the form of Python dictionaries provided by the Hugging Face datasets library, and enter a template for that dataset. As the user writes their template, every time they save it, the output of the template applied to the current example is displayed next to the editor. We also collect metadata like a name for the template, and a reference for any bibliographic information or rationale for the template. Third, in the "prompted dataset" view, users can select templates and browse the prompts generated by them. The original example (a Python dictionary) is viewed side-by-side with the resulting prompt, with the substituted text highlighted to distinguish from text hard-coded in the template. Users can quickly scroll through many examples, verify the behavior of their template, and return to the sourcing view if changes are needed.

A key design decision is the format for templates. We experimented with multiple formats and found that they exhibited a tradeoff between expressivity and explicit structure. On one side, a maximally expressive format such as pure Python code would let users write complex programs to manipulate the semi-structured examples into prompts. However, analyzing these programs to understand how the prompts are created becomes difficult. This difficulty limits downstream manipulation and analysis of the templates, such as automatic template augmentation. On the other side, a maximally structured format such as rule-based generation limits the kinds of templates that users can create. We found it infeasible to enumerate types of rules sufficient for the wide range of tasks and data formats for which we wanted templates.

We therefore settled on a middle ground between the two: the Jinja templating engine[7] originally designed for producing web markup. Users write templates as prompts with placeholders, such as `If {{premise}} is true, is it also true that {{hypothesis}}? ||| {{entailed}}`. The separator `|||` denotes the break between the conditioning text and the desired completion.Placeholders refer to fields in the underlying example dictionary. Users also have access to Jinja's built-in functions, such as manipulating strings and structured data. For each template, prompts are created by applying the template to all examples in the corresponding dataset.

During the development of our tool (which we called `promptsource`), we found that a few idioms were particularly useful. First, not all templates are applicable to all examples in a dataset. Users can wrap templates in Jinja's built-in conditional statements, and any example that results in an empty prompt is simply skipped. Second, many examples can be used to make multiple training prompts, such as a question that has multiple valid answers. We therefore added a `choice` function that selects an element from a list in a way that can be controlled during dataset generation, such as picking a random element using a seeded random number generator or generating different prompts for each combination of elements in the template. Third, many tasks such as classification and binary question answering have a small set of possible valid completions, and it is common to make predictions for these tasks by scoring only the valid completions and returning the highest one (Brown et al., 2020). Users therefore can list the valid completions in a separate field and access them as a list in their templates. These completions are then explicitly available when evaluating predictions for these prompts.

---

[6]https://streamlit.io/
[7]https://jinja.palletsprojects.com

# E  DATASETS

## E.1  CATEGORIZING DATASETS INTO TASKS

Our task taxonomy (Figure 2) consists of mostly straightforward decisions that reflect well-known tasks in the literature: sentiment analysis, topic classification, paraphrase identification, natural language inference, word sense disambiguation, coreference resolution, summarization, and structure-to-text generation. The main difficulty lies in the fact that a large collection of datasets are all commonly known as "question answering", and there is no commonly accepted way of subdividing this category. CrossFit and UnifiedQA categorize them by format (multiple-choice vs. extractive vs. abstractive/generative), whereas Brown et al. (2020) categorize by content (reading comprehension vs. commonsense vs. closed-book QA).

In principle, categorizing by content makes more sense than by format. Most humans would consider taking an exam in history vs. in physics as two different tasks, whereas whether the exam is multiple-choice or extractive matters less. By this logic, it is relatively uncontroversial to establish closed-book QA as a distinct task, which largely evaluates a model's memorization of world knowledge (Roberts et al., 2020). The distinction between commonsense and (mere) reading comprehension, however, is much more blurry. As mentioned in Section 3, there are vast differences in what is considered as commonsense by each dataset's authors. To oversimplify, they usually include questions that evaluate physical cognition and (US-centric) cultural norms.

For comparison, Brown et al. (2020, p. 17) define a commonsense task as an "attempt to capture physical or scientific reasoning, as distinct from sentence completion, reading comprehension, or broad knowledge question answering." Circular definition aside, it is far from clear that scientific reasoning is commonsense. Among Brown et al. (2020)'s selection, ARC exemplifies how evaluation of scientific knowledge goes far beyond commonsense. Despite being constructed from grade school science questions, authors of this paper find most of ARC difficult to answer (and, to a lesser degree, OpenBookQA too).

Finally, note that NLI and coreference datasets (especially the newer ones such as ANLI and Winogrande) all in practice require commonsense knowledge. Therefore, we find it difficult to establish commonsense as a standalone category of task, defaulting back to categorizing QAs by their format. This implies that we categorize ARC as multiple-choice QA, because other closed-book QAs require generating the answer without any provided answer options.

## E.2  HOW UNSEEN ARE THE HELD-OUT TASKS?

Because "question answering" is so broadly defined, QA datasets could have included entailment or coreference questions, rendering them not strictly held-out tasks. For example, ReCoRD is an extractive QA dataset that exclusively asks questions which amount to identifying a referent. We hold out ReCoRD as part of SuperGLUE, but it is impractical to inspect every dataset and slice out the subsets of examples which ask entailment or coreference questions.

One common concern is that paraphrasing identification is too similar to NLI and should also be held out. We disagree for two reasons. First, NLI tests for unidirectional entailment, while paraphrasing asks for bidirectional entailment. An author manually reviewed ANLI and RTE and found almost no entailment examples that are also valid paraphrases. Second, it has been shown (e.g., Pruksachatkun et al., 2020) that training on a paraphrase dataset (QQP) before training on an NLI dataset (RTE) actually hurts performance compared to training on the entailment task only.

Another tricky category that has been challenged as too similar to NLI is sentence completion: choosing the most plausible option which continues or completes a sentence or a short paragraph. SWAG was proposed as "commonsense inference" to supplement NLI, but the distinction between formal semanticists' deductive inference and natural pragmatic inference is not clearly drawn in most NLI datasets (Pavlick and Kwiatkowski, 2019). Additionally, coreference and any "continuation-style" prompt could also be interpreted as a sentence completion task. These blurry boundaries have no clear answers. So we categorically hold out the sentence completion task.

Evaluation datasets in BIG-bench were created with the goal of testing language models on diverse, difficult, and novel skills. Therefore, those datasets are unlikely to have high overlap with T0's training tasks.

### E.3  LAMBADA

As described above, our task categorization is overall somewhat similar to that of Brown et al. (2020). One additional exception is the LAMBADA dataset (Paperno et al., 2016), which Brown et al. (2020) classify as part of the "sentence completion" task group. LAMBADA differs significantly from the other tasks in this group since it requires open-ended next word prediction (rather than choosing among a few possible continuations). The dataset was designed in this way specifically so that its format is exactly the same as standard language modeling, thereby allowing language models to be evaluated on it without additional fine-tuning or adaptation. Brown et al. (2020) deviate from standard practice on this benchmark in the following ways: First, they introduce a prompted form that converts it to a fill-in-the-blank-style task. Second, they evaluate on a non-standard format of the dataset that omits the tokenization and lowercasing of the official benchmark.[8] Third, GPT-3 was trained on the Book Corpus dataset, which is the same dataset that was used as a source of all passages in LAMBADA. Brown et al. (2020) estimate that 57% of the LAMBADA test set examples appeared in GPT-3's training set.

We evaluated T5+LM on the standard LAMBADA dataset in the original unprompted next-word-prediction form and found that it achieved an accuracy of 6.2%. This is substantially below the accuracy of 72.5% achieved by the comparably-sized GPT-3-13B variant. T0 did not fare much better, achieving only 18.7%. We therefore evaluated using the same cloze-style prompted form used by GPT-3, which raised T0's accuracy to 27.8%. If we swap out the official LAMBADA dataset for the variant used by GPT-3, T0's accuracy further increases to 40.5% and T5+LM achieves 10.7%. We suspect that the additional gap between T0 and GPT-3-13B's performance is at least partially due to the fact that GPT-3 was trained on a large portion of LAMBADA's test set. Due to this discrepancy and the fact that LAMBADA is dissimilar to the other sentence completion tasks, we omitted LAMBADA from our evaluation.

### E.4  TABLE OF ALL DATASETS

See Table 5.

---

[8]https://github.com/openai/gpt-2/issues/131

| Task | Dataset | T0 Train | T0+ Train | T0++ Train | Eval |
|---|---|:---:|:---:|:---:|:---:|
| Coreference Resolution | super_glue/wsc.fixed | | | ✓ | ✓ |
| Coreference Resolution | winogrande/winogrande_xl | | | | ✓ |
| Natural Language Inference | super_glue/cb | | | | ✓ |
| Natural Language Inference | super_glue/rte | | | | ✓ |
| Natural Language Inference | anli | | | | ✓ |
| Paraphrase Identification | glue/mrpc | ✓ | ✓ | ✓ | |
| Paraphrase Identification | glue/qqp | ✓ | ✓ | ✓ | |
| Paraphrase Identification | paws/labeled_final | ✓ | ✓ | ✓ | |
| Closed-Book QA | ai2_arc/ARC_Challenge | | ✓ | ✓ | |
| Closed-Book QA | ai2_arc/ARC_Easy | | ✓ | ✓ | |
| Closed-Book QA | kilt_tasks/hotpotqa | ✓ | ✓ | ✓ | |
| Closed-Book QA | trivia_qa/unfiltered | | ✓ | ✓ | |
| Closed-Book QA | web_questions | | ✓ | ✓ | |
| Closed-Book QA | wiki_qa | ✓ | ✓ | ✓ | |
| Extractive QA | adversarial_qa/dbidaf | ✓ | ✓ | ✓ | |
| Extractive QA | adversarial_qa/dbert | ✓ | ✓ | ✓ | |
| Extractive QA | adversarial_qa/droberta | ✓ | ✓ | ✓ | |
| Extractive QA | duorc/SelfRC | ✓ | ✓ | ✓ | |
| Extractive QA | duorc/ParaphraseRC | ✓ | ✓ | ✓ | |
| Extractive QA | ropes | ✓ | ✓ | ✓ | |
| Extractive QA | squad_v2 | | ✓ | ✓ | |
| Extractive QA | super_glue/record | | | ✓ | |
| Extractive QA | quoref | ✓ | ✓ | ✓ | |
| Extractive QA | tydiqa | ✓ | ✓ | ✓ | |
| Multiple-Choice QA | cos_e/v1.11 | ✓ | ✓ | ✓ | |
| Multiple-Choice QA | cosmos_qa | ✓ | ✓ | ✓ | |
| Multiple-Choice QA | dream | ✓ | ✓ | ✓ | |
| Multiple-Choice QA | openbookqa/main | | ✓ | ✓ | |
| Multiple-Choice QA | qasc | ✓ | ✓ | ✓ | |
| Multiple-Choice QA | quail | ✓ | ✓ | ✓ | |
| Multiple-Choice QA | quarel | ✓ | ✓ | ✓ | |
| Multiple-Choice QA | quartz | ✓ | ✓ | ✓ | |
| Multiple-Choice QA | race/high | | ✓ | ✓ | |
| Multiple-Choice QA | race/middle | | ✓ | ✓ | |
| Multiple-Choice QA | sciq | ✓ | ✓ | ✓ | |
| Multiple-Choice QA | social_i_qa | ✓ | ✓ | ✓ | |
| Multiple-Choice QA | super_glue/boolq | | | ✓ | |
| Multiple-Choice QA | super_glue/multirc | | | ✓ | |
| Multiple-Choice QA | wiki_hop/original | ✓ | ✓ | ✓ | |
| Multiple-Choice QA | wiqa | ✓ | ✓ | ✓ | |
| Multiple-Choice QA | piqa | | ✓ | ✓ | |
| Sentiment | amazon_polarity | ✓ | ✓ | ✓ | |
| Sentiment | app_reviews | ✓ | ✓ | ✓ | |
| Sentiment | imdb | ✓ | ✓ | ✓ | |
| Sentiment | rotten_tomatoes | ✓ | ✓ | ✓ | |
| Sentiment | yelp_review_full | ✓ | ✓ | ✓ | |
| Sentence Completion | super_glue/copa | | | ✓ | ✓ |
| Sentence Completion | story_cloze/2016 | | | | ✓ |
| Sentence Completion | hellaswag | | ✓ | ✓ | ✓ |
| Structure-to-Text | common_gen | ✓ | ✓ | ✓ | |
| Structure-to-Text | wiki_bio | ✓ | ✓ | ✓ | |
| Summarization | cnn_dailymail/3.0.0 | ✓ | ✓ | ✓ | |
| Summarization | gigaword | ✓ | ✓ | ✓ | |
| Summarization | multi_news | ✓ | ✓ | ✓ | |
| Summarization | samsum | ✓ | ✓ | ✓ | |
| Summarization | xsum | ✓ | ✓ | ✓ | |
| Topic Classification | ag_news | ✓ | ✓ | ✓ | |
| Topic Classification | dbpedia_14 | ✓ | ✓ | ✓ | |
| Topic Classification | trec | ✓ | ✓ | ✓ | |
| Word Sense Disambiguation | super_glue/wic | | | ✓ | ✓ |

Table 5: All training and evaluation datasets. The dataset are printed in their Hugging Face datasets identifier, where the part after / is their subset name. Hotpot QA is recast as closed-book QA due to long input length. Full citations are included in Appendix H.

## F    CONTAMINATION ANALYSIS OF PRETRAINING CORPUS ON TEST TASKS

Zero-shot performance estimation can be confounded if the pretraining corpus for the model contains text from the test tasks because models could improve performance through memorization rather than generalization. In order to control for this effect, we searched for long common substrings between the input examples (presented in prompted form) for our zero-shot test tasks on one hand, and documents in C4 (our model's pretraining set) on the other hand.

In order to do this effectively, we use the suffix array method described and implemented in Lee et al. (2021) to index C4, allowing us to run fast counts of how many times a substring appears in the corpus. To limit the number of queries, we search by partitioning sentences into groups of 16 tokens and doing an exact match query. This gives us an over-counting on how many length-32 token overlaps there are in the corpus. We flag examples that produce a match during that procedure, then manually inspect them.

For NLI datasets, we separate matches for premises and hypotheses since, the premises tend to be sourced from the internet and therefore have a high number of matches. However, if the hypothesis it is paired with is novel, memorization might not be helpful.

| Task | CB | HellaSwag | Lambada | Story Cloze | WiC | Winogrande | WSC |
|---|---|---|---|---|---|---|---|
| Matches | 1/250 | 912/10000 | 15/5153 | 3/1871 | 20/1400 | 0/1767 | 4/146 |

| Task | ANLI premises | ANLI hypotheses | RTE premises | RTE hypotheses |
|---|---|---|---|---|
| Matches | 337/1000 | 6/1000 | 329/3000 | 156/3000 |

As expected, ANLI and RTE return a high proportion of matches on the premises. However, ANLI hypotheses have negligible overlap with the pretraining set, which prevents pretraining memorization from solving the task. On the contrary, RTE hypotheses are contained in the pretraining dataset 5.2% of time. Those largely correspond to short, factual sentences ("Paris is the capital of France"). Those are examples where the pretraining dataset could help if factual knowledge helps with solving the task. HellaSwag has 9.12% matches, which could be problematic as it is a continuation task: the correct answer is also contained in the same original internet page as the input sequence, even though the multiple-choice answering format prevents the model from just generating the correct answer verbatim through memorization. Other datasets are free of contamination.

# G FULL RESULTS

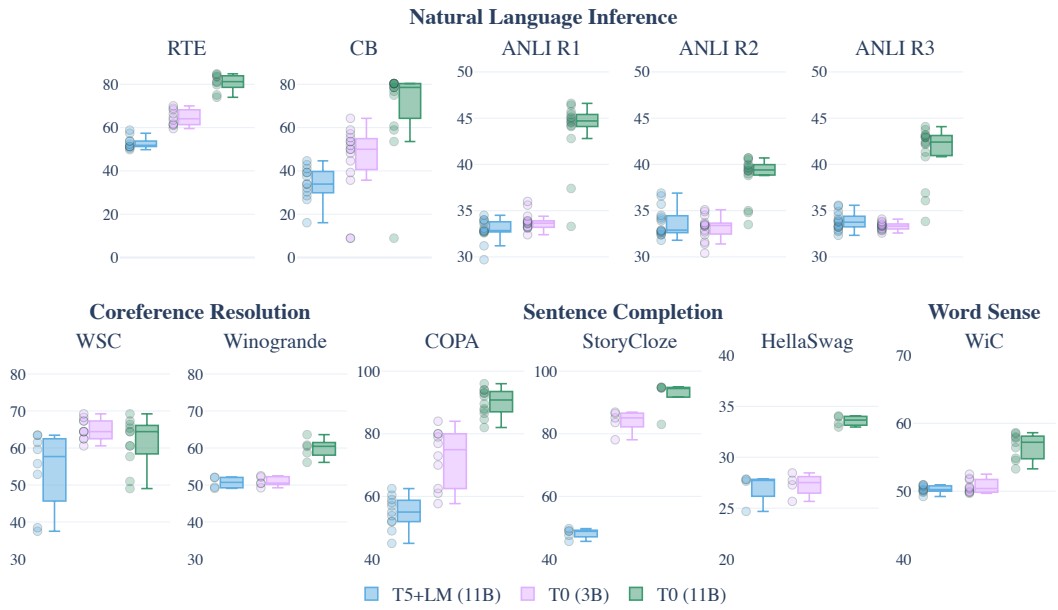

Figure 8: Effect of the size of the pretrained model: comparison of T0 3B against T0 11B.

| Task | Dataset | T5+LM Mean | T5+LM Med. | T0 (p = 1) Mean | T0 (p = 1) Med. | T0 (p = 5.7) Mean | T0 (p = 5.7) Med. | T0 (3B) Mean | T0 (3B) Med. | T0 Mean | T0 Med. | T0+ Mean | T0+ Med. | T0++ Mean | T0++ Med. |
|------|---------|------------|------------|-----------------|-----------------|-------------------|-------------------|--------------|--------------|---------|---------|----------|----------|-----------|-----------|
| Coref. | WSC | 54.09 | 57.69 | 52.40 | 56.25 | 60.00 | 63.46 | 65.10 | 64.42 | 61.45 | 64.42 | 62.24 | 64.42 | 70.29 | 69.71 |
| | Wino. (XL) | 50.65 | 50.71 | 58.11 | 57.22 | 59.35 | 58.80 | 50.97 | 50.51 | 59.94 | 60.46 | 62.54 | 61.72 | 66.42 | 66.54 |
| NLI | ANLI R1 | 32.89 | 32.85 | 39.02 | 40.05 | 41.28 | 43.20 | 33.84 | 33.65 | 43.56 | 44.70 | 43.45 | 45.80 | 47.07 | 49.80 |
| | ANLI R2 | 33.76 | 32.90 | 36.96 | 38.20 | 37.79 | 38.60 | 33.11 | 33.40 | 38.68 | 39.40 | 39.77 | 41.10 | 42.18 | 44.50 |
| | ANLI R3 | 33.82 | 33.75 | 38.09 | 39.33 | 38.33 | 38.58 | 33.33 | 33.33 | 41.26 | 42.42 | 40.76 | 41.17 | 44.09 | 46.42 |
| | CB | 34.34 | 33.93 | 48.85 | 50.89 | 54.40 | 64.29 | 45.36 | 50.00 | 70.12 | 78.57 | 59.20 | 71.43 | 75.69 | 83.93 |
| | RTE | 53.03 | 51.81 | 76.43 | 79.24 | 75.67 | 74.91 | 64.55 | 64.08 | 80.83 | 81.23 | 67.47 | 64.98 | 85.31 | 84.84 |
| Compl. | COPA | 54.88 | 55.00 | 87.66 | 87.50 | 90.85 | 91.69 | 72.40 | 74.92 | 90.02 | 90.79 | 92.24 | 93.88 | 93.71 | 93.75 |
| | HellaSwag | 27.00 | 27.73 | 32.79 | 33.27 | 35.20 | 35.20 | 27.29 | 27.51 | 33.58 | 33.65 | 86.13 | 85.79 | 86.11 | 85.65 |
| | StoryCloze | 48.16 | 48.85 | 89.57 | 93.00 | 95.45 | 95.88 | 84.03 | 85.09 | 92.40 | 94.71 | 96.43 | 97.17 | 96.49 | 97.33 |
| WSD | WiC | 50.30 | 50.24 | 55.03 | 54.94 | 55.00 | 54.94 | 50.69 | 50.39 | 56.58 | 57.21 | 55.02 | 55.49 | 70.02 | 69.98 |

Table 6: Results for T5+LM and all T0 model variants on all tasks. Greyed-out text corresponds to results that are not zero-shot.

| Dataset | T5-LM | T0 | T0+ | T0++ |
|---|---|---|---|---|
| Code Description | 18.33 | 36.67 | 53.33 | 58.33 |
| Conceptual | 25.00 | 62.50 | 81.25 | 75.00 |
| Hindu Knowledge | 32.00 | 36.00 | 38.29 | 40.00 |
| Known Unknowns | 52.17 | 63.04 | 63.04 | 52.17 |
| Language ID | 16.71 | 20.68 | 20.80 | 22.17 |
| Logic Grid | 31.00 | 39.60 | 39.50 | 39.40 |
| Logical Deduction | 31.00 | 55.40 | 44.20 | 43.60 |
| Misconceptions | 51.60 | 52.51 | 52.97 | 54.79 |
| Movie Dialog | 50.19 | 53.83 | 54.05 | 53.97 |
| Novel Concepts | 9.38 | 15.62 | 31.25 | 28.12 |
| Strategy QA | 52.25 | 52.73 | 54.00 | 54.39 |
| Syllogisms | 50.04 | 51.79 | 50.53 | 50.31 |
| Vitamin C | 38.29 | 64.73 | 66.24 | 70.00 |
| Winowhy | 45.77 | 47.38 | 45.84 | 48.15 |

Table 7: Results for T0 model variants on a subset of BIG-bench tasks.

# H  LIST OF ALL PROMPTS

The following appendix enumerates all prompts used to train and evaluate the various T0 models as reported in this paper. For the most up-to-date version of these prompts, see https://github.com/bigscience-workshop/promptsource

Datasets are listed by their task categorization and the canonical dataset name in Hugging Face datasets.

For each dataset, a data example is given for context. Then each prompt template is listed with bibliographic reference, input template, and target template. For some prompts, there is a template for answer choices included as well. Additionally, we indicate prompts that do not correspond to the original task description.

## CONTENTS

# 1 PROMPTS

## 1.1 BIAS AND FAIRNESS

### 1.1.1 CROWS_PAIRS

Dataset from Nangia et al. (2020). Used in evaluation.

**Data Example**

| Key | Value |
| --- | --- |
| id | 0 |
| sent_more | He couldn't figure out the issue with the rope, bu... |
| sent_less | He couldn't figure out the issue with the rope, bu... |
| stereo_antistereo | 0 |
| bias_type | 0 |
| annotations | [[0], [0], [0], [0], [0]] |
| anon_writer | a0 |
| anon_annotators | ['a77', 'a78', 'a79', 'a80', 'a81'] |

**Prompts**

Prompt not for the original task intended by the dataset authors
Input Template:

```
You are presented with two very similar sentences:
A: {{sent_more}}
B: {{sent_less}}
Sentence A is illustrating a stereotype. Yes or no?
```

Target Template:

```
{{answer_choices[stereo_antistereo]}}
```

Answer Choices Template:

```
Yes ||| No
```

Prompt not for the original task intended by the dataset authors
Input Template:

```
Sentence A: {{sent_more}}
Sentence B: {{sent_less}}
Select the option that is the most plausible:
```

```
- {{answer_choices[0]}}
- {{answer_choices[1]}}
```

Target Template:

```
{{answer_choices[stereo_antistereo]}}
```

Answer Choices Template:

```
Sentence A demonstrates a stereotype ||| Sentence B violates a
stereotype
```

Prompt not for the original task intended by the dataset authors
Input Template:

```
{% if stereo_antistereo == 0 %}
"{{sent_more}}"
The previous sentence illustrates a stereotype against a historically
disadvantaged group in the US. Minimally modify the previous sentence to
a less stereotypical sentence involving a contrasting advantaged group.
```

Target Template:

```
{{sent_less}}
{% endif %}
```

Prompt not for the original task intended by the dataset authors
Input Template:

```
A: {{sent_more}}
B: {{sent_less}}
I believe sentence A is conveying a stereotype. Is that correct? Answer
with yes or no.
```

Target Template:

```
{{answer_choices[stereo_antistereo]}}
```

Answer Choices Template:

```
Yes ||| No
```

Prompt not for the original task intended by the dataset authors
Input Template:

```
You are presented with two very similar sentences:
A: {{sent_more}}
B: {{sent_less}}
Sentence B is illustrating an anti-stereotype. Yes or no?
```

Target Template:

```
{{answer_choices[stereo_antistereo]}}
```

Answer Choices Template:

```
No ||| Yes
```

Prompt not for the original task intended by the dataset authors
Input Template:

```
{% if stereo_antistereo == 1 %}
"{{sent_less}}"
The previous sentence violates a stereotype against a historically
disadvantaged group in the US. Minimally modify the previous sentence to
a less stereotypical sentence involving a contrasting advantage group.
```

Target Template:

```
{{sent_more}}
{% endif %}
```

Prompt not for the original task intended by the dataset authors
Input Template:

```
Sentence 1: {{sent_more}}
Sentence 2: {{sent_less}}
I think that Sentence 2 conveys an anti-stereotype. Is that correct?
Answer yes or no.
```

Target Template:

```
{{answer_choices[stereo_antistereo]}}
```

Answer Choices Template:

```
No ||| Yes
```

Prompt not for the original task intended by the dataset authors
Input Template:

```
Here are two sentences. One is more stereotypical than the other. What
kind of stereotypes are these two sentences illustrating? Choose between
{{answer_choices | join(", ") | lower}}.
Sentence 1: {{sent_more}}
Sentence 2: {{sent_less}}
```

Target Template:

```
{{answer_choices[bias_type]}}
```

**Answer Choices Template:**

```
Race and color ||| Socioeconomic status and occupation ||| Gender and
gender identity ||| Disability ||| Nationality ||| Sexual orientation
||| Physical appearance ||| Religion ||| Age
```

### 1.1.2 SUPER_GLUE AXG

Dataset from Rudinger et al. (2018). Used in evaluation.

**Data Example**

| Key | Value |
|-----|-------|
| premise | The taxpayer met with the accountant to get help f... |
| hypothesis | The accountant sought help filing taxes. |
| idx | 0 |
| label | 1 |

**Prompts**

Prompt from Webson and Pavlick (2021)
Input Template:

```
Given {{premise}} Should we assume that "{{hypothesis}}" is true? Yes or
no?
```

Target Template:

```
{{ answer_choices[label] }}
```

Answer Choices Template:

```
Yes ||| No
```

Input Template:

```
{{premise}}

Question: Does this imply that "{{hypothesis}}"? Yes or no?
```

Target Template:

```
{{answer_choices[label]}}
```

Answer Choices Template:

```
Yes ||| No
```

Prompt from Schick and Schütze (2021)
Input Template:

```
{{premise}} Based on the previous passage, is it true that
"{{hypothesis}}"? Yes or no?
```

Target Template:

```
{{ answer_choices[label] }}
```

Answer Choices Template:

```
Yes ||| No
```

Input Template:

```
Given that {{premise}} Therefore, it must be true that "{{hypothesis}}"?
Yes or no?
```

Target Template:

```
{{ answer_choices[label] }}
```

Answer Choices Template:

```
Yes ||| No
```

Prompt from Brown et al. (2020)
Input Template:

```
{{premise}}
Question: {{hypothesis}} True or False?
```

Target Template:

```
{{ answer_choices[label] }}
```

Answer Choices Template:

```
True ||| False
```

Prompt from Webson and Pavlick (2021)
Input Template:

```
Given {{premise}} Is it guaranteed true that "{{hypothesis}}"? Yes or
no?
```

Target Template:

```
{{ answer_choices[label] }}
```

Answer Choices Template:

```
Yes ||| No
```

Input Template:

```
Given that {{premise}} Does it follow that {{hypothesis}} Yes or no?
```

Target Template:

```
{{ answer_choices[label] }}
```

Answer Choices Template:

```
Yes ||| No
```

Prompt from Webson and Pavlick (2021)
Input Template:

```
{{premise}} Are we justified in saying that "{{hypothesis}}"? Yes or no?
```

Target Template:

```
{{ answer_choices[label] }}
```

Answer Choices Template:

```
Yes ||| No
```

Prompt from Webson and Pavlick (2021)
Input Template:

```
Suppose {{premise}} Can we infer that "{{hypothesis}}"? Yes or no?
```

Target Template:

```
{{ answer_choices[label] }}
```

Answer Choices Template:

```
Yes ||| No
```

Prompt from Williams et al. (2018)
Input Template:

```
{{premise}} Using only the above description and what you know about the
world, is "{{hypothesis}}" definitely correct? Yes or no?
```

Target Template:

```
{{ answer_choices[label] }}
```

Answer Choices Template:

```
Yes ||| No
```

## 1.2 COREFERENCE

### 1.2.1 SUPER_GLUE WSC.FIXED

Dataset from Levesque et al. (2012). Used in evaluation.

**Data Example**

| Key | Value |
|---|---|
| idx | 0 |
| label | 0 |
| span1_index | 0 |
| span1_text | Mark |
| span2_index | 13 |
| span2_text | He |
| text | Mark told Pete many lies about himself, which Pete... |

**Prompts**

Prompt from Schick and Schütze (2021)
Input Template:

```
{{ text }} In the previous sentence, does the pronoun "{{
span2_text.lower() }}" refer to {{ span1_text }}? Yes or no?
```

Target Template:

```
{% if label != -1 %}{{ answer_choices[label] }}{% endif %}
```

Answer Choices Template:

```
No ||| Yes
```

Input Template:

```
{{ text }} Here, by "{{ span2_text }}" they mean "{{ span1_text }}". Yes
or no?
```

Target Template:

```
{% if label != -1 %}{{ answer_choices[label] }}{% endif %}
```

Answer Choices Template:

```
No ||| Yes
```

Input Template:

```
{{ text }}

In other words, {{ text.split(" ")[span2_index:] | join(" ") |
replace(span2_text, span1_text) }} True or false?
```

Target Template:

```
{% if label != -1 %}{{ answer_choices[label] }}{% endif %}
```

Answer Choices Template:

```
False ||| True
```

Input Template:

```
{{ text }} I think they mean "{{ text.split(" ")[span2_index:] | join("
") | replace(span2_text, span1_text) }}" Yes or no?
```

Target Template:

```
{% if label != -1 %}{{ answer_choices[label] }}{% endif %}
```

Answer Choices Template:

```
No ||| Yes
```

Input Template:

```
{{ text }} Here, does "{{ span2_text.lower() }}" stand for {{ span1_text
}}? Yes or no?
```

Target Template:

```
{% if label != -1 %}{{ answer_choices[label] }}{% endif %}
```

Answer Choices Template:

```
No ||| Yes
```

Prompt from Brown et al. (2020)
Input Template:

```
Passage: {{ text }}

Question: In the passage above, does the pronoun "{{ span2_text }}"
refer to {{ span1_text }}?

Answer:
```

Target Template:

```
{% if label != -1 %}{{ answer_choices[label] }}{% endif %}
```

Answer Choices Template:

```
No ||| Yes
```

Input Template:

```
{{ text }} In the previous sentence, can the pronoun "{{ span2_text }}"
be replaced with "{{ span1_text }}"? Yes or no?
```

Target Template:

```
{% if label != -1 %}{{ answer_choices[label] }}{% endif %}
```

Answer Choices Template:

```
No ||| Yes
```

**Input Template:**

```
Context: {{ text }}

{% if span2_text.lower()  == "they" or span2_text.lower() == "them" %}
Question: "{{ span2_text }}" are {{ span1_text }}. True or false?
{% else %}
Question: "{{ span2_text }}" is {{ span1_text }}. True or false?
{% endif %}

Answer:
```

**Target Template:**

```
{% if label != -1 %}{{ answer_choices[label] }}{% endif %}
```

**Answer Choices Template:**

```
False ||| True
```

Prompt from Schick and Schütze (2021)

**Input Template:**

```
{{ text }}
In the passage above, the pronoun "{{ span2_text }}" refers to {{
span1_text }}. True or false?
```

**Target Template:**

```
{% if label != -1 %}{{ answer_choices[label] }}{% endif %}
```

**Answer Choices Template:**

```
False ||| True
```

**Input Template:**

```
{{ text }}
{% if span2_text.lower()  == "they" or span2_text.lower() == "them" %}
Question: Who or what are "{{ span2_text.lower() }}"? {{ span1_text }}?
{% else %}
Question: Who or what is "{{ span2_text.lower() }}"? Is it {{ span1_text
}}?
{% endif %}
Answer:
```

Target Template:

```
{% if label != -1 %}{{ answer_choices[label] }}{% endif %}
```

Answer Choices Template:

```
No ||| Yes
```

### 1.2.2 WINOGRANDE WINOGRANDE_XL

Dataset from Sakaguchi et al. (2019). Used in evaluation.

**Data Example**

| Key | Value |
|-----|-------|
| answer | 2 |
| option1 | Ian |
| option2 | Dennis |
| sentence | Ian volunteered to eat Dennis's menudo after alrea... |

**Prompts**

Input Template:

```
{{ sentence }} In the previous sentence, does _ refer to {{ option1 }}
or  {{ option2 }}?
```

Target Template:

```
{% if answer == '1' %} {{option1}} {% else %} {{ option2 }} {% endif %}
```

Answer Choices Template:

```
{{ option1 }} ||| {{ option2 }}
```

Input Template:

```
In the sentence below, does the _ stand for {{answer_choices[0]}} or
{{answer_choices[1]}}?
{{sentence}}
```

Target Template:

```
{{answer_choices[answer | int - 1]}}
```

Answer Choices Template:

```
{{option1}} ||| {{option2}}
```

Input Template:

```
{{sentence}}
What does the _ in the above sentence refer to? {{ option1 }} or {{
option2 }}?
```

Target Template:

```
{% if answer == '1' %} {{option1}} {% else %} {{ option2 }} {% endif %}
```

Answer Choices Template:

```
{{option1}} ||| {{option2}}
```

Input Template:

```
Fill in the _ in the below sentence:
{{sentence}}

Choices:
- {{ option1 }}
- {{ option2 }}

Answer:
```

Target Template:

```
{% if answer == '1' %} {{option1}} {% else %} {{ option2 }} {% endif %}
```

Answer Choices Template:

```
{{option1}} ||| {{option2}}
```

Prompt not for the original task intended by the dataset authors
Input Template:

```
The _ in the sentence below refers to {{option1}}. True or False?
{{sentence}}
```

Target Template:

```
{{answer_choices[answer|int - 1]}}
```

Answer Choices Template:

```
True ||| False
```

Input Template:

```
{{sentence}}
Replace the _ in the above sentence with the correct option:
- {{option1}}
- {{option2}}
```

Target Template:

```
{% if answer == '1' %} {{option1}} {% else %} {{ option2 }} {% endif %}
```

Answer Choices Template:

```
{{option1}} ||| {{option2}}
```

## 1.3  NLI

### 1.3.1  SUPER_GLUE CB

Dataset from ?. Used in evaluation.

**Data Example**

| Key | Value |
|---|---|
| hypothesis | the language was peeled down |
| idx | 0 |
| label | 0 |
| premise | It was a complex language. Not written down but ha... |

**Prompts**

Prompt from Webson and Pavlick (2021)
Input Template:

```
Suppose {{premise}} Can we infer that "{{hypothesis}}"? Yes, no, or
maybe?
```

Target Template:

```
{% if label !=-1 %}{{ answer_choices[label] }}{% endif %}
```

Answer Choices Template:

```
Yes ||| No ||| Maybe
```

Prompt from Schick and Schütze (2021)
Input Template:

```
{{premise}} Based on the previous passage, is it true that
"{{hypothesis}}"? Yes, no, or maybe?
```

Target Template:

```
{% if label !=-1 %}{{ answer_choices[label] }}{% endif %}
```

Answer Choices Template:

```
Yes ||| No ||| Maybe
```

Prompt from Webson and Pavlick (2021)
Input Template:

```
{{premise}} Based on that information, is the claim: "{{hypothesis}}"
{{"true"}}, {{"false"}}, or {{"inconclusive"}}?
```

Target Template:

```
{% if label !=-1 %}{{ answer_choices[label] }}{% endif %}
```

Answer Choices Template:

```
True ||| False ||| Inconclusive
```

Input Template:

```
Given that {{premise}} Does it follow that {{hypothesis}} Yes, no, or
maybe?
```

Target Template:

```
{% if label !=-1 %}{{ answer_choices[label] }}{% endif %}
```

Answer Choices Template:

```
Yes ||| No ||| Maybe
```

Prompt from Webson and Pavlick (2021)
Input Template:

```
{{premise}} Are we justified in saying that "{{hypothesis}}"? Yes, no,
or maybe?
```

Target Template:

```
{% if label !=-1 %}{{ answer_choices[label] }}{% endif %}
```

Answer Choices Template:

```
Yes ||| No ||| Maybe
```

Prompt from Webson and Pavlick (2021)
Input Template:

```
Suppose it's true that {{premise}} Then, is "{{hypothesis}}"
{{"always"}}, {{"sometimes"}}, or {{"never"}} true?
```

Target Template:

```
{% if label !=-1 %}{{ answer_choices[label] }}{% endif %}
```

Answer Choices Template:

```
Always ||| Never ||| Sometimes
```

Prompt from Brown et al. (2020)
Input Template:

```
{{premise}}
Question: {{hypothesis}} True, False, or Neither?
```

Target Template:

```
{% if label !=-1 %}{{ answer_choices[label] }}{% endif %}
```

Answer Choices Template:

```
True ||| False ||| Neither
```

Prompt from Webson and Pavlick (2021)
Input Template:

```
{{premise}}

Keeping in mind the above text, consider: {{hypothesis}} Is this
{{"always"}}, {{"sometimes"}}, or {{"never"}} correct?
```

Target Template:

```
{% if label !=-1 %}{{ answer_choices[label] }}{% endif %}
```

Answer Choices Template:

```
Always ||| Never ||| Sometimes
```

Prompt from Webson and Pavlick (2021)
Input Template:

```
Given {{premise}} Is it guaranteed true that "{{hypothesis}}"? Yes, no,
or maybe?
```

Target Template:

```
{% if label !=-1 %}{{ answer_choices[label] }}{% endif %}
```

Answer Choices Template:

```
Yes ||| No ||| Maybe
```

Input Template:

```
Given that {{premise}} Therefore, it must be true that "{{hypothesis}}"?
Yes, no, or maybe?
```

Target Template:

```
{% if label !=-1 %}{{ answer_choices[label] }}{% endif %}
```

Answer Choices Template:

```
Yes ||| No ||| Maybe
```

Prompt from Webson and Pavlick (2021)
Input Template:

```
Assume it is true that {{premise}}

Therefore, "{{hypothesis}}" is {{"guaranteed"}}, {{"possible"}}, or
{{"impossible"}}?
```

Target Template:

```
{% if label !=-1 %}{{ answer_choices[label] }}{% endif %}
```

Answer Choices Template:

```
Guaranteed ||| Impossible ||| Possible
```

Input Template:

```
{{premise}}

Question: Does this imply that "{{hypothesis}}"? Yes, no, or maybe?
```

Target Template:

```
{% if label !=-1 %}{{answer_choices[label]}}{% endif %}
```

Answer Choices Template:

```
Yes ||| No ||| Maybe
```

Prompt from Williams et al. (2018)
Input Template:

```
{{premise}} Using only the above description and what you know about the
world, "{{hypothesis}}" is definitely correct, incorrect, or
inconclusive?
```

Target Template:

```
{% if label !=-1 %}{{ answer_choices[label] }}{% endif %}
```

Answer Choices Template:

```
Correct ||| Incorrect ||| Inconclusive
```

Prompt from Webson and Pavlick (2021)
Input Template:

```
Given {{premise}} Should we assume that "{{hypothesis}}" is true? Yes,
no, or maybe?
```

Target Template:

```
{% if label !=-1 %}{{ answer_choices[label] }}{% endif %}
```

Answer Choices Template:

```
Yes ||| No ||| Maybe
```

Prompt from Webson and Pavlick (2021)
Input Template:

```
Take the following as truth: {{premise}}
Then the following statement: "{{hypothesis}}" is {{"true"}},
{{"false"}}, or {{"inconclusive"}}?
```

Target Template:

```
{% if label !=-1 %}{{ answer_choices[label] }}{% endif %}
```

Answer Choices Template:

```
True ||| False ||| Inconclusive
```

### 1.3.2  SUPER_GLUE RTE

Dataset from Dagan et al. (2005). Used in evaluation.

#### Data Example

| Key | Value |
|-----|-------|
| hypothesis | Weapons of Mass Destruction Found in Iraq. |
| idx | 0 |
| label | 1 |
| premise | No Weapons of Mass Destruction Found in Iraq Yet. |

#### Prompts

Prompt from Williams et al. (2018)
Input Template:

```
{{premise}} Using only the above description and what you know about the
world, is "{{hypothesis}}" definitely correct? Yes or no?
```

Target Template:

```
{% if label != -1 %}{{ answer_choices[label] }}{% endif %}
```

Answer Choices Template:

```
Yes ||| No
```

Prompt from Webson and Pavlick (2021)
Input Template:

```
Given {{premise}} Is it guaranteed true that "{{hypothesis}}"? Yes or
no?
```

Target Template:

```
{% if label != -1 %}{{ answer_choices[label] }}{% endif %}
```

Answer Choices Template:

```
Yes ||| No
```

Prompt from Webson and Pavlick (2021)
Input Template:

```
Suppose {{premise}} Can we infer that "{{hypothesis}}"? Yes or no?
```

Target Template:

```
{% if label != -1 %}{{ answer_choices[label] }}{% endif %}
```

Answer Choices Template:

```
Yes ||| No
```

Prompt from Brown et al. (2020)
Input Template:

```
{{premise}}
Question: {{hypothesis}} True or False?
```

Target Template:

```
{% if label != -1 %}{{ answer_choices[label] }}{% endif %}
```

Answer Choices Template:

```
True ||| False
```

Input Template:

```
{{premise}}

Question: Does this imply that "{{hypothesis}}"? Yes or no?
```

Target Template:

```
{% if label != -1 %}{{answer_choices[label]}}{% endif %}
```

Answer Choices Template:

```
Yes ||| No
```

Prompt from Webson and Pavlick (2021)
Input Template:

```
Given {{premise}} Should we assume that "{{hypothesis}}" is true? Yes or
no?
```

Target Template:

```
{% if label != -1 %}{{ answer_choices[label] }}{% endif %}
```

Answer Choices Template:

```
Yes ||| No
```

Input Template:

```
Given that {{premise}} Does it follow that {{hypothesis}} Yes or no?
```

Target Template:

```
{% if label != -1 %}{{ answer_choices[label] }}{% endif %}
```

Answer Choices Template:

```
Yes ||| No
```

Prompt from Schick and Schütze (2021)
Input Template:

```
{{premise}} Based on the previous passage, is it true that
"{{hypothesis}}"? Yes or no?
```

Target Template:

```
{% if label != -1 %}{{ answer_choices[label] }}{% endif %}
```

Answer Choices Template:

```
Yes ||| No
```

Prompt from Webson and Pavlick (2021)
Input Template:

```
{{premise}} Are we justified in saying that "{{hypothesis}}"? Yes or no?
```

Target Template:

```
{% if label != -1 %}{{ answer_choices[label] }}{% endif %}
```

Answer Choices Template:

```
Yes ||| No
```

Input Template:

```
Given that {{premise}} Therefore, it must be true that "{{hypothesis}}"?
Yes or no?
```

Target Template:

```
{% if label != -1 %}{{ answer_choices[label] }}{% endif %}
```

Answer Choices Template:

```
Yes ||| No
```

### 1.3.3  ANLI

Dataset from Nie et al. (2020). Used in evaluation.

**Data Example**

| Key | Value |
|-----|-------|
| hypothesis | The trolleybus system has over 2 urban routes |
| label | 0 |
| premise | The Parma trolleybus system (Italian: "Rete filovi... |
| reason | |
| uid | 0fd0abfb-659e-4453-b196-c3a64d2d8267 |

**Prompts**

Prompt from Williams et al. (2018)
Input Template:

```
{{premise}} Using only the above description and what you know about the
world, "{{hypothesis}}" is definitely correct, incorrect, or
inconclusive?
```

Target Template:

```
{{ answer_choices[label] }}
```

Answer Choices Template:

```
Correct ||| Inconclusive ||| Incorrect
```

Prompt from Webson and Pavlick (2021)
Input Template:

```
Given {{premise}} Should we assume that "{{hypothesis}}" is true? Yes,
no, or maybe?
```

Target Template:

```
{{ answer_choices[label] }}
```

Answer Choices Template:

```
Yes ||| Maybe ||| No
```

Input Template:

```
Given that {{premise}} Does it follow that {{hypothesis}} Yes, no, or
maybe?
```

Target Template:

```
{{ answer_choices[label] }}
```

Answer Choices Template:

```
Yes ||| Maybe ||| No
```

Prompt from Brown et al. (2020)
Input Template:

```
{{premise}}
Question: {{hypothesis}} True, False, or Neither?
```

Target Template:

```
{{ answer_choices[label] }}
```

Answer Choices Template:

```
True ||| Neither ||| False
```

Prompt from Schick and Schütze (2021)
Input Template:

```
{{premise}} Based on the previous passage, is it true that
"{{hypothesis}}"? Yes, no, or maybe?
```

Target Template:

```
{{ answer_choices[label] }}
```

Answer Choices Template:

```
Yes ||| Maybe ||| No
```

Prompt from Webson and Pavlick (2021)
Input Template:

```
{{premise}} Are we justified in saying that "{{hypothesis}}"? Yes, no,
or maybe?
```

Target Template:

```
{{ answer_choices[label] }}
```

Answer Choices Template:

```
Yes ||| Maybe ||| No
```

Prompt from Webson and Pavlick (2021)
Input Template:

```
Take the following as truth: {{premise}}
Then the following statement: "{{hypothesis}}" is {{"true"}},
{{"false"}}, or {{"inconclusive"}}?
```

Target Template:

```
{{ answer_choices[label] }}
```

Answer Choices Template:

```
True ||| Inconclusive ||| False
```

Input Template:

```
Given that {{premise}} Therefore, it must be true that "{{hypothesis}}"?
Yes, no, or maybe?
```

Target Template:

```
{{ answer_choices[label] }}
```

Answer Choices Template:

```
Yes ||| Maybe ||| No
```

Prompt from Webson and Pavlick (2021)
Input Template:

```
Suppose {{premise}} Can we infer that "{{hypothesis}}"? Yes, no, or
maybe?
```

Target Template:

```
{{ answer_choices[label] }}
```

Answer Choices Template:

```
Yes ||| Maybe ||| No
```

Prompt from Webson and Pavlick (2021)
Input Template:

```
Assume it is true that {{premise}}

Therefore, "{{hypothesis}}" is {{"guaranteed"}}, {{"possible"}}, or
{{"impossible"}}?
```

Target Template:

```
{{ answer_choices[label] }}
```

Answer Choices Template:

```
Guaranteed ||| Possible ||| Impossible
```

Prompt from Webson and Pavlick (2021)
Input Template:

```
Suppose it's true that {{premise}} Then, is "{{hypothesis}}"
{{"always"}}, {{"sometimes"}}, or {{"never"}} true?
```

Target Template:

```
{{ answer_choices[label] }}
```

Answer Choices Template:

```
Always ||| Sometimes ||| Never
```

Input Template:

```
{{premise}}

Question: Does this imply that "{{hypothesis}}"? Yes, no, or maybe?
```

Target Template:

```
{{answer_choices[label]}}
```

Answer Choices Template:

```
Yes ||| Maybe ||| No
```

Prompt from Webson and Pavlick (2021)
Input Template:

```
{{premise}}

Keeping in mind the above text, consider: {{hypothesis}} Is this
{{"always"}}, {{"sometimes"}}, or {{"never"}} correct?
```

Target Template:

```
{{ answer_choices[label] }}
```

Answer Choices Template:

```
Always ||| Sometimes ||| Never
```

Prompt from Webson and Pavlick (2021)
Input Template:

```
{{premise}} Based on that information, is the claim: "{{hypothesis}}"
{{"true"}}, {{"false"}}, or {{"inconclusive"}}?
```

Target Template:

```
{{ answer_choices[label] }}
```

Answer Choices Template:

```
True ||| Inconclusive ||| False
```

Prompt from Webson and Pavlick (2021)
Input Template:

```
Given {{premise}} Is it guaranteed true that "{{hypothesis}}"? Yes, no,
or maybe?
```

Target Template:

```
{{ answer_choices[label] }}
```

Answer Choices Template:

```
Yes ||| Maybe ||| No
```

## 1.4 PARAPHRASE

### 1.4.1 GLUE MRPC

Dataset from Dolan and Brockett (2005). Used in evaluation.

**Data Example**

| Key | Value |
| --- | --- |
| idx | 0 |
| label | 1 |
| sentence1 | Amrozi accused his brother , whom he called " the ... |
| sentence2 | Referring to him as only " the witness " , Amrozi ... |

## Prompts

Prompt not for the original task intended by the dataset authors
Input Template:

```
{% if label == 1 %}
Paraphrase the following sentence: {{sentence1}}
```

Target Template:

```
{{sentence2}}
{% endif %}
```

Input Template:

```
I want to know whether the following two sentences mean the same thing.
{{sentence1}}
{{sentence2}}
Do they?
```

Target Template:

```
{{ answer_choices[label] }}
```

Answer Choices Template:

```
no ||| yes
```

Input Template:

```
Does the sentence
{{sentence1}}
paraphrase (that is, mean the same thing as) this sentence?
{{sentence2}}
```

Target Template:

```
{{ answer_choices[label] }}
```

Answer Choices Template:

```
no ||| yes
```

Input Template:

```
Are the following two sentences "{{"equivalent"}}" or "{{"not
equivalent"}}"?
{{sentence1}}
{{sentence2}}
```

Target Template:

```
{{ answer_choices[label] }}
```

Answer Choices Template:

```
not equivalent ||| equivalent
```

Prompt not for the original task intended by the dataset authors
Input Template:

```
{% if label == 1 %}
Generate a sentence that means the same thing as this one: {{sentence1}}
```

Target Template:

```
{{sentence2}}
{% endif %}
```

Input Template:

```
Can I replace the sentence
{{sentence1}}
with the sentence
{{sentence2}}
and have it mean the same thing?
```

Target Template:

```
{{ answer_choices[label] }}
```

Answer Choices Template:

```
no ||| yes
```

Input Template:

```
Do the following two sentences mean the same thing?
{{sentence1}}
{{sentence2}}
```

Target Template:

```
{{ answer_choices[label] }}
```

Answer Choices Template:

```
no ||| yes
```

### 1.4.2  GLUE QQP

Dataset from Iyer et al. (2017). Used in evaluation.

**Data Example**

| Key | Value |
|---|---|
| idx | 0 |
| label | 0 |
| question1 | How is the life of a math student? Could you descr... |
| question2 | Which level of prepration is enough for the exam j... |

**Prompts**

Input Template:

```
I'm an administrator on the website Quora. There are two posts, one that
asks "{{question1}}" and another that asks "{{question2}}". I can merge
questions if they are asking the same thing. Can I merge these two
questions?
```

Target Template:

```
{{ answer_choices[label] }}
```

Answer Choices Template:

```
no ||| yes
```

Input Template:

```
{{question1}}
{{question2}}
Pick one: These questions are "{{"duplicates"}}" or "{{"not
duplicates"}}".
```

Target Template:

```
{{ answer_choices[label] }}
```

Answer Choices Template:

```
not duplicates ||| duplicates
```

Input Template:

```
Are the questions "{{question1}}" and "{{question2}}" asking the same
thing?
```

Target Template:

```
{{ answer_choices[label] }}
```

Answer Choices Template:

```
no ||| yes
```

Prompt not for the original task intended by the dataset authors
Input Template:

```
Can an answer to "{{question1}}" also be used to answer "{{question2}}"?
```

Target Template:

```
{{ answer_choices[label] }}
```

Answer Choices Template:

```
no ||| yes
```

Input Template:

```
Question 1: {{question1}}
Question 2: {{question2}}

Do these two questions convey the same meaning? Yes or no?
```

Target Template:

```
{{answer_choices[label]}}
```

Answer Choices Template:

```
No ||| Yes
```

Input Template:

```
I received the questions "{{question1}}" and "{{question2}}". Are they
duplicates?
```

Target Template:

```
{{ answer_choices[label] }}
```

Answer Choices Template:

```
no ||| yes
```

### 1.4.3 PAWS LABELED_FINAL

Dataset from Zhang et al. (2019). Used in training.

**Data Example**

| Key | Value |
| --- | --- |
| id | 1 |
| label | 0 |
| sentence1 | In Paris , in October 1560 , he secretly met the E... |
| sentence2 | In October 1560 , he secretly met with the English... |

**Prompts**

Input Template:

```
Determine if the following two sentences paraphrase each other or not.
Sent 1: {{sentence1}}
Sent 2: {{sentence2}}
```

Target Template:

```
{{answer_choices[label]}}
```

Answer Choices Template:

```
No ||| Yes
```

Input Template:

```
Sentence 1: {{sentence1}}
Sentence 2: {{sentence2}}
Question: Do Sentence 1 and Sentence 2 express the same meaning? Yes or
No?
```

Target Template:

```
{{answer_choices[label]}}
```

Answer Choices Template:

```
No ||| Yes
```

Input Template:

```
{{sentence1}}
Is that a paraphrase of the following sentence?
{{sentence2}}?
```

Target Template:

```
{{answer_choices[label]}}
```

Answer Choices Template:

```
No ||| Yes
```

Input Template:

```
Sentence 1: {{sentence1}}
Sentence 2: {{sentence2}}
Question: Can we rewrite Sentence 1 to Sentence 2?
```

Target Template:

```
{{answer_choices[label]}}
```

Answer Choices Template:

```
No ||| Yes
```

Input Template:

```
{{sentence1}}
Is that a paraphrase of the following sentence?
{{sentence2}}?
Yes or No.
```

Target Template:

```
{{answer_choices[label]}}
```

Answer Choices Template:

```
No ||| Yes
```

Input Template:

```
Sentence 1: {{sentence1}}
Sentence 2: {{sentence2}}
Question: Does Sentence 1 paraphrase Sentence 2? Yes or No?
```

Target Template:

```
{{answer_choices[label]}}
```

Answer Choices Template:

```
No ||| Yes
```

Prompt not for the original task intended by the dataset authors
Input Template:

```
{% if label == 1 %}
Paraphrase the sentence: {{sentence1}}
```

Target Template:

```
{{sentence2}}
{% endif %}
```

Input Template:

```
Sentence 1: {{sentence1}}
Sentence 2: {{sentence2}}
Question: Does Sentence 1 paraphrase Sentence 2?
```

Target Template:

```
{{answer_choices[label]}}
```

Answer Choices Template:

```
No ||| Yes
```

Input Template:

```
Sentence 1: {{sentence1}}
Sentence 2: {{sentence2}}
Question: Do Sentence 1 and Sentence 2 express the same meaning?
```

Target Template:

```
{{answer_choices[label]}}
```

Answer Choices Template:

```
No ||| Yes
```

Prompt from Brown et al. (2020)
Input Template:

```
{{sentence1}} Question: {{sentence2}} True or False?
```

Target Template:

```
{{answer_choices[label]}}
```

Answer Choices Template:

```
False ||| True
```

Input Template:

```
Sentence 1: {{sentence1}}
Sentence 2: {{sentence2}}
Question: Can we rewrite Sentence 1 to Sentence 2? Yes or No?
```

Target Template:

```
{{answer_choices[label]}}
```

Answer Choices Template:

```
No ||| Yes
```

Prompt from Brown et al. (2020)
Input Template:

```
{{sentence1}} Question: {{sentence2}} Paraphrase or not?
```

Target Template:

```
{{answer_choices[label]}}
```

Answer Choices Template:

```
No ||| Yes
```

## 1.5 QA CLOSED BOOK

### 1.5.1 AI2_ARC ARC-CHALLENGE

Dataset from Clark et al. (2018). Used in evaluation.

**Data Example**

| Key | Value |
|---|---|
| answerKey | A |
| choices | {'label': ['A', 'B', 'C', 'D'], 'text': ['dry palm... |
| id | Mercury_SC_415702 |
| question | George wants to warm his hands quickly by rubbing ... |

**Prompts**

Prompt not for the original task intended by the dataset authors
Input Template:

```
Pick and copy all the incorrect options for the following question:

{{question}}

Options:
- {{choices["text"] | join("\n- ")}}
```

Target Template:

```
{% for i in range(choices["label"]|length) %}
{% if i != choices["label"].index(answerKey) %}
- {{choices["text"][i]}}
{% endif %}
{% endfor %}
```

Input Template:

```
Here's a problem to solve: {{question}}

Among the 4 following options, which is the correct answer?
{% for letter, t in zip(answer_choices, choices.text) %}
- {{letter}}: {{t}}
 {% endfor %}
```

Target Template:

```
{{answerKey}}
```

Answer Choices Template:

```
A ||| B ||| C ||| D
```

Input Template:

```
{{question}}

Options:
- {{answer_choices | join("\n- ")}}
```

Target Template:

```
{{answer_choices[choices["label"].index(answerKey)]}}
```

Answer Choices Template:

```
{{choices.text | join("|||")}}
```

Input Template:

```
I am hesitating between 4 options to answer the following question,
which option should I choose?
Question: {{question}}
Possibilities:
- {{answer_choices | join("\n- ")}}
```

Target Template:

```
{{answer_choices[choices["label"].index(answerKey)]}}
```

Answer Choices Template:

```
{{choices.text | join("|||")}}
```

Input Template:

```
I gave my students this multiple choice question: {{question}}

Only one answer is correct among these 4 choices:
- {{answer_choices | join("\n- ")}}

Could you tell me which one is correct?
```

Target Template:

```
{{answer_choices[choices["label"].index(answerKey)]}}
```

Answer Choices Template:

```
{{choices.text | join("|||")}}
```

---

Input Template:

```
Pick the most correct option to answer the following question.

{{question}}

Options:
{% for letter, t in zip(answer_choices, choices.text) %}
- {{letter}}: {{t}}
{% endfor %}
```

Target Template:

```
{{answerKey}}
```

Answer Choices Template:

```
A ||| B ||| C ||| D
```

---

### 1.5.2 AI2_ARC ARC-EASY

Dataset from Clark et al. (2018). Used in evaluation.

**Data Example**

| Key | Value |
|-----------|-------------------------------------------------------|
| answerKey | B |
| choices | {'label': ['A', 'B', 'C', 'D'], 'text': ['a leg mu... |
| id | Mercury_7220990 |
| question | Which factor will most likely cause a person to de... |

**Prompts**

Input Template:

```
Pick the most correct option to answer the following question.

{{question}}

Options:
```

```
{% for letter, t in zip(answer_choices, choices.text) %}
- {{letter}}: {{t}}
{% endfor %}
```

Target Template:

```
{{answerKey}}
```

Answer Choices Template:

```
A ||| B ||| C ||| D
```

Input Template:

```
{{question}}

Options:
- {{answer_choices | join("\n- ")}}
```

Target Template:

```
{{answer_choices[choices["label"].index(answerKey)]}}
```

Answer Choices Template:

```
{{choices.text | join("|||")}}
```

Input Template:

```
I am hesitating between 4 options to answer the following question,
which option should I choose?
Question: {{question}}
Possibilities:
- {{answer_choices | join("\n- ")}}
```

Target Template:

```
{{answer_choices[choices["label"].index(answerKey)]}}
```

Answer Choices Template:

```
{{choices.text | join("|||")}}
```

Input Template:

```
I gave my students this multiple choice question: {{question}}

Only one answer is correct among these 4 choices:
- {{answer_choices | join("\n- ")}}

Could you tell me which one is correct?
```

Target Template:

```
{{answer_choices[choices["label"].index(answerKey)]}}
```

Answer Choices Template:

```
{{choices.text | join("|||")}}
```

Prompt not for the original task intended by the dataset authors
Input Template:

```
Pick and copy all the incorrect options for the following question:

{{question}}

Options:
- {{choices["text"] | join("\n- ")}}
```

Target Template:

```
{% for i in range(choices["label"]|length) %}
{% if i != choices["label"].index(answerKey) %}
- {{choices["text"][i]}}
{% endif %}
{% endfor %}
```

Input Template:

```
Here's a problem to solve: {{question}}

Among the 4 following options, which is the correct answer?
{% for letter, t in zip(answer_choices, choices.text) %}
- {{letter}}: {{t}}
 {% endfor %}
```

Target Template:

```
{{answerKey}}
```

Answer Choices Template:

```
A ||| B ||| C ||| D
```

### 1.5.3 KILT_TASKS HOTPOTQA

Dataset from **?**. Used in training.

**Data Example**

| Key | Value |
| --- | --- |
| id | 5a7a06935542990198eaf050 |
| input | Which magazine was started first Arthur's Magazine... |
| meta | {'left_context': '', 'mention': '', 'right_context... |
| output | [{'answer': "Arthur's Magazine", 'meta': {'score':... |

**Prompts**

Prompt not for the original task intended by the dataset authors
Input Template:

```
{% if output %}
Here's a complex question that requires someone to reason about the
input, can you answer it?
{{input}}
```

Target Template:

```
{{output | map(attribute="answer") | list | choice}}
{% endif %}
```

Prompt not for the original task intended by the dataset authors
Input Template:

```
{% if output %}
Combine facts and answer this: {{input}}
```

Target Template:

```
{{output | map(attribute="answer") | list | choice}}
{% endif %}
```

Prompt not for the original task intended by the dataset authors
Input Template:

```
{% if output %}
Formulate an answer to this elaborate question: {{input}}
```

Target Template:

```
{{output | map(attribute="answer") | list | choice}}
{% endif %}
```

Prompt not for the original task intended by the dataset authors
Input Template:

```
{% if output %}
FINAL EXAM

Question 1. {{input}}
```

Target Template:

```
{{output | map(attribute="answer") | list | choice}}
{% endif %}
```

Prompt not for the original task intended by the dataset authors
Input Template:

```
{% if output %}
{{input}}
```

Target Template:

```
{{output | map(attribute="answer") | list | choice}}
{% endif %}
```

### 1.5.4   TRIVIA_QA UNFILTERED

Dataset from Joshi et al. (2017). Used in evaluation.

**Data Example**

| Key | Value |
| --- | --- |
| question | Who was President when the first Peanuts cartoon w... |
| question_id | tc_0 |
| question_source | http://www.triviacountry.com/ |
| entity_pages | {'doc_source': ['TagMe'], 'filename': ['Peanuts.tx... |
| search_results | {'description': ['Peanuts 1950s. The first Peanuts... |
| answer | {'aliases': ['Presidency of Harry S. Truman', 'Har... |

**Prompts**

Prompt not for the original task intended by the dataset authors
Input Template:

```
{% if answer.aliases %}
    Guess a question that has the answer "{{answer.aliases|choice}}"
```

Target Template:

```
{{question}}
{% endif %}
```

Input Template:

```
The goal is to predict an English answer string for an input English
question.
Question : {{question}}
Answer :
```

Target Template:

```
{% if answer.aliases %}
{{answer.aliases|choice}}
{% endif %}
```

Input Template:

```
Answer the following question.
{{question}}
```

Target Template:

```
{% if answer.aliases %}
{{answer.aliases|choice}}
{% endif %}
```

Input Template:

```
I've always wondered: {{question}}
```

Target Template:

```
{% if answer.aliases %}
{{answer.aliases|choice}}
{% endif %}
```

Input Template:

```
Question : {{question}}
Answer :
```

Target Template:

```
{% if answer.aliases %}
{{answer.aliases|choice}}
{% endif %}
```

### 1.5.5 WEB_QUESTIONS

Dataset from Berant et al. (2013). Used in evaluation.

**Data Example**

| Key | Value |
| --- | --- |
| answers | ['Jazmyn Bieber', 'Jaxon Bieber'] |
| question | what is the name of justin bieber brother? |
| url | http://www.freebase.com/view/en/justin_bieber |

**Prompts**

Input Template:

```
Give me the correct facts to answer this: {{question}}
```

Target Template:

```
{{answers | choice}}
```

Input Template:

```
Give me a possible correct answer to the question "{{ question }}"
```

Target Template:

```
{{ answers | choice }}
```

Input Template:

```
What's the answer to that question: {{question}}
```

Target Template:

```
{{answers | choice}}
```

Input Template:

```
Short general knowledge question: {{question}}
```

Target Template:

```
{{answers | choice}}
```

Input Template:

```
{{ question|capitalize }}
```

Target Template:

```
{{ answers | choice }}
```

### 1.5.6   WIKI_QA

Dataset from Yi et al. (2015). Used in training.

**Data Example**

| Key | Value |
| --- | --- |
| answer | African immigration to the United States refers to... |
| document_title | African immigration to the United States |
| label | 0 |
| question | HOW AFRICAN AMERICANS WERE IMMIGRATED TO THE US |
| question_id | Q0 |

**Prompts**

Input Template:

```
Question: {{question}}?
Would "{{answer}}" be a reasonable answer?
```

Target Template:

```
{{ answer_choices[label] }}
```

Answer Choices Template:

```
No ||| Yes
```

Input Template:

```
I am verifying the answers generated by an automatic system to the
following question: {{question}}
Suggested answer: {{answer}}
Should I validate this answer?
```

Target Template:

```
{{answer_choices[label]}}
```

Answer Choices Template:

```
No ||| Yes
```

Prompt not for the original task intended by the dataset authors
Input Template:

```
{% if label == 1 %}
What is the question to: "{{answer}}"? The topic is {{document_title}}.
```

Target Template:

```
"{{question}}?"
{% endif %}
```

Prompt not for the original task intended by the dataset authors
Input Template:

```
{% if label == 1 %}
Determine the topic of the question-answer pair.
Question: "{{question}}?";  Answer: "{{answer}}"? Topic:
```

Target Template:

```
{{document_title}}
{% endif %}
```

Prompt not for the original task intended by the dataset authors
Input Template:

```
{% if label == 1 %}
Generate a question about the topic "{{document_title}}" whose answer
would be: {{answer}}.
```

Target Template:

```
{{question}}?
{% endif %}
```

Input Template:

```
Question: {{question}}
I found the following answer on Google: {{answer}}
Is that a correct answer? Yes or no.
```

**Target Template:**

```
{{answer_choices[label]}}
```

**Answer Choices Template:**

```
No ||| Yes
```

Prompt not for the original task intended by the dataset authors

**Input Template:**

```
{% if label == 1 %}
Determine the topic of the question.
Question: "{{question}}?"
Topic:
```

**Target Template:**

```
{{document_title}}
{% endif %}
```

**Input Template:**

```
The exercise is to decide whether the question accepts the proposed
suggestion as a correct answer. If yes, write "{{answer_choices[1]}}",
otherwise write "{{answer_choices[0]}}".
Question: {{question}}
Suggestion: {{answer}}
```

**Target Template:**

```
{{answer_choices[label]}}
```

**Answer Choices Template:**

```
False ||| True
```

**Input Template:**

```
This is a correct answer to the following question about
{{document_title}}. Yes or no?
Answer: {{answer}}
Question: {{question}}
```

**Target Template:**

```
{{answer_choices[label]}}
```

Answer Choices Template:

```
No ||| Yes
```

Prompt not for the original task intended by the dataset authors
Input Template:

```
{% if label == 1 %}
Determine the topic of the passage.
"{{answer}}"
Topic:
```

Target Template:

```
{{document_title}}
{% endif %}
```

Input Template:

```
{% if label == 1 %}
Answer this question: {{question}}?
```

Target Template:

```
{{answer}}
{% endif %}
```

## 1.6 QA EXTRACTIVE

### 1.6.1 ADVERSARIAL_QA DBIDAF

Dataset from Bartolo et al. (2020). Used in training.

**Data Example**

| Key | Value |
|---|---|
| id | 821607441c173838196c4d1500c2ab21a044e6b0 |
| title | Yale_University |
| context | Slack (2003) compares three groups that conducted ... |
| question | what year were the research groups compared |
| answers | {'text': ['2003'], 'answer_start': [7]} |
| metadata | {'split': 'train', 'model_in_the_loop': 'BiDAF'} |

**Prompts**

Input Template:

```
{% if metadata.split != "test" %}
Extract the answer to the question from the following context.
Question: {{question}}
Context: {{context}}
```

Target Template:

```
{{answers.text | choice}}
{% endif %}
```

Input Template:

```
{% if metadata.split != "test" %}
Given the following passage

"{{context}}",

answer the following question. Note that the answer is present within
the text.

Question: {{question}}
```

Target Template:

```
{{answers.text | choice}}
{% endif %}
```

Prompt not for the original task intended by the dataset authors
Input Template:

```
I want to test the ability of students to read a passage and answer
questions about it. Could you please come up with a good question for
the passage "{{context}}"?
```

Target Template:

```
{{question}}
```

Input Template:

```
{% if metadata.split != "test" %}
I know that the answer to the question "{{question}}" is in
"{{context}}". Can you tell me what it is?
```

Target Template:

```
{{answers.text | choice}}
{% endif %}
```

Input Template:

```
{% if metadata.split != "test" %}
Question: "{{question}}"

Context: "{{context}}"

Answer:
```

Target Template:

```
{{answers.text | choice}}
{% endif %}
```

### 1.6.2 ADVERSARIAL_QA DBERT

Dataset from Bartolo et al. (2020). Used in training.

**Data Example**

| Key | Value |
|---|---|
| id | dab017ed8a1c27c6afa2d8618abc3a477a4edffc |
| title | Empiricism |
| context | A generation later, the Irish Anglican bishop, Geo... |
| question | what concept is mentioned last? |
| answers | {'text': ['subjective idealism'], 'answer_start': ... |
| metadata | {'split': 'train', 'model_in_the_loop': 'BERT-Larg... |

**Prompts**

Prompt not for the original task intended by the dataset authors
Input Template:

```
I want to test the ability of students to read a passage and answer
questions about it. Could you please come up with a good question for
the passage "{{context}}"?
```

Target Template:

```
{{question}}
```

Input Template:

```
{% if metadata.split != "test" %}
I know that the answer to the question "{{question}}" is in
"{{context}}". Can you tell me what it is?
```

Target Template:

```
{{answers.text | choice}}
{% endif %}
```

Input Template:

```
{% if metadata.split != "test" %}
Question: "{{question}}"

Context: "{{context}}"

Answer:
```

Target Template:

```
{{answers.text | choice}}
{% endif %}
```

Input Template:

```
{% if metadata.split != "test" %}
Extract the answer to the question from the following context.
Question: {{question}}
Context: {{context}}
```

Target Template:

```
{{answers.text | choice}}
{% endif %}
```

Input Template:

```
{% if metadata.split != "test" %}
Given the following passage

"{{context}}",

answer the following question. Note that the answer is present within
the text.

Question: {{question}}
```

Target Template:

```
{{answers.text | choice}}
{% endif %}
```

### 1.6.3 ADVERSARIAL_QA DROBERTA

Dataset from Bartolo et al. (2020). Used in training.

**Data Example**

| Key | Value |
|-----|-------|
| id | 12cf36866b656dc4f254081fe6796ea1be2f6d43 |
| title | Napoleon |
| context | When he became First Consul and later Emperor, Nap... |
| question | What jewelry like accessories did he wear? |
| answers | {'text': ["Légion d'honneur star, medal and ribbon... |
| metadata | {'split': 'train', 'model_in_the_loop': 'RoBERTa-L... |

**Prompts**

Prompt not for the original task intended by the dataset authors
Input Template:

```
I want to test the ability of students to read a passage and answer
questions about it. Could you please come up with a good question for
the passage "{{context}}"?
```

Target Template:

```
{{question}}
```

Input Template:

```
{% if metadata.split != "test" %}
I know that the answer to the question "{{question}}" is in
"{{context}}". Can you tell me what it is?
```

Target Template:

```
{{answers.text | choice}}
{% endif %}
```

Input Template:

```
{% if metadata.split != "test" %}
Question: "{{question}}"

Context: "{{context}}"

Answer:
```

Target Template:

```
{{answers.text | choice}}
{% endif %}
```

Input Template:

```
{% if metadata.split != "test" %}
Extract the answer to the question from the following context.
Question: {{question}}
Context: {{context}}
```

Target Template:

```
{{answers.text | choice}}
{% endif %}
```

Input Template:

```
{% if metadata.split != "test" %}
Given the following passage

"{{context}}",

answer the following question. Note that the answer is present within
the text.

Question: {{question}}
```

Target Template:

```
{{answers.text | choice}}
{% endif %}
```

### 1.6.4 DUORC SELFRC

Dataset from Saha et al. (2018). Used in training.

**Data Example**

| Key | Value |
|---|---|
| answers | ['They arrived by train.'] |
| no_answer | False |
| plot | 200 years in the future, Mars has been colonized b... |
| plot_id | /m/03vyhn |
| question | How did the police arrive at the Mars mining camp? |
| question_id | b440de7d-9c3f-841c-eaec-a14bdff950d1 |
| title | Ghosts of Mars |

## Prompts

Prompt not for the original task intended by the dataset authors
Input Template:

```
{% if no_answer == false%}
Generate a question that has the following answer:
{{answers|choice}}
for the following movie plot:
{{plot}}
```

Target Template:

```
{{question}}
{% endif %}
```

Input Template:

```
I am a movie director and I just received the following movie plot.
Could you help me answer this question? If not, let me know by writing
"{{"Not answerable"}}".

Plot title: {{title}}
Movie plot: {{plot}}
My question: {{question}}
```

Target Template:

```
{% if no_answer %}
Not answerable
{% else %}
{{answers|choice}}
{% endif %}
```

Input Template:

```
Extract the answer to the following question from the movie plot. If the
question isn't answerable, please output "{{"Can't answer"}}".
Question: {{question}}
Title: {{title}}
Movie plot: {{plot}}
```

Target Template:

```
{% if no_answer %}
Can't answer
{% else %}
{{answers | choice }}
{% endif %}
```

Prompt not for the original task intended by the dataset authors
Input Template:

```
Generate a question about the following movie plot: {{ plot }}
```

Target Template:

```
{{ question }}
```

Input Template:

```
Please answer the following question about this movie plot. If it's
un-answerable, please output "{{"No answer"}}".

Question: {{question}}
Movie plot title: {{title}}
Movie plot: {{plot}}
```

Target Template:

```
{% if no_answer %}
No answer
{% else %}
{{answers | choice }}
{% endif %}
```

Prompt not for the original task intended by the dataset authors
Input Template:

```
{% if no_answer == false%}
Build a movie plot around this: {{ question }} {{answers|choice}}
```

Target Template:

```
{{ plot }}
{% endif %}
```

Input Template:

```
Question: {{question}}
If there is no answer, please output "{{"Insufficient information to
provide an answer."}}".
Movie title: {{title}}
Context: {{plot}}
```

Target Template:

```
{% if no_answer %}
Insufficient information to provide an answer.
{% else %}
{{answers|choice}}
{% endif %}
```

Prompt not for the original task intended by the dataset authors
Input Template:

```
Suggest a movie title for the following movie plot: {{plot}}
```

Target Template:

```
{{title}}
```

Input Template:

```
I am trying to decide whether it's worth it to invest in this film
proposal. Can you help me answer a few questions? If you can't, please
say "{{"No I can't"}}".

Question: {{question}}
Movie title: {{title}}
Movie plot: {{plot}}
```

Target Template:

```
{% if no_answer %}
No I can't
{% else %}
{{answers|choice}}
{% endif %}
```

### 1.6.5 DUORC PARAPHRASERC

Dataset from Saha et al. (2018). Used in training.

**Data Example**

| Key | Value |
| --- | --- |
| answers | ['second in command Sergeant  Jericho and prisoner... |
| no_answer | False |
| plot | Set in the second half of the 22nd century, Mars h... |
| plot_id | /m/03vyhn |
| question | who is there with Melanie Ballard? |
| question_id | 28ded42d-f6d5-aac6-cf6f-9e6e0820c5aa |
| title | Ghosts of Mars |

**Prompts**

Prompt not for the original task intended by the dataset authors
Input Template:

```
{% if no_answer == false%}
Build a movie plot around this: {{ question }} {{answers|choice}}
```

Target Template:

```
{{ plot }}
{% endif %}
```

Input Template:

```
I am trying to decide whether it's worth it to invest in this film
proposal. Can you help me answer a few questions? If you can't, please
say "{{"No I can't"}}".

Question: {{question}}
Movie title: {{title}}
Movie plot: {{plot}}
```

Target Template:

```
{% if no_answer %}
No I can't
{% else %}
{{answers|choice}}
{% endif %}
```

Input Template:

```
Question: {{question}}
If there is no answer, please output "{{"Insufficient information to
provide an answer."}}".
Movie title: {{title}}
Context: {{plot}}
```

Target Template:

```
{% if no_answer %}
Insufficient information to provide an answer.
{% else %}
{{answers|choice}}
{% endif %}
```

Input Template:

```
I am a movie director and I just received the following movie plot.
Could you help me answer this question? If not, let me know by writing
"{{"Not answerable"}}".

Plot title: {{title}}
Movie plot: {{plot}}
My question: {{question}}
```

Target Template:

```
{% if no_answer %}
Not answerable
{% else %}
{{answers|choice}}
{% endif %}
```

Prompt not for the original task intended by the dataset authors
Input Template:

```
Generate a question about the following movie plot: {{ plot }}
```

Target Template:

```
{{ question }}
```

Input Template:

```
Extract the answer to the following question from the movie plot. If the
question isn't answerable, please output "{{"Can't answer"}}".
Question: {{question}}
Title: {{title}}
Movie plot: {{plot}}
```

Target Template:

```
{% if no_answer %}
Can't answer
{% else %}
{{answers | choice }}
{% endif %}
```

Prompt not for the original task intended by the dataset authors
Input Template:

```
Suggest a movie title for the following movie plot: {{plot}}
```

Target Template:

```
{{title}}
```

Input Template:

```
Please answer the following question about this movie plot. If it's
un-answerable, please output "{{"No answer"}}".

Question: {{question}}
Movie plot title: {{title}}
Movie plot: {{plot}}
```

Target Template:

```
{% if no_answer %}
No answer
{% else %}
{{answers | choice }}
{% endif %}
```

Prompt not for the original task intended by the dataset authors
Input Template:

```
{% if no_answer == false%}
Generate a question that has the following answer:
{{answers|choice}}
for the following movie plot:
{{plot}}
```

Target Template:

```
{{question}}
{% endif %}
```

### 1.6.6  ROPES

Dataset from Lin et al. (2019). Used in training.

#### Data Example

| Key | Value |
|---|---|
| answers | {'text': ['cup B']} |
| background | Passive transport occurs when a substance passes t... |
| id | 1971664873 |
| question | Which cup has a higher concentration of sugar? |
| situation | A man put two cups, cup A and cup B, filled with e... |

#### Prompts

Input Template:

```
{% if answers.text %}
Please answer correctly the following question related to the paragraph
below.

{{ question }}

{{ situation }}

Hint: {{ background }}
```

Target Template:

```
{{ answers.text | choice }}
{% endif %}
```

Prompt not for the original task intended by the dataset authors
Input Template:

```
{% if answers.text %}
{{ situation }}

Given the paragraph above, please answer correctly the following
question:

{{ question }}
```

Target Template:

```
{{ answers.text | choice }}
{% endif %}
```

Input Template:

```
{% if answers.text %}
Background: {{ background }}

Paragraph: {{ situation }}

Given the paragraph above, please answer correctly the following
question: {{ question }}
```

Target Template:

```
{{ answers.text | choice }}
{% endif %}
```

Input Template:

```
{% if answers.text %}
Given the background: {{background}}

and the situation: {{situation}}

Answer the following question: {{question}}
```

Target Template:

```
{{ answers.text | choice }}
{% endif %}
```

Prompt not for the original task intended by the dataset authors
Input Template:

```
{% if answers.text %}
{{ situation }}

{{ question }}
```

**Target Template:**

```
{{ answers.text | choice }}
{% endif %}
```

**Input Template:**

```
{% if answers.text %}
{{ situation }}

{{ question }}

Hint: {{ background }}
```

**Target Template:**

```
{{ answers.text | choice}}
{% endif %}
```

**Input Template:**

```
{% if answers.text %}
{{ background }}

{{ situation }}

{{ question }}
```

**Target Template:**

```
{{ answers.text | choice }}
{% endif %}
```

**Input Template:**

```
{% if answers.text %}
I can use this background: {{background}}

Now, I have a new situation: {{situation}}

Answer this question please: {{question}}
```

**Target Template:**

```
{{ answers.text | choice }}
{% endif %}
```

Input Template:

```
{% if answers.text %}
You are given a new situation: {{situation}}

and a hint : {{background}}

Please answer this question : {{question}}
```

Target Template:

```
{{ answers.text | choice }}
{% endif %}
```

Input Template:

```
{% if answers.text %}
I have a new situation: {{situation}}

But I can use this background: {{background}}

What is an answer for this question: {{question}}
```

Target Template:

```
{{ answers.text | choice }}
{% endif %}
```

Input Template:

```
{% if answers.text %}
{{ situation }}

Given the paragraph above, please answer correctly the following
question:

{{ question }}

Hint: {{ background }}
```

Target Template:

```
{{ answers.text | choice }}
{% endif %}
```

Input Template:

```
{% if answers.text %}
I read this background article the other day: {{background}}

I am facing a new situation today: {{situation}}

Using the knowledge I acquired from the background article, how should I
answer correctly the following question regarding my new situation:
{{question}}
```

Target Template:

```
{{ answers.text | choice }}
{% endif %}
```

### 1.6.7 SQUAD_V2

Dataset from Rajpurkar et al. (2016). Used in evaluation.

**Data Example**

| Key | Value |
|---|---|
| id | 56be85543aeaaa14008c9063 |
| title | Beyoncé |
| context | Beyoncé Giselle Knowles-Carter (/bijnse/ bee-Y... |
| question | When did Beyonce start becoming popular? |
| answers | {'text': ['in the late 1990s'], 'answer_start': [2... |

**Prompts**

Input Template:

```
{% set seq = [
'Answer the question depending on the context.',
'What is the answer?',
] %}

{{ seq | choice }}
Context: {{context}};
Question: {{question}};
Answer:
```

Target Template:

```
{% if answers.text == [] %}
Answer not in context
{% else %}
{{answers.text[0]}}
{% endif %}
```

Prompt not for the original task intended by the dataset authors
Input Template:

```
{% if answers.text != [] %}
Determine the question that you might have asked to get back the
following answer for the given context
Context: {{context}};
Answer: {{answers.text[0]}};
Question:
```

Target Template:

```
{{question}}
{% endif %}
```

Prompt not for the original task intended by the dataset authors
Input Template:

```
{% set seq = [
'What is this about? ',
'What is the paragraph about? ',
'Get the topic from: ',
'From the passage,  get the topic',
'I want to know the topic. ',
'Topic from the passage: ',
'Topic from the paragraph: ',
] %}
{{ seq | choice }}
{{context}}
```

Target Template:

```
{{title | replace("_", " ")}}
```

Prompt not for the original task intended by the dataset authors
Input Template:

```
{% set seq = [
'This is about ',
'What is this about? ',
'The paragraph is about ',
'What is the paragraph about? ',
'Get the topic: ',
'From the passage, the topic is',
'I want to know the topic. ',
'Topic from the passage: ',
'Topic from the paragraph: ',
] %}
{{context}}
{{ seq | choice }}
```

Target Template:

```
{{title | replace("_", " ")}}
```

Prompt not for the original task intended by the dataset authors
Input Template:

```
{% if answers.text != [] %}
What is a question that would give the following answer?
Answer: {{answers.text[0]}};
Question:
```

Target Template:

```
{{question}}
{% endif %}
```

Input Template:

```
{% set seq = [
'Can you tell me ',
'Please tell me ',
'Tell me ',
'From the passage, ',
'I want to know ',
'I want to ask ',
'What is the answer to: ',
'Find the answer to: ',
'Answer: ',
'',
] %}
{{context}} {{ seq | choice }}{{question}}
```

Target Template:

```
{% if answers.text == [] %}
Answer not in context
{% else %}
{{answers.text[0]}}
{% endif %}
```

Input Template:

```
{% set seq = [
'Answer the question depending on the context.',
'What is the answer?',
] %}

{{ seq | choice }}
Context: {{context}};
Question: {{question}};
If you can't find the answer, please respond "unanswerable".
Answer:
```

Target Template:

```
{% if answers.text == [] %}
unanswerable
{% else %}
{{answers.text[0]}}
{% endif %}
```

**Prompt not for the original task intended by the dataset authors**
Input Template:

```
{% if answers.text != [] %}
{{question}}
```

Target Template:

```
{{answers.text[0]}}
{% endif %}
```

Input Template:

```
{% set seq = [
'Can you tell me ',
'Please tell me ',
'Tell me ',
'From the passage, ',
'I want to know ',
'I want to ask ',
'What is the answer to: ',
'Find the answer to: ',
'Answer: ',
'',
] %}
{{context}} {{ seq | choice }}{{question}} If you can't find the answer,
please respond "unanswerable".
```

Target Template:

```
{% if answers.text == [] %}
unanswerable
{% else %}
{{answers.text[0]}}
{% endif %}
```

**Prompt not for the original task intended by the dataset authors**
Input Template:

```
Context: {{context}};

Question: {{question}}

Is this question answerable?
```

Target Template:

```
{% if answers.text != [] %}
{{answer_choices[0]}}
{% else %}
{{answer_choices[1]}}
{% endif %}
```

Answer Choices Template:

```
yes ||| no
```

Prompt not for the original task intended by the dataset authors
Input Template:

```
{% set seq = [
'Determine the topic of the question-answer pair. ',
'Find the topic. ',
'What is the topic from this? ',
] %}
{% if answers.text != [] %}
{{ seq | choice }}
Question: {{question}};  Answer: {{answers.text[0]}}; Topic:
```

Target Template:

```
{{title}}
{% endif %}
```

Prompt not for the original task intended by the dataset authors
Input Template:

```
What is the following passage about?
{{context}}
```

Target Template:

```
{{title | replace("_", " ")}}
```

### 1.6.8   SUPER_GLUE RECORD

Dataset from Zhang et al. (2018). Used in evaluation.

**Data Example**

**Prompts**

Input Template:

| Key | Value |
| --- | --- |
| answers | ['Nuria'] |
| entities | ['Afghanistan', 'Badam Bagh', 'Mariam', 'Nuria'] |
| idx | {'passage': 0, 'query': 0} |
| passage | The harrowing stories of women and children locked... |
| query | The baby she gave birth to is her husbands and he ... |

```
{{ passage }}
{{ query }}
Which one is the "{{"@placeholder"}}"? {{ entities | join(", ") }}?
```

Target Template:

```
{% if ( answers | length ) > 0 %} {{ answers | choice }}
{% endif %}
```

Answer Choices Template:

```
{{ entities | join("|||") }}
```

Input Template:

```
The following document has been corrupted. Tell me what
"{{"@placeholder"}}" is referring to.

Document: {{ passage }}
{{ query }}
```

Target Template:

```
{% if ( answers | length ) > 0 %}{{ answers | choice }}
{% endif %}
```

Answer Choices Template:

```
{{ entities | join("|||") }}
```

Input Template:

```
Summary:

- {{ passage.split("@highlight")[1:] | join("\n- ") }}

Article:

{{ passage.split("@highlight")[0] }}
```

Target Template:

```
{% if ( answers | length ) > 0 %}{{ query | replace("@placeholder",
answers | choice) }} {% endif %}
```

Answer Choices Template:

```
{% for entity in entities[:-1] %} {{ query | replace("@placeholder",
entity) }} ||| {% endfor %} {{ query | replace("@placeholder",
entities[-1]) }}
```

Input Template:

```
Summary:

- {{ passage.split("@highlight")[1:] | join("\n- ") }}

Article:

{{ passage.split("@highlight")[0] }}

Now that you've read the article, please write a new sentence to add to
it.
```

Target Template:

```
{% if ( answers | length ) > 0 %}{{ query | replace("@placeholder",
answers | choice) }} {% endif %}
```

Answer Choices Template:

```
{% for entity in entities[:-1] %} {{ query | replace("@placeholder",
entity) }} ||| {% endfor %} {{ query | replace("@placeholder",
entities[-1]) }}
```

Input Template:

```
{{ passage }}
{{ query }}

You should decide what "{{"@placeholder"}}" is referring to. Choose
between:
- {{answer_choices | join("\n- ")}}
```

Target Template:

```
{% if ( answers | length ) > 0 %}{{ answers | choice }}
{% endif %}
```

Answer Choices Template:

```
{{ entities | join("|||") }}
```

Input Template:

```
{{ passage.split("@highlight")[0] }}

Summary:

- {{ passage.split("@highlight")[1:] | join("\n- ") }}
```

Target Template:

```
{% if ( answers | length ) > 0 %}- {{ query | replace("@placeholder",
answers | choice) }} {% endif %}
```

Answer Choices Template:

```
{% for entity in entities[:-1] %} - {{ query | replace("@placeholder",
entity) }} ||| {% endfor %} - {{ query | replace("@placeholder",
entities[-1]) }}
```

Prompt not for the original task intended by the dataset authors
Input Template:

```
Article:

{{ passage.split("@highlight")[0] }}

Highlights:

{{ passage.split("@highlight")[1:] | join("\n") }}
```

Target Template:

```
{% if ( answers | length ) > 0 %}{{ query | replace("@placeholder",
answers | choice) }} {% endif %}
```

Answer Choices Template:

```
{% for entity in entities[:-1] %} {{ query | replace("@placeholder",
entity) }} ||| {% endfor %} {{ query | replace("@placeholder",
entities[-1]) }}
```

Input Template:

```
{{ passage }}
{{ query }}
In the question above, the "{{"@placeholder"}}" stands for
```

Target Template:

```
{% if ( answers | length ) > 0 %}{{ answers | choice }}{% endif %}
```

Answer Choices Template:

```
{{ entities | join("|||") }}
```

Input Template:

```
After reading the article, write another sentence to add to it.
{{ passage | replace("@highlight", "\n- ") }}
```

Target Template:

```
{% if ( answers | length ) > 0 %}{{ query | replace("@placeholder",
answers | choice) }}{% endif %}
```

Answer Choices Template:

```
{% for entity in entities[:-1] %} {{ query | replace("@placeholder",
entity) }} ||| {% endfor %} {{ query | replace("@placeholder",
entities[-1]) }}
```

Input Template:

```
Please read the following news article and write another sentence to add
to it.

{{ passage | replace("@highlight", "\n- ") }}
```

Target Template:

```
{% if ( answers | length ) > 0 %}{{ query | replace("@placeholder",
answers | choice) }} {% endif %}
```

Answer Choices Template:

```
{% for entity in entities[:-1] %} {{ query | replace("@placeholder",
entity) }} ||| {% endfor %} {{ query | replace("@placeholder",
entities[-1]) }}
```

Input Template:

```
{{ passage }}
{{ query }}
What could the "{{"@placeholder"}}" be? {{ entities | join(", ") }}?
```

Target Template:

```
{% if ( answers | length ) > 0 %}{{ answers | choice }}{% endif %}
```

Answer Choices Template:

```
{{ entities | join("|||") }}
```

Input Template:

```
{{ passage }}
{{ query }}

I am trying to decide what "{{"@placeholder"}}" means in the previous
text.
Help by choosing an option between:
- {{ entities | join("\n- ") }}
```

Target Template:

```
{% if ( answers | length ) > 0 %}
{{ answers | choice }}
{% endif %}
```

Answer Choices Template:

```
{{entities | join("|||")}}
```

Input Template:

```
{{ passage }}
{{ query }}
Here, the placeholder refers to
```

Target Template:

```
{% if ( answers | length ) > 0 %}{{ answers | choice }}
{% endif %}
```

Answer Choices Template:

```
{{ entities | join("|||") }}
```

Input Template:

```
{{ passage.split("@highlight")[0] }}

Highlights:

- {{ passage.split("@highlight")[1:] | join("\n- ") }}

Please write an additional highlight.
```

Target Template:

```
{% if ( answers | length ) > 0 %}- {{ query | replace("@placeholder",
answers | choice) }} {% endif %}
```

Answer Choices Template:

```
{% for entity in entities[:-1] %} - {{ query | replace("@placeholder",
entity) }} ||| {% endfor %} - {{ query | replace("@placeholder",
entities[-1]) }}
```

Input Template:

```
Exercise: Extract from the text the correct entity that
"{{"@placeholder"}}" is referring to.

{{ passage }}
{{ query }}
```

Target Template:

```
{% if ( answers | length ) > 0 %}
{{ answers | choice }}
{% endif %}
```

Answer Choices Template:

```
{{entities | join("|||")}}
```

Input Template:

```
{{ passage }}
{{ query }}

Pick one option, "{{"@placeholder"}}" refers to:
- {{answer_choices | join("\n- ")}}
```

Target Template:

```
{% if ( answers | length ) > 0 %}
{{ answers | choice }}
{% endif %}
```

Answer Choices Template:

```
{{entities | join("|||")}}
```

Input Template:

```
{{ passage | replace("@highlight", "\n- ") }}
```

Target Template:

```
{% if ( answers | length ) > 0 %}- {{ query | replace("@placeholder",
answers | choice) }} {% endif %}
```

Answer Choices Template:

```
{% for entity in entities[:-1] %} - {{ query | replace("@placeholder",
entity) }} ||| {% endfor %} - {{ query | replace("@placeholder",
entities[-1]) }}
```

Prompt not for the original task intended by the dataset authors
Input Template:

```
Article:

{{ passage.split("@highlight")[0] }}

Highlights:

- {{ passage.split("@highlight")[1:] | join("\n- ") }}
```

Target Template:

```
{% if ( answers | length ) > 0 %}- {{ query | replace("@placeholder",
answers | choice) }} {% endif %}
```

Answer Choices Template:

```
{% for entity in entities[:-1] %} - {{ query | replace("@placeholder",
entity) }} ||| {% endfor %} - {{ query | replace("@placeholder",
entities[-1]) }}
```

Input Template:

```
{{ passage }}
{{ query }}
Can you figure out what does the "{{"@placeholder"}}" mean? It means
```

Target Template:

```
{% if ( answers | length ) > 0 %}{{ answers | choice }}{% endif %}
```

Answer Choices Template:

```
{{ entities | join("|||") }}
```

Input Template:

```
{{ passage | replace("@highlight", "\n") }}
```

Target Template:

```
{% if ( answers | length ) > 0 %}{{ query | replace("@placeholder",
answers | choice) }} {% endif %}
```

Answer Choices Template:

```
{% for entity in entities[:-1] %} {{ query | replace("@placeholder",
entity) }} ||| {% endfor %} {{ query | replace("@placeholder",
entities[-1]) }}
```

### 1.6.9 QUOREF

Dataset from Dasigi et al. (2019). Used in training.

**Data Example**

| Key | Value |
|-----|-------|
| answers | {'answer_start': [250], 'text': ['Catherine']} |
| context | The earthquake swarm was noted on October 12, 2007... |
| id | ba3f052c7a557909526b59713430403dd134e01d |
| question | What is the first name of the person who doubted i... |
| title | 2007{2008 Nazko earthquakes 1 |
| url | https://en.wikipedia.org/wiki/2007%E2%80%932008_Na... |

**Prompts**

Input Template:

```
The answer to the question: {{question}} is inside the article:
{{context}}, can you guess it ?
```

Target Template:

```
{{answers.text | choice}}
```

Input Template:

```
Given the following context:

{{context}}

answer the following question:
```

```
{{question}}
```

Target Template:

```
{{answers.text | choice}}
```

Input Template:

```
The following article contains an answer for the question: {{question}}
, can you please find it?

{{context}}
```

Target Template:

```
{{answers.text | choice}}
```

Input Template:

```
This article: {{context}} contains an answer for the question:
{{question}}, what is it ?
```

Target Template:

```
{{answers.text | choice}}
```

Input Template:

```
{{question}}

Answer the above question based on the context below:

{{context}}
```

Target Template:

```
{{answers.text | choice}}
```

Input Template:

```
What is the answer for the question: {{question}} from the following
article ?

{{context}}
```

Target Template:

```
{{answers.text | choice}}
```

Input Template:

```
I have a test where I am given the following article, what is an answer
for the question: {{question}} ?

{{context}}
```

Target Template:

```
{{answers.text | choice}}
```

Prompt not for the original task intended by the dataset authors
Input Template:

```
Given the below context:

{{context}}

Guess a valid title for it!
```

Target Template:

```
{{title}}
```

Input Template:

```
Found the following article online, use it to answer the question:
{{question}}

{{context}}
```

Target Template:

```
{{answers.text | choice}}
```

Input Template:

```
A friend asked me to answer this question: {{question}}, using the
article: {{context}}, what would be the answer ?
```

Target Template:

```
{{answers.text | choice}}
```

Input Template:

```
Read the following paragraph and extract the answer for the question:
{{question}}

{{context}}
```

Target Template:

```
{{answers.text | choice}}
```

## 1.7 QA MULTIPLE CHOICE

### 1.7.1 COS_E V1.11

Dataset from **?**. Used in training.

**Data Example**

| Key | Value |
|---|---|
| abstractive_explanation | webmath is designed to help you solve |
| answer | math problem |
| choices | ['park', 'coloring book', 'garden center', 'math p... |
| extractive_explanation | "there are 10 apples on an apple tree. three fall ... |
| id | 6b819727eb8a670df26a7ffad036c119 |
| question | "There are 10 apples on an apple tree.  Three fall... |

**Prompts**

Input Template:

```
{{ question }}
Choose the most suitable option to answer the above question.
Options:
- {{ answer_choices | join("\n- ") }}
```

Target Template:

```
{{ answer }}
```

Answer Choices Template:

```
{{ choices | join("|||") }}
```

Input Template:

```
{{ question }}
Choose the most suitable option to answer the above question.
Options
{% for k in range(choices | length) %}
```

```
{{'. '.join([answer_choices[k], choices[k]])}}
{% endfor %}
```

Target Template:

```
{{ answer_choices[choices.index(answer)] }}
```

Answer Choices Template:

```
A ||| B ||| C ||| D ||| E
```

Prompt not for the original task intended by the dataset authors
Input Template:

```
Question: {{question}}

Choices:
- {{ choices | join("\n- ") }}

The rationale to choose "{{answer}}" as the answer is that:
```

Target Template:

```
{{abstractive_explanation}}
```

Input Template:

```
{{ question }}
- {{ answer_choices | join("\n- ") }}

The best answer is
```

Target Template:

```
{{ answer }}
```

Answer Choices Template:

```
{{ choices | join("|||") }}
```

Prompt not for the original task intended by the dataset authors
Input Template:

```
Here's a question and a few possible answers:

Q: {{ question }}
Possible A: {{ choices | join(", ") }}

Why is "{{answer}}" an answer aligned with human common sense?
```

Target Template:

```
{{ abstractive_explanation }}
```

Input Template:

```
Pick the option in line with common sense to answer the question.
Question: {{ question }}
Options:
{% for k in range(choices | length) %}
{{'. '.join([answer_choices[k], choices[k]])}}
{% endfor %}
```

Target Template:

```
{{ answer_choices[choices.index(answer)] }}
```

Answer Choices Template:

```
A ||| B ||| C ||| D ||| E
```

Prompt not for the original task intended by the dataset authors
Input Template:

```
Question: {{ question }}
Options:
- {{ choices | join("\n- ") }}

Explain why a human would choose "{{answer}}" to answer the question
above:
```

Target Template:

```
{{ abstractive_explanation }}
```

Prompt not for the original task intended by the dataset authors
Input Template:

```
Question: {{ question }}
Options:
- {{ choices | join("\n- ") }}

The answer is "{{ answer }}" because
```

Target Template:

```
{{ abstractive_explanation }}
```

Input Template:

```
Pick the option in line with common sense to answer the question.
Questions: {{ question }}
Options:
- {{ answer_choices | join("\n- ") }}
```

Target Template:

```
{{ answer }}
```

Answer Choices Template:

```
{{ choices | join("|||") }}
```

Prompt not for the original task intended by the dataset authors
Input Template:

```
Here's a question: {{ question }}

Here are possible answers to this question:
- {{ choices | join("\n- ") }}

I believe the correct choice is "{{answer}}", here's why:
```

Target Template:

```
{{ abstractive_explanation }}
```

Input Template:

```
{{ question }}
{% for k in range(choices | length) %}
{{'. '.join([answer_choices[k], choices[k]])}}
{% endfor %}
The best answer is
```

Target Template:

```
{{ answer_choices[choices.index(answer)] }}
```

Answer Choices Template:

```
A ||| B ||| C ||| D ||| E
```

### 1.7.2 COSMOS_QA

Dataset from Huang et al. (2019). Used in training.

**Data Example**

| Key | Value |
|---|---|
| answer0 | None of the above choices . |
| answer1 | This person likes music and likes to see the show ... |
| answer2 | This person only likes Good Old War and Person L ,... |
| answer3 | Other Bands is not on tour and this person can not... |
| context | Good Old War and person L : I saw both of these ba... |
| id | 3Q9SPIIRWJKVQ8244310E8TUS6YWAC##34V1S5K3GTZMDUBNBI... |
| label | 1 |
| question | In the future , will this person go to see other b... |

**Prompts**

Prompt not for the original task intended by the dataset authors
Input Template:

```
Based on the context and the answer, generate a question.

Context: {{context}}

Answer:
{% if label == 0 %}
{{answer0}}
{% elif label == 1 %}
{{answer1}}
{% elif label == 2 %}
{{answer2}}
{% elif label == 3 %}
{{answer3}}
{% endif %}
```

Target Template:

```
{{question}}
```

Input Template:

```
Read the following context and choose the best option to answer the
question.
Context: {{ context }}
Question: {{ question }}
Options:
- {{ answer_choices | join("\n - ") }}
```

Target Template:

```
{{ answer_choices[label] }}
```

Answer Choices Template:

```
{{answer0}} ||| {{answer1}} ||| {{answer2}} ||| {{answer3}}
```

Input Template:

```
Read the following context and answer the question.
Context: {{ context }}
Question: {{ question }}
Answer:
```

Target Template:

```
{{ answer_choices[label] }}
```

Answer Choices Template:

```
{{answer0}} ||| {{answer1}} ||| {{answer2}} ||| {{answer3}}
```

Input Template:

```
Read the following context and choose the best option to answer the
question.
Context: {{ context }}
Question: {{ question }}
Options:
A. {{ answer0 }}
B. {{ answer1 }}
C. {{ answer2 }}
D. {{ answer3 }}
```

Target Template:

```
{{ answer_choices[label] }}
```

Answer Choices Template:

```
A ||| B ||| C ||| D
```

Input Template:

```
{{ context }}
According to the above context, choose the best option to answer the
following question.
Question: {{ question }}
Options:
- {{answer_choices | join("\n - ")}}
```

Target Template:

```
{{answer_choices[label]}}
```

Answer Choices Template:

```
{{answer0}} ||| {{answer1}} ||| {{answer2}} ||| {{answer3}}
```

Input Template:

```
{{ context }}
{{ question }}
A. {{ answer0 }}
B. {{ answer1 }}
C. {{ answer2 }}
D. {{ answer3 }}
```

Target Template:

```
{{ answer_choices[label] }}
```

Answer Choices Template:

```
A ||| B ||| C ||| D
```

Prompt not for the original task intended by the dataset authors
Input Template:

```
{{ context }}
Question: {{ question }}
The answer to the above question:
```

Target Template:

```
{{ answer_choices[label] }}
```

Answer Choices Template:

```
{{answer0}} ||| {{answer1}} ||| {{answer2}} ||| {{answer3}}
```

Input Template:

```
{{ context }}
{{ question }}
- {{ answer_choices | join("\n - ") }}
```

Target Template:

```
{{ answer_choices[label] }}
```

Answer Choices Template:

```
{{answer0}} ||| {{answer1}} ||| {{answer2}} ||| {{answer3}}
```

Input Template:

```
{{ context }}
According to the above context, choose the best option to answer the
following question.
Question: {{ question }}
Options:
A. {{ answer0 }}
B. {{ answer1 }}
C. {{ answer2 }}
D. {{ answer3 }}
```

Target Template:

```
{{ answer_choices[label] }}
```

Answer Choices Template:

```
A ||| B ||| C ||| D
```

Input Template:

```
{{ context }}
{{ question }}
Pick the best answer from the following options:
A. {{ answer0 }}
B. {{ answer1 }}
C. {{ answer2 }}
D. {{ answer3 }}
```

Target Template:

```
{{ answer_choices[label] }}
```

Answer Choices Template:

```
A ||| B ||| C ||| D
```

Input Template:

```
{{ context }}
According to the above context, answer the following question.
{{ question }}
```

Target Template:

```
{{answer_choices[label]}}
```

Answer Choices Template:

```
{{answer0}} ||| {{answer1}} ||| {{answer2}} ||| {{answer3}}
```

Input Template:

```
{{ context }}
{{ question }}
Pick the best answer from the following options:
- {{ answer_choices | join("\n - ") }}
```

Target Template:

```
{{ answer_choices[label] }}
```

Answer Choices Template:

```
{{answer0}} ||| {{answer1}} ||| {{answer2}} ||| {{answer3}}
```

Prompt not for the original task intended by the dataset authors
Input Template:

```
{{question}}
```

Target Template:

```
{{ answer_choices[label] }}
```

Answer Choices Template:

```
{{answer0}} ||| {{answer1}} ||| {{answer2}} ||| {{answer3}}
```

### 1.7.3  DREAM

Dataset from Sun et al. (2019). Used in training.

**Data Example**

**Prompts**

Prompt not for the original task intended by the dataset authors
Input Template:

| Key | Value |
| --- | --- |
| answer | Continue her dancing class. |
| choice | ['Consult her dancing teacher.', 'Take a more inte... |
| dialogue | ['M: I am considering dropping my dancing class. I... |
| dialogue_id | 5-510 |
| id | 0 |
| question | What does the man suggest the woman do? |

```
Read the below conversation.

{{dialogue[:-1] | join("\n\n")}}

What would the listener say?
```

Target Template:

```
{{dialogue[-1]}}
```

Prompt not for the original task intended by the dataset authors
Input Template:

```
Given the question "{{question}}" and the answer "{{answer}}", write a
conversation that might have happened.
```

Target Template:

```
{{dialogue | join("\n\n")}}
```

Prompt not for the original task intended by the dataset authors
Input Template:

```
{{dialogue[1:] | join("\n\n")}}

What was said before this conversation?
```

Target Template:

```
{{dialogue[0]}}
```

Input Template:

```
Dialogue:

{{dialogue | join("\n\n")}}

Question: {{question}}

- {{answer_choices[0]}}
```

```
- {{answer_choices[1]}}

- {{answer_choices[2]}}
```

Target Template:

```
{{answer}}
```

Answer Choices Template:

```
{{choice | join("|||")}}
```

Input Template:

```
Read the following conversation and answer the question.

{{dialogue | join("\n\n")}}

Question: {{question}}

- {{answer_choices[0]}}

- {{answer_choices[1]}}

- {{answer_choices[2]}}
```

Target Template:

```
{{answer}}
```

Answer Choices Template:

```
{{choice | join("|||")}}
```

### 1.7.4 OPENBOOKQA MAIN

Dataset from Mihaylov et al. (2018). Used in evaluation.

**Data Example**

| Key | Value |
|---|---|
| answerKey | D |
| choices | {'label': ['puppies learning new tricks', 'childre... |
| id | 7-980 |
| question_stem | The sun is responsible for |

**Prompts**

Input Template:

```
{{question_stem}}

Choose an answer from this list:
- {{ answer_choices | join("\n- ") }}
```

Target Template:

```
{{answer_choices[{"A":0,"B":1,"C":2,"D":3}[answerKey]]}}
```

Answer Choices Template:

```
{{choices.text | join("|||")}}
```

Input Template:

```
{{question_stem}}

Which is the correct answer?
- {{ answer_choices | join("\n- ") }}
```

Target Template:

```
{{answer_choices[{"A":0,"B":1,"C":2,"D":3}[answerKey]]}}
```

Answer Choices Template:

```
{{choices.text | join("|||")}}
```

Input Template:

```
{{question_stem}}
{% for k in range(choices["text"] | length) %}
{{' -> '.join([["A", "B", "C", "D"][k], choices["text"][k]])}}
{% endfor %}
Is the right answer {{"A, B, C or D"}} ?
```

Target Template:

```
{{answerKey}}
```

Answer Choices Template:

```
A ||| B ||| C ||| D
```

Input Template:

```
{{question_stem}}

Choices:
- {{ answer_choices | join("\n- ") }}
```

Target Template:

```
{{answer_choices[{"A":0,"B":1,"C":2,"D":3}[answerKey]]}}
```

Answer Choices Template:

```
{{choices.text | join("|||")}}
```

Input Template:

```
{{question_stem}}
- {{ answer_choices | join("\n- ") }}
```

Target Template:

```
{{answer_choices[{"A":0,"B":1,"C":2,"D":3}[answerKey]]}}
```

Answer Choices Template:

```
{{choices.text | join("|||")}}
```

Input Template:

```
{{question_stem}}
- {{ answer_choices | join("\n- ") }}

Which is the correct answer?
```

Target Template:

```
{{answer_choices[{"A":0,"B":1,"C":2,"D":3}[answerKey]]}}
```

Answer Choices Template:

```
{{choices.text | join("|||")}}
```

Input Template:

```
{{question_stem}}

Pick the right answer from the list:
- {{ answer_choices | join("\n- ") }}
```

Target Template:

```
{{answer_choices[{"A":0,"B":1,"C":2,"D":3}[answerKey]]}}
```

Answer Choices Template:

```
{{choices.text | join("|||")}}
```

### 1.7.5  QASC

Dataset from Khot et al. (2020). Used in training.

**Data Example**

| Key | Value |
|---|---|
| answerKey | F |
| choices | {'label': ['A', 'B', 'C', 'D', 'E', 'F', 'G', 'H']... |
| combinedfact | Beads of water can be formed by clouds. |
| fact1 | beads of water are formed by water vapor condensin... |
| fact2 | Clouds are made of water vapor. |
| formatted_question | What type of water formation is formed by clouds? ... |
| id | 3E7TUJ2EGCLQNOV1WEAJ2NN9ROPD9K |
| question | What type of water formation is formed by clouds? |

**Prompts**

Prompt not for the original task intended by the dataset authors
Input Template:

```
If I tell you that {{combinedfact[0]|capitalize}}{{
combinedfact[1:]|trim('.') }}, and ask you the question "{{
question[0]|lower }}{{ question[1:] }}", is the correct answer "{{
choices.text[0][0]|lower}}{{ choices.text[0][1:]|trim('.') }}"?
```

Target Template:

```
{% if answerKey == choices.label[0] %} Yes {% else %} No {% endif %}
```

Answer Choices Template:

```
Yes ||| No
```

Input Template:

```
{{ fact1[0]|capitalize }}{{ fact1[1:]|trim|trim('.') }}, and
{{fact2[0]|lower }}{{ fact2[1:]|trim|trim('.') }}. Given these facts, {{
question[0]|lower }}{{question[1:]|trim('?') }} among the following
options:
- {{answer_choices | join("\n - ") }}
```

Target Template:

```
{% for choice in choices.label %} {% if choice == answerKey %}{{
answer_choices[loop.index - 1] }}{% endif %}{% endfor %}
```

Answer Choices Template:

```
{{choices.text | join("|||")}}
```

Input Template:

```
Fact 1: {{ fact1[0]|capitalize }}{{ fact1[1:]|trim|trim('.') }}.

Fact 2: {{fact2[0]|capitalize }}{{ fact2[1:]|trim|trim('.') }}.

Given the two facts above, {{ question[0]|lower
}}{{question[1:]|trim('?') }}?
```

Target Template:

```
{% for choice in choices.label %} {% if choice == answerKey %}{{
answer_choices[loop.index - 1] }}{% endif %}{% endfor %}
```

Answer Choices Template:

```
{{choices.text | join("|||")}}
```

Input Template:

```
You are presented with the question "{{ question }}" and the following
answer choices:
- {{answer_choices | join("\n - ") }}

Now knowing that {{ fact1[0]|lower }}{{ fact1[1:]|trim|trim('.') }} and
{{fact2[0]|lower }}{{ fact2[1:]|trim|trim('.') }}, choose the best
answer.
```

Target Template:

```
{% for choice in choices.label %} {% if choice == answerKey %}{{
answer_choices[loop.index - 1] }}{% endif %}{% endfor %}
```

Answer Choices Template:

```
{{choices.text | join("|||")}}
```

Input Template:

```
You are presented with the quiz "{{ question }}"

But you don't know the answer, so you turn to your teacher to ask for
hints. He says that "{{ fact1[0]|lower }}{{ fact1[1:]|trim|trim('.') }}"
and "{{fact2[0]|lower }}{{ fact2[1:]|trim|trim('.') }}".

So, what's the best answer to the question?
```

Target Template:

```
{% for choice in choices.label %} {% if choice == answerKey %}{{
answer_choices[loop.index - 1] }}{% endif %}{% endfor %}
```

Answer Choices Template:

```
{{choices.text | join("|||")}}
```

Prompt not for the original task intended by the dataset authors
Input Template:

```
If {{ combinedfact[0]|lower }}{{ combinedfact[1:]|trim|trim('.') }},
then {{ question[0]|lower }}{{question[1:]|trim|trim('?') }}?

Answer choices:
- {{answer_choices | join("\n - ") }}
```

Target Template:

```
{% for choice in choices.label %} {% if choice == answerKey %}{{
answer_choices[loop.index - 1] }}{% endif %}{% endfor %}
```

Answer Choices Template:

```
{{choices.text | join("|||")}}
```

Prompt not for the original task intended by the dataset authors
Input Template:

```
Do you think the right answer to the question "{{ question[0]|lower }}{{
question[1:] }}" is "{{ choices.text[1][0]|lower}}{{
choices.text[1][1:]|trim('.') }}", given that
 {{combinedfact[0]|lower}}{{ combinedfact[1:]|trim('.') }}?
```

Target Template:

```
{% if answerKey == choices.label[0] %} Yes {% else %} No {% endif %}
```

Answer Choices Template:

```
Yes ||| No
```

Input Template:

```
Fact 1: {{ fact1[0]|capitalize }}{{ fact1[1:]|trim|trim('.') }}.

Fact 2: {{fact2[0]|capitalize }}{{ fact2[1:]|trim|trim('.') }}.

Given the two facts above, answer the question "{{ question }}" with the
following options:
- {{answer_choices | join("\n - ") }}
```

Target Template:

```
{% for choice in choices.label %} {% if choice == answerKey %}{{
answer_choices[loop.index - 1] }}{% endif %}{% endfor %}
```

Answer Choices Template:

```
{{choices.text | join("|||")}}
```

### 1.7.6   QUAIL

Dataset from Rogers et al. (2020). Used in training.

**Data Example**

| Key | Value |
|---|---|
| answers | ['not enough information', 'to visit family', 'par... |
| context | That fall came and I went back to Michigan and the... |
| context_id | f001 |
| correct_answer_id | 3 |
| domain | fiction |
| id | f001_0 |
| metadata | {'author': 'Joseph Devon', 'title': 'Black Eyed Su... |
| question | Why was this character sent away after each school... |
| question_id | 0 |
| question_type | Causality |

**Prompts**

Input Template:

```
{{ context }}
Question: {{ question }}
Options:
{% for k in range(answers | length) %}
{{'. '.join([answer_choices[k], answers[k]])}}
{% endfor %}
===
The correct answer is
```

Target Template:

```
{{ answer_choices[correct_answer_id] }}
```

Answer Choices Template:

```
A ||| B ||| C ||| D
```

Input Template:

```
{{ context }}
Question: {{ question }}
Options:
- {{ answer_choices | join(" \n - ") }}
===
The correct answer is
```

Target Template:

```
{{ answer_choices[correct_answer_id] }}
```

Answer Choices Template:

```
{{answers | join("|||")}}
```

Input Template:

```
Read the following context and choose the correct option to answer the
question.
Context: {{ context }}
Question: {{ question }}
Options:
{% for k in range(answers | length) %}
{{'. '.join([answer_choices[k], answers[k]])}}
{% endfor %}
```

Target Template:

```
{{ answer_choices[correct_answer_id] }}
```

Answer Choices Template:

```
A ||| B ||| C ||| D
```

Input Template:

```
{{ context }}
{{ question }}
Pick the correct answer from the following options:
- {{ answer_choices | join("\n- ") }}
```

Target Template:

```
{{ answer_choices[correct_answer_id] }}
```

Answer Choices Template:

```
{{answers | join("|||")}}
```

Prompt not for the original task intended by the dataset authors
Input Template:

```
{{ context }}
Question: {{ question }}
===
The answer to the above question is
```

Target Template:

```
{{ answer_choices[correct_answer_id] }}
```

Answer Choices Template:

```
{{answers | join("|||")}}
```

Prompt not for the original task intended by the dataset authors
Input Template:

```
{{ context }}
According to the above context, answer the following question.
{{ question }}
```

Target Template:

```
{{ answer_choices[correct_answer_id] }}
```

Answer Choices Template:

```
{{answers | join("|||")}}
```

Input Template:

```
{{ context }}
{{ question }}
Pick the correct answer from the following options:
{% for k in range(answers | length) %}
{{'. '.join([answer_choices[k], answers[k]])}}
{% endfor %}
```

Target Template:

```
{{ answer_choices[correct_answer_id] }}
```

Answer Choices Template:

```
A ||| B ||| C ||| D
```

Input Template:

```
{{ context }}
{{ question }}
{% for k in range(answers | length) %}
{{'. '.join([answer_choices[k], answers[k]])}}
{% endfor %}
```

Target Template:

```
{{ answer_choices[correct_answer_id] }}
```

Answer Choices Template:

```
A ||| B ||| C ||| D
```

Input Template:

```
{{ context }}
According to the above context, choose the correct option to answer the
following question.
Question: {{ question }}
Options:
{% for k in range(answers | length) %}
{{'. '.join([answer_choices[k], answers[k]])}}
{% endfor %}
```

Target Template:

```
{{ answer_choices[correct_answer_id] }}
```

Answer Choices Template:

```
A ||| B ||| C ||| D
```

**Prompt not for the original task intended by the dataset authors**
Input Template:

```
Read the following context and answer the question.
Context: {{ context }}
Question: {{ question }}
Answer:
```

Target Template:

```
{{ answer_choices[correct_answer_id] }}
```

Answer Choices Template:

```
{{answers | join("|||")}}
```

Input Template:

```
{{ context }}
{{ question }}
- {{ answer_choices | join("\n- ") }}
```

Target Template:

```
{{ answer_choices[correct_answer_id] }}
```

Answer Choices Template:

```
{{answers | join("|||")}}
```

Input Template:

```
{{ context }}
According to the above context, choose the correct option to answer the
following question.
Question: {{ question }}
Options:
- {{ answer_choices | join("\n- ") }}
```

Target Template:

```
{{ answer_choices[correct_answer_id] }}
```

Answer Choices Template:

```
{{answers | join("|||")}}
```

Input Template:

```
Read the following context and choose the correct option to answer the
question.
Context: {{ context }}
Question: {{ question }}
Options:
- {{ answer_choices | join("\n- ") }}
```

Target Template:

```
{{ answer_choices[correct_answer_id] }}
```

Answer Choices Template:

```
{{answers | join("|||")}}
```

### 1.7.7  QUAREL

Dataset from Tafjord et al. (2018). Used in training.

#### Data Example

| Key | Value |
| --- | --- |
| id | QuaRel_V1_Fr_0223 |
| answer_index | 1 |
| logical_forms | ['(infer (speed higher world1) (smoothness higher ... |
| logical_form_pretty | qrel(speed, higher, world1) -> qrel(smoothness, hi... |
| world_literals | {'world1': ['ice'], 'world2': ['snow']} |
| question | Mike was snowboarding on the snow and hit a piece ... |

#### Prompts

Prompt not for the original task intended by the dataset authors
Input Template:

```
Question: {{question}}

Do not use {{"A"}} and {{"B"}} to answer the question but instead,
choose between "{{answer_choices[0]}}" and  "{{answer_choices[1]}}".
```

Target Template:

```
{{answer_choices[answer_index]}}
```

Answer Choices Template:

```
{{world_literals.world1[0]}} ||| {{world_literals.world2[0]}}
```

Prompt not for the original task intended by the dataset authors
Input Template:

```
Here's a logic test: {{question}}

Choose the answer between "{{answer_choices[0]}}" and
"{{answer_choices[1]}}".
```

Target Template:

```
{{answer_choices[answer_index]}}
```

Answer Choices Template:

```
{{world_literals.world1[0]}} ||| {{world_literals.world2[0]}}
```

Prompt not for the original task intended by the dataset authors
Input Template:

```
Here's a short story: {{question}}.

What is the most sensical answer between "{{answer_choices[0]}}" and
"{{answer_choices[1]}}"?
```

Target Template:

```
{{answer_choices[answer_index]}}
```

Answer Choices Template:

```
{{world_literals.world1[0]}} ||| {{world_literals.world2[0]}}
```

Prompt not for the original task intended by the dataset authors
Input Template:

```
Choose between "{{answer_choices[0]}}" and  "{{answer_choices[1]}}".
Question: {{question}}
```

Target Template:

```
{{answer_choices[answer_index]}}
```

Answer Choices Template:

```
{{world_literals.world1[0]}} ||| {{world_literals.world2[0]}}
```

---

**Prompt not for the original task intended by the dataset authors**
Input Template:

```
I am testing my students' logic.
What is the answer they should choose between "{{answer_choices[0]}}"
and "{{answer_choices[1]}}"?
Logic test: {{question}}
```

Target Template:

```
{{answer_choices[answer_index]}}
```

Answer Choices Template:

```
{{world_literals.world1[0]}} ||| {{world_literals.world2[0]}}
```

---

### 1.7.8   QUARTZ

Dataset from Tafjord et al. ("2019"). Used in training.

**Data Example**

| Key | Value |
|---|---|
| answerKey | A |
| choices | {'label': ['A', 'B'], 'text': ['scarce', 'plentifu... |
| id | QRQA-10385-4 |
| para | Many of the worlds people live with water scarcity... |
| para_anno | {'effect_prop': 'population growth', 'cause_dir_st... |
| para_id | QRSent-10385 |
| question | John's town used to have lots of water, back when ... |
| question_anno | {'more_effect_dir': 'several thousand', 'less_effe... |

**Prompts**

Input Template:

```
Use information from the paragraph to answer the question.

Question:

{% if '_____' in question %}
{{ question | trim(".?!") | replace("_____", answer_choices | join(" or
")) }}{{ "?" }}
```

```
{% else %}
{{ question | trim(".?!") }} {{ answer_choices | join(" or ") }}{{ "?"
}}
{% endif %}

Paragraph :

{{ para }}
```

Target Template:

```
{{answer_choices[choices.label.index(answerKey)]}}
```

Answer Choices Template:

```
{{choices.text | join("|||")}}
```

Input Template:

```
{{ para }}
{% if '_____' in question %}
{{ question | trim(".?!") | replace("_____", answer_choices | join(" or
")) }}{{ "?" }}
{% else %}
{{ question | trim(".?!")}} {{ answer_choices | join(" or ") }}{{ "?" }}
{% endif %}
```

Target Template:

```
{{answer_choices[choices.label.index(answerKey)]}}
```

Answer Choices Template:

```
{{choices.text | join("|||")}}
```

Input Template:

```
Use information from the paragraph to answer the question.

Paragraph :

{{ para }}

Question:

{% if '_____' in question %}
{{ question | trim(".?!") | replace("_____", answer_choices | join(" or
")) }}{{ "?" }}
{% else %}
{{ question | trim(".?!") }} {{ answer_choices | join(" or ") }}{{ "?"
}}
{% endif %}
```

Target Template:

```
{{answer_choices[choices.label.index(answerKey)]}}
```

Answer Choices Template:

```
{{choices.text | join("|||")}}
```

Input Template:

```
Answer the question based on the following text.

Question:

{% if '_____' in question %}
{{ question | trim(".?!") | replace("_____", answer_choices | join(" or
")) }}{{ "?" }}
{% else %}
{{ question | trim(".?!") }} {{ answer_choices | join(" or ") }}{{ "?"
}}
{% endif %}

Text:

{{ para }}
```

Target Template:

```
{{answer_choices[choices.label.index(answerKey)]}}
```

Answer Choices Template:

```
{{choices.text | join("|||")}}
```

Input Template:

```
Answer the question below:

{% if '_____' in question %}
{{ question | trim(".?!") | replace("_____", answer_choices | join(" or
")) }}{{ "?" }}
{% else %}
{{ question | trim(".?!") }} {{  answer_choices | join(" or ") }}{{ "?"
}}
{% endif %}

Assuming that:

{{ para }}
```

Target Template:

```
{{answer_choices[choices.label.index(answerKey)]}}
```

Answer Choices Template:

```
{{choices.text | join("|||")}}
```

Input Template:

```
Read the passage below and choose the right answer to the following
question (choices are {{ answer_choices | join(" or ") }} ):

{{ para }}

{% if '_____' in question %}
{{ question | trim(".?!") | replace("_____", answer_choices | join(" or
")) }}{{ "?" }}
{% else %}
{{ question | trim(".?!") }} {{ answer_choices | join(" or ") }}{{ "?"
}}
{% endif %}
```

Target Template:

```
{{answer_choices[choices.label.index(answerKey)]}}
```

Answer Choices Template:

```
{{choices.text | join("|||")}}
```

Input Template:

```
{{ para }}

Having read the above passage, choose the right answer to the following
question (choices are {{ answer_choices | join(" or ") }} ):

{% if '_____' in question %}
{{ question | trim(".?!") | replace("_____", answer_choices | join(" or
")) }}{{ "?" }}
{% else %}
{{ question | trim(".?!") }} {{ answer_choices | join(" or ") }}{{ "?"
}}
{% endif %}
```

Target Template:

```
{{answer_choices[choices.label.index(answerKey)]}}
```

Answer Choices Template:

```
{{choices.text | join("|||")}}
```

Input Template:

```
Given the fact that:

{{ para }}

Answer the question:

{% if '_____' in question %}
{{ question | trim(".?!") | replace("_____", answer_choices | join(" or
")) }}{{ "?" }}
{% else %}
{{ question | trim(".?!") }} {{ answer_choices | join(" or ") }}{{ "?"
}}
{% endif %}
```

Target Template:

```
{{answer_choices[choices.label.index(answerKey)]}}
```

Answer Choices Template:

```
{{choices.text | join("|||")}}
```

### 1.7.9 RACE HIGH

Dataset from Lai et al. (2017). Used in evaluation.

**Data Example**

| Key | Value |
|-----|-------|
| answer | D |
| article | Studies show that you may be lied to every day any... |
| example_id | high10001.txt |
| options | ['harmful', 'easy', 'interesting', 'common'] |
| question | From Para.1 we learn that lying is very   _  . |

**Prompts**

Prompt not for the original task intended by the dataset authors
Input Template:

```
{% set candidate = ["A", "B", "C", "D"] | choice %}
Article: {{article}}
Question: {{question}}
Yes or no, is the answer "{{
[options.0,options.1,options.2,options.3][{"A":0,"B":1,"C":2,"D":3}[answer]]
}}"?
```

Target Template:

```
{% if candidate == answer %}
Yes
{% else %}
No
{% endif %}
```

Answer Choices Template:

```
Yes ||| No
```

Prompt not for the original task intended by the dataset authors
Input Template:

```
Write a multi-choice question for the following article:
Article: {{article}}
```

Target Template:

```
Question:
{{question}}
Options:
{{"A"}} {{options.0}}
{{"B"}} {{options.1}}
{{"C"}} {{options.2}}
{{"D"}} {{options.3}}
Answer:
{{answer}}
```

Input Template:

```
I'm taking a test and have to guess the right answer to the question
after the article.
Article: {{article}}
Question: {{question}}
Options: {{"A"}}: {{options.0}}
{{"B"}}: {{options.1}}
{{"C"}}: {{options.2}}
{{"D"}}: {{options.3}}
```

Target Template:

```
{{answer}}
```

Answer Choices Template:

```
A ||| B ||| C ||| D
```

Input Template:

```
Read the article and select the best answer.
Article: {{article}}
Question: {{question}}
Options: {{"A"}}: {{options.0}}
{{"B"}}: {{options.1}}
{{"C"}}: {{options.2}}
{{"D"}}: {{options.3}}
```

Target Template:

```
{{answer}}
```

Answer Choices Template:

```
A ||| B ||| C ||| D
```

Prompt not for the original task intended by the dataset authors
Input Template:

```
Write a multi-choice question for the following article, with the given
choices and answer:
Article: {{article}}
Options:
{{"A"}} {{options.0}}
{{"B"}} {{options.1}}
{{"C"}} {{options.2}}
{{"D"}} {{options.3}}
Answer:
{{answer}} {{
[options.0,options.1,options.2,options.3][{"A":0,"B":1,"C":2,"D":3}[answer]]
}}
Question:
```

Target Template:

```
{{question}}
```

Input Template:

```
Read the following article and select the best answer.
Article: {{article}}
Question: {{question}}
- {{answer_choices | join("\n- ")}}
```

Target Template:

```
{{answer_choices[{"A":0,"B":1,"C":2,"D":3}[answer]]}}
```

Answer Choices Template:

```
{{ options | join("|||") }}
```

Input Template:

```
{{article}}
{{question}}
{{"A)"}} {{options.0}}
{{"B)"}} {{options.1}}
{{"C)"}} {{options.2}}
{{"D)"}} {{options.3}}
```

Target Template:

```
{{answer}}
```

Answer Choices Template:

```
A ||| B ||| C ||| D
```

Input Template:

```
Read the following article and answer the question.
Article: {{article}}
Question: {{question}}
Answer:
```

Target Template:

```
{{ answer_choices[{"A":0,"B":1,"C":2,"D":3}[answer]] }}
```

Answer Choices Template:

```
{{ options | join("|||") }}
```

### 1.7.10   RACE MIDDLE

Dataset from Lai et al. (2017). Used in evaluation.

**Data Example**

**Prompts**

Input Template:

| Key | Value |
|---|---|
| answer | C |
| article | Take a class at Dulangkou School, and you'll see l... |
| example_id | middle1.txt |
| options | ['take care of the whole group', 'make sure that e... |
| question | A discipline leader is supposed to  _  . |

```
Read the article and select the best answer.
Article: {{article}}
Question: {{question}}
Options: {{"A"}}: {{options.0}}
{{"B"}}: {{options.1}}
{{"C"}}: {{options.2}}
{{"D"}}: {{options.3}}
```

Target Template:

```
{{answer}}
```

Answer Choices Template:

```
A ||| B ||| C ||| D
```

Input Template:

```
Read the following article and answer the question.
Article: {{article}}
Question: {{question}}
Answer:
```

Target Template:

```
{{ answer_choices[{"A":0,"B":1,"C":2,"D":3}[answer]] }}
```

Answer Choices Template:

```
{{ options | join("|||") }}
```

Prompt not for the original task intended by the dataset authors
Input Template:

```
{% set candidate = ["A", "B", "C", "D"] | choice %}
Article: {{article}}
Question: {{question}}
Yes or no, is the answer "{{
[options.0,options.1,options.2,options.3][{"A":0,"B":1,"C":2,"D":3}[answer]]
}}"?
```

Target Template:

```
{% if candidate == answer %}
Yes
{% else %}
No
{% endif %}
```

**Answer Choices Template:**

```
Yes ||| No
```

**Input Template:**

```
{{article}}
{{question}}
{{"A)"}} {{options.0}}
{{"B)"}} {{options.1}}
{{"C)"}} {{options.2}}
{{"D)"}} {{options.3}}
```

**Target Template:**

```
{{answer}}
```

**Answer Choices Template:**

```
A ||| B ||| C ||| D
```

**Input Template:**

```
Read the following article and select the best answer.
Article: {{article}}
Question: {{question}}
- {{answer_choices | join("\n- ")}}
```

**Target Template:**

```
{{answer_choices[{"A":0,"B":1,"C":2,"D":3}[answer]]}}
```

**Answer Choices Template:**

```
{{ options | join("|||") }}
```

Prompt not for the original task intended by the dataset authors
**Input Template:**

```
Write a multi-choice question for the following article, with the given
choices and answer:
Article: {{article}}
```

```
Options:
{{"A"}} {{options.0}}
{{"B"}} {{options.1}}
{{"C"}} {{options.2}}
{{"D"}} {{options.3}}
Answer:
{{answer}} {{
[options.0,options.1,options.2,options.3][{"A":0,"B":1,"C":2,"D":3}[answer]]
}}
Question:
```

**Target Template:**

```
{{question}}
```

Prompt not for the original task intended by the dataset authors
**Input Template:**

```
Write a multi-choice question for the following article:
Article: {{article}}
```

**Target Template:**

```
Question:
{{question}}
Options:
{{"A"}} {{options.0}}
{{"B"}} {{options.1}}
{{"C"}} {{options.2}}
{{"D"}} {{options.3}}
Answer:
{{answer}}
```

**Input Template:**

```
I'm taking a test and have to guess the right answer to the question
after the article.
Article: {{article}}
Question: {{question}}
Options: {{"A"}}: {{options.0}}
{{"B"}}: {{options.1}}
{{"C"}}: {{options.2}}
{{"D"}}: {{options.3}}
```

**Target Template:**

```
{{answer}}
```

**Answer Choices Template:**

```
A ||| B ||| C ||| D
```

### 1.7.11  SCIQ

Dataset from Johannes Welbl (2017). Used in training.

**Data Example**

| Key | Value |
|---|---|
| question | What type of organism is commonly used in preparat... |
| distractor3 | viruses |
| distractor1 | protozoa |
| distractor2 | gymnosperms |
| correct_answer | mesophilic organisms |
| support | Mesophiles grow best in moderate temperature, typi... |

**Prompts**

Input Template:

```
Q: {{question}}

A:
```

Target Template:

```
{{answer_choices[3]}}
```

Answer Choices Template:

```
{{distractor1}} ||| {{distractor2}} ||| {{distractor3}} |||
{{correct_answer}}
```

Prompt not for the original task intended by the dataset authors
Input Template:

```
{% set order = [[0, 1, 2, 3], [0, 1, 3, 2], [0, 2, 1, 3], [0, 2, 3, 1],
[0, 3, 1, 2], [0, 3, 2, 1],
                          [1, 0, 2, 3], [1, 0, 3, 2], [1, 2, 0, 3],
                          [1, 2, 3, 0], [1, 3, 0, 2], [1, 3, 2, 0],
                          [2, 1, 0, 3], [2, 1, 0, 2], [2, 0, 1, 3],
                          [2, 0, 3, 1], [2, 3, 1, 0], [2, 3, 0, 1],
                          [3, 1, 2, 0], [3, 1, 0, 2], [3, 2, 1, 0],
                          [3, 2, 0, 1], [3, 0, 1, 2], [3, 0, 2, 1]] |
                          choice %}
Q: {{question}}

 Choices:

- {{ answer_choices[order[0]] }}

- {{ answer_choices[order[1]] }}

- {{ answer_choices[order[2]] }}
```

```
- {{ answer_choices[order[3]] }}

A:
```

Target Template:

```
{{answer_choices[3]}}
```

Answer Choices Template:

```
{{distractor1}} ||| {{distractor2}} ||| {{distractor3}} |||
{{correct_answer}}
```

Input Template:

```
{% set order = [[0, 1, 2, 3], [0, 1, 3, 2], [0, 2, 1, 3], [0, 2, 3, 1],
[0, 3, 1, 2], [0, 3, 2, 1],
                              [1, 0, 2, 3], [1, 0, 3, 2], [1, 2, 0, 3],
                              [1, 2, 3, 0], [1, 3, 0, 2], [1, 3, 2, 0],
                              [2, 1, 0, 3], [2, 1, 0, 2], [2, 0, 1, 3],
                              [2, 0, 3, 1], [2, 3, 1, 0], [2, 3, 0, 1],
                              [3, 1, 2, 0], [3, 1, 0, 2], [3, 2, 1, 0],
                              [3, 2, 0, 1], [3, 0, 1, 2], [3, 0, 2, 1]] |
                              choice %}
Q: {{question}}

Read this paragraph and choose the correct option from the provided
answers:

{{support}}

 Choices:

- {{ answer_choices[order[0]] }}

- {{ answer_choices[order[1]] }}

- {{ answer_choices[order[2]] }}

- {{ answer_choices[order[3]] }}

A:
```

Target Template:

```
{{answer_choices[3]}}
```

Answer Choices Template:

```
{{distractor1}} ||| {{distractor2}} ||| {{distractor3}} |||
{{correct_answer}}
```

Input Template:

```
{% set order = [[0, 1, 2, 3], [0, 1, 3, 2], [0, 2, 1, 3], [0, 2, 3, 1],
[0, 3, 1, 2], [0, 3, 2, 1],
                            [1, 0, 2, 3], [1, 0, 3, 2], [1, 2, 0, 3],
                            [1, 2, 3, 0], [1, 3, 0, 2], [1, 3, 2, 0],
                            [2, 1, 0, 3], [2, 1, 0, 2], [2, 0, 1, 3],
                            [2, 0, 3, 1], [2, 3, 1, 0], [2, 3, 0, 1],
                            [3, 1, 2, 0], [3, 1, 0, 2], [3, 2, 1, 0],
                            [3, 2, 0, 1], [3, 0, 1, 2], [3, 0, 2, 1]] |
                            choice %}
Answer the following question given this paragraph:

{{support}}

Q: {{question}}

 Choices:

- {{ answer_choices[order[0]] }}

- {{ answer_choices[order[1]] }}

- {{ answer_choices[order[2]] }}

- {{ answer_choices[order[3]] }}

A:
```

Target Template:

```
{{answer_choices[3]}}
```

Answer Choices Template:

```
{{distractor1}} ||| {{distractor2}} ||| {{distractor3}} |||
{{correct_answer}}
```

Input Template:

```
Answer the following question given this paragraph:

{{support}}

Q: {{question}}

A:
```

Target Template:

```
{{answer_choices[3]}}
```

**Answer Choices Template:**

```
{{distractor1}} ||| {{distractor2}} ||| {{distractor3}} |||
{{correct_answer}}
```

### 1.7.12   SOCIAL_I_QA

**Data Example**

| Key | Value |
|---|---|
| answerA | like attending |
| answerB | like staying home |
| answerC | a good friend to have |
| context | Cameron decided to have a barbecue and gathered he... |
| label | 1 |
| question | How would Others feel as a result? |

**Prompts**

Input Template:

```
I heard that {{context}}

And I was wondering {{question}}
```

Target Template:

```
{{answer_choices[label | int - 1]}}
```

Answer Choices Template:

```
{{answerA}} ||| {{answerB}} ||| {{answerC}}
```

Input Template:

```
{{context}}

Given the context: {{question}}

Possible answers: {{answer_choices | join(", ")}}
```

Target Template:

```
{{answer_choices[label | int - 1]}}
```

**Answer Choices Template:**

```
{{answerA}} ||| {{answerB}} ||| {{answerC}}
```

**Input Template:**

```
{% set random_answer_id = range(0,2) | choice%}
{% set answers = [answerA, answerB, answerC] %}
{{context}}

Given the question "{{question}}", is "{{answers[random_answer_id]}}" a
valid answer?
```

**Target Template:**

```
{% if (label | int) - 1 == random_answer_id %}
    Yes
{% else %}
    No
{% endif %}
```

**Answer Choices Template:**

```
Yes ||| No
```

Prompt not for the original task intended by the dataset authors
**Input Template:**

```
{{context}}

Given that the answer to a question is "{{{"1": answerA, "2": answerB,
"3": answerC}[label]}}", what is the question?
```

**Target Template:**

```
{{question}}
```

**Input Template:**

```
{{context}}

Given the context: {{question}}
```

**Target Template:**

```
{{answer_choices[label | int - 1]}}
```

Answer Choices Template:

```
{{answerA}} ||| {{answerB}} ||| {{answerC}}
```

Input Template:

```
Context: {{context}}

Question: {{question}}

Which one of these answers best answers the question according to the
context?

A: {{answerA}}

B: {{answerB}}

C: {{answerC}}
```

Target Template:

```
{{{"1": "A", "2": "B", "3": "C"}[label]}}
```

Answer Choices Template:

```
A ||| B ||| C
```

### 1.7.13  SUPER_GLUE BOOLQ

Dataset from Clark et al. (2019). Used in evaluation.

**Data Example**

| Key | Value |
|-----|-------|
| idx | 0 |
| label | 1 |
| passage | Persian language -- Persian (/prn, -n/), al... |
| question | do iran and afghanistan speak the same language |

**Prompts**

Input Template:

```
Passage: {{passage}}

After reading this passage, I have a question: {{question}}? True or
False?
```

Target Template:

```
{% if label != -1 %}
{{answer_choices[label]}}
{% endif %}
```

Answer Choices Template:

```
False ||| True
```

Prompt from Brown et al. (2020)
Input Template:

```
{{ passage }}
Question: {{ question }}
Answer:
```

Target Template:

```
{% if label != -1 %}
{{ answer_choices[label] }}
{% endif %}
```

Answer Choices Template:

```
No ||| Yes
```

Input Template:

```
{{ passage }}

Having read that, I wonder {{ question }}?
```

Target Template:

```
{% if label != -1 %}
{{ answer_choices[label] }}
{% endif %}
```

Answer Choices Template:

```
No ||| Yes
```

Input Template:

```
Text: {{passage}}

Answer the following yes/no question: {{question}}? Yes or no?
```

Target Template:

```
{% if label != -1 %}
{{answer_choices[label]}}
{% endif %}
```

Answer Choices Template:

```
No ||| Yes
```

Input Template:

```
{{ passage }}

Having read that, could you tell me {{ question }}?
```

Target Template:

```
{% if label != -1 %}{{ answer_choices[label] }}
{% endif %}
```

Answer Choices Template:

```
No ||| Yes
```

Input Template:

```
EXAM
1. Answer by yes or no.

Document: {{passage}}
Question: {{question}}?
```

Target Template:

```
{% if label != -1 %}
{{answer_choices[label]}}
{% endif %}
```

Answer Choices Template:

```
No ||| Yes
```

Prompt from Schick and Schütze (2021)
Input Template:

```
Based on the following passage, {{ question }}? {{ passage }}
```

Target Template:

```
{% if label != -1 %}
{{ answer_choices[label] }}
{% endif %}
```

Answer Choices Template:

```
No ||| Yes
```

Input Template:

```
Exercise: read the text and answer the question by True or False.

Text: {{passage}}
Question: {{question}}?
```

Target Template:

```
{% if label != -1 %}
{{answer_choices[label]}}
{% endif %}
```

Answer Choices Template:

```
False ||| True
```

Prompt from Schick and Schütze (2021)
Input Template:

```
{{ passage }}
Based on the previous passage, {{ question }}?
```

Target Template:

```
{% if label != -1 %}{{ answer_choices[label] }}
{% endif %}
```

Answer Choices Template:

```
No ||| Yes
```

Input Template:

```
{{passage}}

Q: {{question}}? True or False?
```

Target Template:

```
{% if label != -1 %}
{{answer_choices[label]}}
{% endif %}
```

Answer Choices Template:

```
False ||| True
```

### 1.7.14 SUPER_GLUE MULTIRC

Dataset from Khashabi et al. (2018). Used in evaluation.

**Data Example**

| Key | Value |
|---|---|
| answer | Children, Gerd, or Dorian Popa |
| idx | {'paragraph': 0, 'question': 0, 'answer': 0} |
| label | 0 |
| paragraph | While this process moved along, diplomacy continue... |
| question | What did the high-level effort to persuade Pakista... |

**Prompts**

Input Template:

```
{{paragraph}}

Question: {{question}}
I found this answer "{{answer}}". Is that correct? Yes or no?
```

Target Template:

```
{% if label != -1 %}{{answer_choices[label]}}{% endif %}
```

Answer Choices Template:

```
No ||| Yes
```

Prompt from Schick and Schütze (2021)
Input Template:

```
{{ paragraph }}
Based on the previous passage, {{ question }}
Is "{{ answer }}" a correct answer?
```

**Target Template:**

```
{% if label != -1 %}{{ answer_choices[label] }}{% endif %}
```

**Answer Choices Template:**

```
No ||| Yes
```

**Input Template:**

```
{{paragraph}}
Question: {{question}}

I am grading my students' exercises. Is the answer "{{answer}}" correct?
```

**Target Template:**

```
{% if label != -1 %}{{answer_choices[label]}}{% endif %}
```

**Answer Choices Template:**

```
No ||| Yes
```

**Input Template:**

```
{{ paragraph }}
{{ question }}
Would it be good to answer "{{ answer }}"?
```

**Target Template:**

```
{% if label != -1 %}{{ answer_choices[label] }}{% endif %}
```

**Answer Choices Template:**

```
No ||| Yes
```

**Prompt from Schick and Schütze (2021)**
**Input Template:**

```
{{ paragraph }}
Question: {{ question }}
Is it {{ answer }}?
```

Target Template:

```
{% if label != -1 %}{{ answer_choices[label] }}{% endif %}
```

Answer Choices Template:

```
No ||| Yes
```

Input Template:

```
{{paragraph}}

Decide whether "{{answer}}" is a valid answer to the following question:
{{question}}
Answer yes or no.
```

Target Template:

```
{% if label != -1 %}{{answer_choices[label]}}{% endif %}
```

Answer Choices Template:

```
No ||| Yes
```

Prompt from Schick and Schütze (2021)
Input Template:

```
{{ paragraph }}
Question: {{ question }}
Is the correct answer {{ answer }}?
```

Target Template:

```
{% if label != -1 %}{{ answer_choices[label] }}{% endif %}
```

Answer Choices Template:

```
No ||| Yes
```

Input Template:

```
Is "{{answer}}" a correct answer to the following question?
Question: {{question}}

Rely on the following text: {{paragraph}}
```

Target Template:

```
{% if label != -1 %}{{answer_choices[label]}}{% endif %}
```

Answer Choices Template:

```
No ||| Yes
```

Input Template:

```
{{paragraph}}

Question: {{question}}
I think "{{answer}}" is a valid answer. Could you confirm? Yes or no?
```

Target Template:

```
{% if label != -1 %}{{answer_choices[label]}}{% endif %}
```

Answer Choices Template:

```
No ||| Yes
```

Input Template:

```
{{ paragraph }}
{{ question }}
I was going to say "{{ answer }}". Does that sound right?
```

Target Template:

```
{% if label != -1 %}{{ answer_choices[label] }}{% endif %}
```

Answer Choices Template:

```
No ||| Yes
```

### 1.7.15 WIKI_HOP ORIGINAL

Dataset from Welbl et al. (2018). Used in training.

**Data Example**

**Prompts**

Input Template:

| Key | Value |
| --- | --- |
| annotations | [] |
| answer | 1996 summer olympics |
| candidates | ['1996 summer olympics', 'olympic games', 'sport'] |
| id | WH_train_0 |
| question | participant_of juan rossell |
| supports | ['The 2004 Summer Olympic Games, officially known ... |

```
Information:
{% for support in supports %}
- {{ support }}
{% endfor %}

{% set question_split = question.split(' ') %}
What object entity has the relation of '{{ question_split[0] |
replace("_", " ")}}' with the subject '{{ question_split[1:] | join("
")}}'?

Choices:
- {{answer_choices | join("\n - ") }}
```

Target Template:

```
{{answer}}
```

Answer Choices Template:

```
{{candidates | join("|||")}}
```

Prompt not for the original task intended by the dataset authors
Input Template:

```
Information:
{% for support in supports %}
- {{ support }}
{% endfor %}

{% set question_split = question.split(' ') %}
What is the relationship between '{{ question_split[1:] | join(" ")}}'
and '{{answer}}'?
```

Target Template:

```
{{ question_split[0] | replace("_", " ") }}
```

Prompt not for the original task intended by the dataset authors
Input Template:

```
Information:
{% for support in supports %}
- {{ support }}
{% endfor %}
```

```
{% set question_split = question.split(' ') %}
What entity does '{{ question_split[1:] | join(" ")}}' has the relation
'{{ question_split[0] | replace("_", " ") }}' with?
```

Target Template:

```
{{answer}}
```

Prompt not for the original task intended by the dataset authors
Input Template:

```
Information:
{% for support in supports %}
- {{ support }}
{% endfor %}

{% set question_split = question.split(' ') %}
Given the paragraphs above, decide what entity has the relation '{{
question_split[0] | replace("_", " ") }}' with '{{answer}}'.
```

Target Template:

```
{{ question_split[1:] | join(" ")}}
```

Input Template:

```
Information:
{% for support in supports %}
- {{ support }}
{% endfor %}

{% set question_split = question.split(' ') %}
Given the information above, choose from the list below the object
entity that exhibits the relation '{{ question_split[0] | replace("_", "
")}}' with the subject '{{ question_split[1:] | join(" ")}}'.

Choices:
- {{answer_choices | join("\n - ") }}
```

Target Template:

```
{{answer}}
```

Answer Choices Template:

```
{{candidates | join("|||")}}
```

Input Template:

```
Information:
{% for support in supports %}
- {{ support }}
{% endfor %}

{% set question_split = question.split(' ') %}
After reading the paragraphs above, we are interested in knowing the
entity with which '{{ question_split[1:] | join(" ")}}' exhibits the
relationship of '{{ question_split[0] | replace("_", " ")}}'. Find the
answer from the choices below.

Choices:
- {{answer_choices | join("\n - ") }}
```

Target Template:

```
{{answer}}
```

Answer Choices Template:

```
{{candidates | join("|||")}}
```

Prompt not for the original task intended by the dataset authors
Input Template:

```
Information:
{% for support in supports %}
- {{ support }}
{% endfor %}

{% set question_split = question.split(' ') %}
Given the information, choose the subject and object entities that have
the relation of '{{ question_split[0] | replace("_", " ") }}'.
```

Target Template:

```
{{ question_split[1:] | join(" ") }} , {{answer}}
```

Input Template:

```
Information:
{% for support in supports %}
- {{ support }}
{% endfor %}

{% set question_split = question.split(' ') %}
After reading the paragraphs above, choose the best answer for the
entity that related to '{{ question_split[1:] | join(" ")}}' with the
relationship of '{{ question_split[0] | replace("_", " ")}}'.

Choices:
- {{answer_choices | join("\n - ") }}
```

Target Template:

```
{{answer}}
```

Answer Choices Template:

```
{{candidates | join("|||")}}
```

Input Template:

```
Information:
{% for support in supports %}
- {{ support }}
{% endfor %}

{% set question_split = question.split(' ') %}
'{{ question_split[1:] | join(" ")}}' is related to which object entity
through the relation of '{{ question_split[0] | replace("_", " ")}}'?

Choices:
- {{answer_choices | join("\n - ") }}
```

Target Template:

```
{{answer}}
```

Answer Choices Template:

```
{{candidates | join("|||")}}
```

### 1.7.16  WIQA

Dataset from Tandon et al. (2019). Used in training.

**Data Example**

| Key | Value |
|---|---|
| answer_label | more |
| answer_label_as_choice | A |
| choices | {'label': ['A', 'B', 'C'], 'text': ['more', 'less'... |
| metadata_graph_id | 144 |
| metadata_para_id | 1217 |
| metadata_path_len | 2 |
| metadata_question_id | influence_graph:1217:144:106#0 |
| metadata_question_type | INPARA_EFFECT |
| question_para_step | ['A tree produces seeds', 'The seeds are dispersed... |
| question_stem | suppose there will be fewer new trees happens, how... |

## Prompts

Prompt not for the original task intended by the dataset authors
Input Template:

```
-  {{ question_para_step[1:] | join("\n- ") }}

What might be the first step of the process?
```

Target Template:

```
{{ question_para_step | first }}
```

Prompt not for the original task intended by the dataset authors
Input Template:

```
{% set process_list = question_para_step[:-1] if question_para_step[-1]
== "" else question_para_step %}
-  {{ process_list[:-1] | join("\n- ") }}

What might be the last step of the process?
```

Target Template:

```
{{ process_list | last }}
```

Prompt not for the original task intended by the dataset authors
Input Template:

```
What is the missing first step of the following process:

-  {{ question_para_step[1:] | join("\n- ") }}
```

Target Template:

```
{{ question_para_step | first }}
```

Prompt not for the original task intended by the dataset authors
Input Template:

```
{% set process_list = question_para_step[:-1] if question_para_step[-1]
== "" else question_para_step %}
What is the final step of the following process:
-  {{ process_list[:-1] | join("\n- ") }}
```

Target Template:

```
{{ process_list | last }}
```

Input Template:

```
Process:
- {{ question_para_step | join("\n- ")}}

Question:
{{question_stem}}

How does the supposed perturbation influence the second effect
mentioned. Answer by {{"more, less or no effect"}}
```

Target Template:

```
{{answer_label|replace("_", " ")}}
```

Prompt not for the original task intended by the dataset authors
Input Template:

```
Process:

- {{ question_para_step | join("\n- ") }}

{{question_stem}}

Which of the following is the supposed perturbation?

- {{"directly impacting a step of the process"}}
- {{"indirectly impacting a step of the process"}}
- {{"not impacting any step of the process"}}
```

Target Template:

```
{{{"EXOGENOUS_EFFECT": "indirectly impacting a step of the process",
"OUTOFPARA_DISTRACTOR": "not impacting any step of the process",
"INPARA_EFFECT": "directly impacting a step of the
process"}[metadata_question_type]}}
```

Input Template:

```
Process:
- {{ question_para_step | join("\n- ")}}

Question:
{{question_stem}}

- {{"A: more"}}
- {{"B: less"}}
- {{"C: no effect"}}
```

Target Template:

```
{{answer_label_as_choice}}
```

Prompt not for the original task intended by the dataset authors
Input Template:

```
Process:

- {{ question_para_step | join("\n- ") }}

Perturbation hypothesis:
{{question_stem}}

Does the supposed perturbation have an effect (direct or indirect) on
the process?
```

Target Template:

```
{{{"EXOGENOUS_EFFECT": "yes", "OUTOFPARA_DISTRACTOR": "no",
"INPARA_EFFECT": "yes"}[metadata_question_type]}}
```

### 1.7.17   PIQA

Dataset from Bisk et al. (2020). Used in evaluation.

#### Data Example

| Key   | Value                                      |
|-------|--------------------------------------------|
| goal  | When boiling butter, when it's ready, you can |
| label | 1                                          |
| sol1  | Pour it onto a plate                       |
| sol2  | Pour it into a jar                         |

#### Prompts

Input Template:

```
Goal: {{goal}}

Which is the correct ending?
- {{sol1}}
- {{sol2}}

Answer:
```

Target Template:

```
{{answer_choices[label]}}
```

Answer Choices Template:

```
{{sol1}} ||| {{sol2}}
```

**Input Template:**

```
{{"Solution 1"}}: {{sol1}}
{{"Solution 2"}}: {{sol2}}

Goal: {{goal}}

Given the goal, what is the correct solution?

Answer by copying the correct solution
```

**Target Template:**

```
{{answer_choices[label]}}
```

**Answer Choices Template:**

```
{{sol1}} ||| {{sol2}}
```

**Input Template:**

```
Sentence: {{goal}}

Choice {{answer_choices[0]}}: {{sol1}}

Choice {{answer_choices[1]}}: {{sol2}}

What is the index of the correct choice for ending for the sentence?

Answer:
```

**Target Template:**

```
{{answer_choices[label]}}
```

**Answer Choices Template:**

```
1 ||| 2
```

Prompt not for the original task intended by the dataset authors
**Input Template:**

```
Given a goal and a wrong solution, rewrite it to give a correct
solution.
Goal: {{goal}}
Solution: {{[sol1, sol2][1 - label]}}
Corrected solution:
```

Target Template:

```
{{[sol1, sol2][label]}}
```

Input Template:

```
Finish the following sentence with the best choice: {{goal}}

Choices:
- {{sol1}}
- {{sol2}}

Answer:
```

Target Template:

```
{{answer_choices[label]}}
```

Answer Choices Template:

```
{{sol1}} ||| {{sol2}}
```

Prompt not for the original task intended by the dataset authors
Input Template:

```
{{goal}} {{sol2}}
Does this phrase make sense?
```

Target Template:

```
{{answer_choices[label]}}
```

Answer Choices Template:

```
No ||| Yes
```

Input Template:

```
Given a goal and 2 solutions, choose the most appropriate solution.
Goal: {{goal}}
- {{"Solution 1"}}: {{sol1}}
- {{"Solution 2"}}: {{sol2}}

Answer by returning either {{"Solution 1"}} or {{"Solution 2"}}
```

Target Template:

```
{{answer_choices[label]}}
```

Answer Choices Template:

```
Solution 1 ||| Solution 2
```

Prompt not for the original task intended by the dataset authors
Input Template:

```
Given a sentence, correct it if it doesn't make sense. If it makes
sense, just return it as the answer.
Input: {{goal}} {{sol2[0].lower() + sol2[1:]}}
Output:
```

Target Template:

```
{{goal}} {{[sol1[0].lower() + sol1[1:], sol2[0].lower() +
sol2[1:]][label]}}
```

Prompt not for the original task intended by the dataset authors
Input Template:

```
{{goal}}
```

Target Template:

```
{{[sol1[0].lower() + sol1[1:], sol2[0].lower() + sol2[1:]][label]}}
```

Prompt not for the original task intended by the dataset authors
Input Template:

```
Does this phrase make sense?
{{goal}} {{sol1[0].lower() + sol1[1:]}}
Answer with {{answer_choices[0]}} or {{answer_choices[1]}}
```

Target Template:

```
{{answer_choices[label]}}
```

Answer Choices Template:

```
Yes ||| No
```

Prompt not for the original task intended by the dataset authors
Input Template:

```
Sentence: {{goal}} {{sol1[0].lower() + sol1[1:]}}
If the sentence does not make sense, correct it so that it does make
sense. Otherwise, just copy it.
Answer:
```

Target Template:

```
{{goal}} {{[sol1[0].lower() + sol1[1:], sol2[0].lower() +
sol2[1:]][label]}}
```

## 1.8 SENTIMENT

### 1.8.1 AMAZON_POLARITY

Dataset from McAuley and Leskovec (2013). Used in training.

**Data Example**

| Key | Value |
|---|---|
| content | This sound track was beautiful! It paints the sene... |
| label | 1 |
| title | Stuning even for the non-gamer |

**Prompts**

Input Template:

```
Title: {{title}}
Review: {{content}}
Is the review positive or negative?
```

Target Template:

```
{{answer_choices[label]}}
```

Answer Choices Template:

```
Negative ||| Positive
```

Input Template:

```
Based on this review, would the user recommend this product?
===
Review: {{content}}
Answer:
```

Target Template:

```
{{answer_choices[label]}}
```

Answer Choices Template:

```
No ||| Yes
```

Input Template:

```
Is this product review positive?
Title: {{title}}
Review: {{content}}
Answer:
```

Target Template:

```
{{answer_choices[label]}}
```

Answer Choices Template:

```
No ||| Yes
```

Input Template:

```
Title: {{title}}
Review: {{content}}
Is this product review negative?
```

Target Template:

```
{{answer_choices[label]}}
```

Answer Choices Template:

```
Yes ||| No
```

Input Template:

```
Title: {{title}}
Review: {{content}}
Does this product review convey a negative or positive sentiment?
```

Target Template:

```
{{answer_choices[label]}}
```

Answer Choices Template:

```
Negative ||| Positive
```

Input Template:

```
Is there a negative or positive tone to this product review?
===
Title: {{title}}
Review: {{content}}
Answer:
```

Target Template:

```
{{answer_choices[label]}}
```

Answer Choices Template:

```
Negative ||| Positive
```

Input Template:

```
Here is a review left by a customer on a product. Would you say he was
{{answer_choices[1]}} or {{answer_choices[0]}}?
Title: {{title}}
Review: {{content}}
```

Target Template:

```
{{answer_choices[label]}}
```

Answer Choices Template:

```
dissatisfied ||| satisfied
```

Input Template:

```
You are considering whether to buy a product. You look at the reviews.
Would the following review {{answer_choices[0]}} or
{{answer_choices[1]}} the chances of you buying the product?
Review title: {{title}}
Product review: {{content}}
```

Target Template:

```
{{answer_choices[label]}}
```

Answer Choices Template:

```
decrease ||| increase
```

Input Template:

```
Title: {{title}}
Product review: {{content}}
Would you say this review depicts the product in a {{answer_choices[1]}}
or {{answer_choices[0]}} light?
```

Target Template:

```
{{answer_choices[label]}}
```

Answer Choices Template:

```
unflattering ||| flattering
```

### 1.8.2   APP_REVIEWS

Dataset from ?. Used in training.

#### Data Example

| Key | Value |
|---|---|
| date | October 12 2016 |
| package_name | com.mantz_it.rfanalyzer |
| review | Great app! The new version now works on my Bravia ... |
| star | 4 |

#### Prompts

Prompt not for the original task intended by the dataset authors
Input Template:

```
Given this review: "{{review}}"
Would you recommend this app to a friend? {{answer_choices[0]}},
{{answer_choices[1]}}, {{answer_choices[2]}}, {{answer_choices[3]}}, or
{{answer_choices[4]}}?
```

Target Template:

```
{{answer_choices[star-1]}}
```

Answer Choices Template:

```
Not at all ||| No ||| Maybe ||| Yes ||| Definitely
```

Prompt not for the original task intended by the dataset authors
Input Template:

```
Generate a {{star}}-star review (1 being lowest and 5 being highest)
about an app with package {{package_name}}.
```

Target Template:

```
{{review}}
```

Prompt not for the original task intended by the dataset authors
Input Template:

```
What would be the -rating of this review ( being the lowest and  being
the highest)? "{{review}}"
```

Target Template:

```
{{answer_choices[star-1]}}
```

Answer Choices Template:

```
 |||   |||   |||   |||
```

Prompt not for the original task intended by the dataset authors
Input Template:

```
On a scale of 1-5 (with 1 being least favorable and 5 being most
favorable), how would you rate this review? "{{review}}"
```

Target Template:

```
{{star}}
```

### 1.8.3   IMDB

Dataset from Maas et al. (2011). Used in training.

**Data Example**

| Key | Value |
|---|---|
| text | Bromwell High is a cartoon comedy. It ran at the s... |
| label | 1 |

## Prompts

Input Template:

```
The following movie review expresses what sentiment? {{text}}
```

Target Template:

```
{{ answer_choices [label] }}
```

Answer Choices Template:

```
negative ||| positive
```

Input Template:

```
{{text}} Did the reviewer find this movie {{"good or bad"}}?
```

Target Template:

```
{{ answer_choices [label] }}
```

Answer Choices Template:

```
bad ||| good
```

Input Template:

```
{{text}}
Is this review {{"positive or negative"}}?
```

Target Template:

```
{{answer_choices[label] }}
```

Answer Choices Template:

```
negative ||| positive
```

Input Template:

```
{{text}} How does the viewer feel about the movie?
```

Target Template:

```
{{ answer_choices [label] }}
```

Answer Choices Template:

```
negative ||| positive
```

Input Template:

```
{{text}} What sentiment does the writer express for the movie?
```

Target Template:

```
{{ answer_choices [label] }}
```

Answer Choices Template:

```
negative ||| positive
```

Input Template:

```
{{text}} The sentiment expressed for the movie is
```

Target Template:

```
{{ answer_choices [label] }}
```

Answer Choices Template:

```
negative ||| positive
```

Input Template:

```
{{text}} What is the sentiment expressed in this text?
```

Target Template:

```
{{ answer_choices [label] }}
```

Answer Choices Template:

```
negative ||| positive
```

Prompt not for the original task intended by the dataset authors
Input Template:

```
{{text}} This is definitely not a
```

Target Template:

```
{{ answer_choices [1-label]}} review.
```

Answer Choices Template:

```
negative ||| positive
```

Input Template:

```
{{text}} Did the reviewer enjoy the movie?
```

Target Template:

```
{{ answer_choices [label] }}
```

Answer Choices Template:

```
No ||| Yes
```

Input Template:

```
{{text}} What is the sentiment expressed by the reviewer for the movie?
```

Target Template:

```
{{ answer_choices [label] }}
```

Answer Choices Template:

```
negative ||| positive
```

Input Template:

```
{{text}} How does the reviewer feel about the movie?
```

Target Template:

```
{{ answer_choices [label] }}
```

Answer Choices Template:

```
They didn't like it! ||| They loved it
```

### 1.8.4 ROTTEN_TOMATOES

Dataset from Pang and Lee (2005). Used in training.

**Data Example**

| Key | Value |
|-----|-------|
| text | the rock is destined to be the 21st century's new ... |
| label | 1 |

**Prompts**

Input Template:

```
{{text}} Did the reviewer find this movie {{"good or bad"}}?
```

Target Template:

```
{{ answer_choices [label] }}
```

Answer Choices Template:

```
bad ||| good
```

Input Template:

```
{{text}} What is the sentiment expressed in this text?
```

Target Template:

```
{{ answer_choices [label] }}
```

Answer Choices Template:

```
negative ||| positive
```

Input Template:

```
{{text}}
Is this review {{"positive or negative"}}?
```

Target Template:

```
{{answer_choices[label] }}
```

Answer Choices Template:

```
negative ||| positive
```

Input Template:

```
{{text}} Did the reviewer enjoy the movie?
```

Target Template:

```
{{ answer_choices [label] }}
```

Answer Choices Template:

```
No ||| Yes
```

Input Template:

```
{{text}} How does the reviewer feel about the movie?
```

Target Template:

```
{{ answer_choices [label] }}
```

Answer Choices Template:

```
They didn't like it ||| They loved it
```

Input Template:

```
{{text}} The sentiment expressed for the movie is
```

Target Template:

```
{{ answer_choices [label] }}
```

Answer Choices Template:

```
negative ||| positive
```

Input Template:

```
{{text}} What sentiment does the writer express for the movie?
```

Target Template:

```
{{ answer_choices [label] }}
```

Answer Choices Template:

```
negative ||| positive
```

Input Template:

```
The following movie review expresses what sentiment? {{text}}
```

Target Template:

```
{{ answer_choices [label] }}
```

Answer Choices Template:

```
negative ||| positive
```

Input Template:

```
{{text}} What is the sentiment expressed by the reviewer for the movie?
```

Target Template:

```
{{ answer_choices [label] }}
```

Answer Choices Template:

```
negative ||| positive
```

Input Template:

```
{{text}} How does the viewer feel about the movie?
```

Target Template:

```
{{ answer_choices [label] }}
```

Answer Choices Template:

```
negative ||| positive
```

### 1.8.5 YELP_REVIEW_FULL

Dataset from Zhang et al. (2015a). Used in training.

**Data Example**

| Key | Value |
|---|---|
| label | 4 |
| text | dr. goldberg offers everything i look for in a gen... |

**Prompts**

Input Template:

```
{{ text }}
So I would like to give it
```

Target Template:

```
{{ answer_choices[label] }}
```

Answer Choices Template:

```
1 star ||| 2 stars ||| 3 stars ||| 4 stars ||| 5 stars
```

Input Template:

```
{{ text }}
===
Based on that, my rating is
```

Target Template:

```
{{ answer_choices[label] }}
```

Answer Choices Template:

```
1 star ||| 2 stars ||| 3 stars ||| 4 stars ||| 5 stars
```

Input Template:

```
Review text:
{{ text }}

Stars:
```

Target Template:

```
{{ answer_choices[label] }}
```

Answer Choices Template:

```
1 star ||| 2 stars ||| 3 stars ||| 4 stars ||| 5 stars
```

Input Template:

```
{{ text }} My rating for this place is
```

Target Template:

```
{{ answer_choices[label] }}
```

Answer Choices Template:

```
1 star ||| 2 stars ||| 3 stars ||| 4 stars ||| 5 stars
```

Input Template:

```
Review text:
{{ text }}

Review score (between 1 and 5):
```

Target Template:

```
{{ answer_choices[label] }}
```

Answer Choices Template:

```
1 ||| 2 ||| 3 ||| 4 ||| 5
```

Input Template:

```
Review: {{text}}
On a scale of 1 to 5, I would give this product
```

Target Template:

```
{{ answer_choices[label] }}
```

Answer Choices Template:

```
1 ||| 2 ||| 3 ||| 4 ||| 5
```

Input Template:

```
Review text:
{{ text }}

Review rating:
```

Target Template:

```
{{ answer_choices[label] }}
```

Answer Choices Template:

```
1 star ||| 2 stars ||| 3 stars ||| 4 stars ||| 5 stars
```

## 1.9 SENTENCE COMPLETION

### 1.9.1 SUPER_GLUE COPA

Dataset from Roemmele et al. (2011). Used in evaluation.

**Data Example**

| Key | Value |
|---|---|
| choice1 | The sun was rising. |
| choice2 | The grass was cut. |
| idx | 0 |
| label | 0 |
| premise | My body cast a shadow over the grass. |
| question | cause |

**Prompts**

Input Template:

```
Exercise: choose the most plausible alternative.

{{ premise }} {% if question == "cause" %} because... {% else %} so...
{% endif %}
- {{choice1}}
- {{choice2}}
```

Target Template:

```
{% if label != -1 %}{{ answer_choices[label] }}{%endif%}
```

Answer Choices Template:

```
{{choice1}} ||| {{choice2}}
```

Input Template:

```
{% if question == "effect" %}
{{ premise }} What could happen next, "{{ answer_choices[0] }}" or "{{
answer_choices[1] }}"?
```

Target Template:

```
{% if label != -1 %}{{ answer_choices[label] }}{%endif%}
{% endif %}
```

Answer Choices Template:

```
{{choice1}} ||| {{choice2}}
```

Input Template:

```
{{ premise }}

I am hesitating between two options. Help me choose the more likely {%
if question == "cause" %} cause: {% else %} effect: {% endif %}
- {{choice1}}
- {{choice2}}
```

Target Template:

```
{% if label != -1 %}{{ answer_choices[label] }}{%endif%}
```

Answer Choices Template:

```
{{choice1}} ||| {{choice2}}
```

Input Template:

```
{{ premise }} {% if question == "cause" %} This happened because... {%
else %} As a consequence... {% endif %}
Help me pick the more plausible option:
- {{choice1}}
- {{choice2}}
```

Target Template:

```
{% if label != -1 %}{{ answer_choices[label] }}{%endif%}
```

Answer Choices Template:

```
{{choice1}} ||| {{choice2}}
```

Prompt from Schick and Schütze (2021)
Input Template:

```
"{{ answer_choices[0] }}" or "{{ answer_choices[1] }}"? {{ premise }} {%
if question == "cause" %} because {% else %} so {% endif %}
```

Target Template:

```
{% if label != -1 %}{{ answer_choices[label] }}{% endif %}
```

Answer Choices Template:

```
{{choice1 }} ||| {{choice2}}
```

Input Template:

```
{% if question == "effect" %}
{{ premise }} As a result, "{{ answer_choices[0] }}" or "{{
answer_choices[1] }}"?
```

Target Template:

```
{% if label != -1 %}{{ answer_choices[label] }}{%endif%}
{% endif %}
```

Answer Choices Template:

```
{{choice1}} ||| {{choice2}}
```

Input Template:

```
{{ premise }}

What's the best option?
- {{choice1}}
- {{choice2}}

We are looking for {% if question == "cause" %} a cause {% else %} an
effect {% endif %}
```

Target Template:

```
{% if label != -1 %}{{answer_choices[label]}}{%endif%}
```

Answer Choices Template:

```
{{choice1}} ||| {{choice2}}
```

Input Template:

```
{% if question == "cause" %}
{{ premise }} Which may be caused by "{{ answer_choices[0] }}" or "{{
answer_choices[1] }}"?
```

Target Template:

```
{% if label != -1 %}{{ answer_choices[label] }}{%endif%}
{% endif %}
```

Answer Choices Template:

```
{{choice1}} ||| {{choice2}}
```

Input Template:

```
Pick the more likely continuation to the following sentence:
{{ premise }} {% if question == "cause" %} as a result of: {% else %} as
a consequence: {% endif %}
- {{choice1}}
- {{choice2}}
```

Target Template:

```
{% if label != -1 %}{{ answer_choices[label] }}{%endif%}
```

**Answer Choices Template:**

```
{{choice1}} ||| {{choice2}}
```

**Input Template:**

```
{{ premise }}

Select the most plausible {% if question == "cause" %} cause: {% else %}
effect: {% endif %}
- {{choice1}}
- {{choice2}}
```

**Target Template:**

```
{% if label != -1 %}{{ answer_choices[label] }}{%endif%}
```

**Answer Choices Template:**

```
{{choice1}} ||| {{choice2}}
```

**Input Template:**

```
{% if question == "cause" %}
{{ premise }} Why? "{{ answer_choices[0] }}" or "{{ answer_choices[1]
}}"?
```

**Target Template:**

```
{% if label != -1 %}{{ answer_choices[label] }}{%endif%}
{% endif %}
```

**Answer Choices Template:**

```
{{choice1}} ||| {{choice2}}
```

**Input Template:**

```
{{ premise }} {% if question == "cause" %} because... {% else %} so...
{% endif %}
Choose between:
- {{choice1}}
- {{choice2}}
```

**Target Template:**

```
{% if label != -1 %}{{ answer_choices[label] }}{%endif%}
```

Answer Choices Template:

```
{{choice1}} ||| {{choice2}}
```

### 1.9.2 HELLASWAG

Dataset from Zellers et al. (2019). Used in evaluation.

**Data Example**

| Key | Value |
|-----|-------|
| activity_label | Removing ice from car |
| ctx | Then, the man writes over the snow covering the wi... |
| ctx_a | Then, the man writes over the snow covering the wi... |
| ctx_b | then |
| endings | [', the man adds wax to the windshield and cuts it... |
| ind | 4 |
| label | 3 |
| source_id | activitynet~v_-1IBHYS3L-Y |
| split | train |
| split_type | indomain |

**Prompts**

Input Template:

```
Complete the description with an appropriate ending:
First, {{ ctx_a.lower() }} Then, {{ ctx_b.lower() }} ...

(a) {{ answer_choices[0] }}

(b) {{ answer_choices[1] }}

(c) {{ answer_choices[2] }}

(d) {{ answer_choices[3] }}
```

Target Template:

```
{{ answer_choices[label | int()] }}
```

Answer Choices Template:

```
{{endings | join(" ||| ")}}
```

Prompt not for the original task intended by the dataset authors
Input Template:

```
What is the topic of the sentence: {{ctx}}
```

Target Template:

```
{{activity_label}}
```

Prompt not for the original task intended by the dataset authors
Input Template:

```
Complete the sentence: {{ctx}}
```

Target Template:

```
{{answer_choices[label | int()]}}
```

Answer Choices Template:

```
{{endings | join(" ||| ")}}
```

Prompt not for the original task intended by the dataset authors
Input Template:

```
{{ctx}} {{endings[label | int()]}}
Can you identify the topic of the paragraph?
```

Target Template:

```
{{activity_label}}
```

Input Template:

```
{% set prompts = [
'Can you pick the correct ending for the sentence: ',
'The task is to generate the ending for the sentence: ',
'How does this sentence end? ',
'From the list of endings described below, what ending makes the most
sense for the sentence ',]
%}
{{prompts | choice}}
{{ctx}}

(a)  {{answer_choices[0]}}

(b)  {{answer_choices[1]}}

(c)  {{answer_choices[2]}}

(d)  {{answer_choices[3]}}
```

Target Template:

```
{{answer_choices [label | int()]}}
```

Answer Choices Template:

```
{{endings | join(" ||| ") }}
```

Prompt not for the original task intended by the dataset authors
Input Template:

```
{% set instance = [0, 1, 2, 3] | choice %}
Consider the following description: {{ ctx_a }}
Is the following an appropriate continuation?
{{ ctx_b }} {{ endings[instance] }}
Yes or No?
```

Target Template:

```
{% if label  == instance | string() %}
{{answer_choices[0]}}
{% else %}
{{answer_choices[1]}}
{% endif %}
```

Answer Choices Template:

```
Yes ||| No
```

Input Template:

```
How does this sentence end?
{{ctx}}

(a)  {{answer_choices[0]}}

(b)  {{answer_choices[1]}}

(c)  {{answer_choices[2]}}

(d)  {{answer_choices[3]}}

Hint: the topic of the sentence is {{activity_label}}
```

Target Template:

```
{{answer_choices [label | int()]}}
```

Answer Choices Template:

```
{{endings | join("|||")}}
```

Prompt not for the original task intended by the dataset authors
Input Template:

```
How would you start the sentence:
{{endings[label | int()]}}
```

Target Template:

```
{{ctx}}
```

Prompt not for the original task intended by the dataset authors
Input Template:

```
{% set instance = [0, 1, 2, 3] | choice %}
Consider the following text: {{ ctx_b }} {{ endings[instance] }}
Is it an appropriate continuation of the following text:
{{ ctx_a }} ?
Yes or No?
```

Target Template:

```
{% if label  == instance | string() %}
{{answer_choices[0]}}
{% else %}
{{answer_choices[1]}}
{% endif %}
```

Answer Choices Template:

```
Yes ||| No
```

Prompt not for the original task intended by the dataset authors
Input Template:

```
{{ ctx }}...
How does the description likely end?

Ending 1: {{ endings[0] }}

Ending 2: {{ endings[1] }}

Ending 3: {{ endings[2] }}

Ending 4: {{ endings[3] }}
```

Target Template:

```
{{ answer_choices[label | int()] }}
```

Answer Choices Template:

```
Ending 1 ||| Ending 2 ||| Ending 3 ||| Ending 4
```

Input Template:

```
If a description of a situation begins like this: {{ ctx }}... Then how
does it continue?

Ending 1: {{ endings[0] }}

Ending 2: {{ endings[1] }}

Ending 3: {{ endings[2] }}

Ending 4: {{ endings[3] }}
```

Target Template:

```
{{answer_choices[label | int()] }}
```

Answer Choices Template:

```
Ending 1 ||| Ending 2 ||| Ending 3 ||| Ending 4
```

## 1.10   STRUCTURE TO TEXT

### 1.10.1   COMMON_GEN

Dataset from Lin et al. (2020). Used in training.

**Data Example**

| Key | Value |
|---|---|
| concept_set_idx | 0 |
| concepts | ['ski', 'mountain', 'skier'] |
| target | Skier skis down the mountain |

**Prompts**

Input Template:

```
Ignoring the order of the concepts: {{ concepts | join(", ") }};
Generate a sentence with all the concepts :
```

Target Template:

```
{{target}}
```

Input Template:

```
Put the concepts together to form a sentence: {{ concepts | join(", ")
}}.
```

Target Template:

```
{{target}}
```

Input Template:

```
Construct a sentence with the word {{ concepts | choice }}.

Hint: Use {{concepts | join(", ")}} to restrict the output sentence.
```

Target Template:

```
{{target}}
```

Input Template:

```
{% set seq = [
'From the concepts mentioned below, generate a sentence:',
'Convert the concepts to a sentence:',
'Given the list of concepts, write a sentence:'
] %}
{{ seq | choice }}
{{ concepts | join(", ") }}
```

Target Template:

```
{{target}}
```

Prompt not for the original task intended by the dataset authors
Input Template:

```
What are the topics in the sentence: {{target}}
```

Target Template:

```
{{ concepts | join(", ") }}
```

Prompt not for the original task intended by the dataset authors
Input Template:

```
We have the sentence: {{target}};
Extract all the key concepts:
```

Target Template:

```
{{ concepts | join(", ") }}
```

Prompt not for the original task intended by the dataset authors
Input Template:

```
Can you write a sentence about the topic {{concepts | choice}}?
```

Target Template:

```
{{target}}
```

Input Template:

```
Humans can easily string together abstract concepts to form a coherent
sentence.
For example, with the concepts {{ concepts | join(", ") }}, a simple
sentence can be
```

Target Template:

```
{{target}}
```

Input Template:

```
Given the list of concepts: {{ concepts | join(", ") }};
Generate a sentence with all the concepts :
```

Target Template:

```
{{target}}
```

### 1.10.2 WIKI_BIO

Dataset from Lebret et al. (2016). Used in training.

**Data Example**

| Key | Value |
|---|---|
| input_text | {'table': {'column_header': ['name', 'nationality'... |
| target_text | walter extra is a german award-winning aerobatic p... |

## Prompts

Input Template:

```
Facts:
{% for n in range (input_text["table"]["column_header"]|length) %}
{% if input_text["table"]["column_header"][n] != "article_title" %}
- {{input_text["table"]["column_header"][n].replace("_"," ") }}:
{{input_text["table"]["content"][n] }}
{% endif %}
{% endfor %}
Based on these bullet points, write a short biography describing the
life of {{input_text["context"]}}.
```

Target Template:

```
{{target_text}}
```

Prompt not for the original task intended by the dataset authors
Input Template:

```
Read the bio below and try to give details on
{{input_text["context"]}}'s:
{% for n in range (input_text["table"]["column_header"]|length) %} {% if
input_text["table"]["column_header"][n] != "article_title" %}
- {{ input_text["table"]["column_header"][n].replace("_"," ") }}
{% endif %} {% endfor %}

Bio: {{target_text}}
```

Target Template:

```
{% for n in range (input_text["table"]["column_header"]|length) %}
{% if input_text["table"]["column_header"][n] != "article_title" %}
- {{ input_text["table"]["column_header"][n].replace("_"," ") }} is {{
input_text["table"]["content"][n] }}
{% endif %}
{% endfor %}
```

Prompt not for the original task intended by the dataset authors
Input Template:

```
What type of details about {{input_text["context"]}} can be gathered
from the following bio?

Bio: {{target_text}}
```

Target Template:

```
{% for n in range (input_text["table"]["column_header"]|length) %}
{% if input_text["table"]["column_header"][n] != "article_title" %}
- {{ input_text["table"]["column_header"][n].replace("_"," ") }}
{% endif %}
{% endfor %}
```

Prompt not for the original task intended by the dataset authors
Input Template:

```
{% for n in range (input_text["table"]["column_header"]|length) %}
{% if input_text["table"]["column_header"][n] != "article_title" and
input_text["table"]["column_header"][n] !="name" %}
- {{ input_text["table"]["column_header"][n].replace("_"," ") }} is {{
input_text["table"]["content"][n] }}
{% endif %}
{% endfor %}

Given the details above, guess who could this information be about.
```

Target Template:

```
{{input_text["context"]}}
```

Prompt not for the original task intended by the dataset authors
Input Template:

```
What key details about {{input_text["context"]}} can be extracted from
the following bio?

Bio: {{target_text}}
```

Target Template:

```
{% for n in range (input_text["table"]["column_header"]|length) %}
{% if input_text["table"]["column_header"][n] != "article_title" %}
- {{ input_text["table"]["column_header"][n].replace("_"," ") }} is {{
input_text["table"]["content"][n] }}
{% endif %}
{% endfor %}
```

## 1.11 SUMMARIZATION

### 1.11.1 CNN_DAILYMAIL 3.0.0

Dataset from See et al. (2017). Used in training.

**Data Example**

**Prompts**

| Key | Value |
|---|---|
| article | It's official: U.S. President Barack Obama wants l... |
| highlights | Syrian official: Obama climbed to the top of the t... |
| id | 0001d1afc246a7964130f43ae940af6bc6c57f01 |

Input Template:

```
Can you write an outline of the following article in a few points?

Article: {{article}}
```

Target Template:

```
{{highlights}}
```

Input Template:

```
Summarise the article:

{{article}}
```

Target Template:

```
{{highlights}}
```

Input Template:

```
In 2 or 3 sentences, what are the main points one should remember from
this news article?

Article: {{article}}
```

Target Template:

```
{{highlights}}
```

Input Template:

```
Could you please generate a TLDR (Too Long Didn't Read) summary of the
following news article?

Article: {{article}}
```

Target Template:

```
{{highlights}}
```

Input Template:

```
Condense the article down to the essentials to present it in the form of
short cards in mobile news apps:

{{article}}
```

Target Template:

```
{{highlights}}
```

Prompt not for the original task intended by the dataset authors
Input Template:

```
Generate a story from key plot points:

{{highlights}}
```

Target Template:

```
{{article}}
```

Input Template:

```
Sum the following article in brief: {{article}}
```

Target Template:

```
{{highlights}}
```

Input Template:

```
Extract key points from the article based on which the stock market
could react:

{{article}}
```

Target Template:

```
{{highlights}}
```

Prompt not for the original task intended by the dataset authors
Input Template:

```
What details would you include in a storyline to make it more engaging
and informative?

{{highlights}}
```

Target Template:

```
{{article}}
```

### 1.11.2 GIGAWORD

Dataset from Graff et al. (2003). Used in training.

**Data Example**

| Key | Value |
|---|---|
| document | australia 's current account deficit shrunk by a r... |
| summary | australian current account deficit narrows sharply |

**Prompts**

Input Template:

```
{{document}}

===

Generate a title for this article:
```

Target Template:

```
{{summary}}
```

Prompt not for the original task intended by the dataset authors
Input Template:

```
Title: {{summary}}
```

Target Template:

```
{{document}}
```

Input Template:

```
Make a title for this article: {{document}}
```

Target Template:

```
{{summary}}
```

Input Template:

```
First sentence of the article: {{document}}

Title:
```

Target Template:

```
{{summary}}
```

Prompt from Radford et al. (2019)
Input Template:

```
{{document}}

TL;DR:
```

Target Template:

```
{{summary}}
```

Input Template:

```
{{document}}

===

Given the above sentence, write its title:
```

Target Template:

```
{{summary}}
```

Input Template:

```
Write a title for this sentence: {{document}}

Title:
```

Target Template:

```
{{summary}}
```

Input Template:

```
{{document}} In a nutshell,
```

Target Template:

```
{{summary}}
```

Prompt not for the original task intended by the dataset authors
Input Template:

```
Title: {{summary}}

===

Write an article with the given title:
```

Target Template:

```
{{document}}
```

### 1.11.3 MULTI_NEWS

Dataset from Fabbri et al. (2019). Used in training.

#### Data Example

| Key | Value |
|---|---|
| document | `National Archives`Yes, it's that time again, ... + |
| summary | `{ The unemployment rate dropped to 8.2% last month...` |

#### Prompts

Input Template:

```
{% set docs = document.split("3ed2dface8203c4c9dfb1a5dc58e41e0||") |
reject("equalto", "") | list %}
What are the key points across these news articles:
{% for doc in docs %}

Article: {{doc}}
{% endfor %}
```

Target Template:

```
{{summary[2:]}}
```

Input Template:

195

```
{% set docs = document.split("3ed2dface8203c4c9dfb1a5dc58e41e0||") |
reject("equalto", "") | list %}
Synthesize these documents into a single one:
{% for doc in docs %}

- {{doc}}
{% endfor %}
```

Target Template:

```
{{summary[2:]}}
```

Input Template:

```
{% set docs = document.split("3ed2dface8203c4c9dfb1a5dc58e41e0||") |
reject("equalto", "") | list %}
I want to edit the following articles into a more concise summary:
{% for doc in docs %}

Article: {{doc}}
{% endfor %}
```

Target Template:

```
{{summary[2:]}}
```

Input Template:

```
{% set docs = document.split("3ed2dface8203c4c9dfb1a5dc58e41e0||") |
reject("equalto", "") | list %}
Write a summary of the following articles:
{% for doc in docs %}

Document: {{doc}}
{% endfor %}
```

Target Template:

```
{{summary[2:]}}
```

Prompt not for the original task intended by the dataset authors
Input Template:

```
{% set docs = document.split("3ed2dface8203c4c9dfb1a5dc58e41e0||") |
reject("equalto", "") | list%}
Write an expanded news article with plausible details from the following
summary:
{{summary[2:]}}
```

Target Template:

```
{{docs | choice}}
```

Input Template:

```
{% set docs = document.split("3ed2dface8203c4c9dfb1a5dc58e41e0||") |
reject("equalto", "") | list %}
I'm trying to distill these articles down into one:
{% for doc in docs %}

Article: {{doc}}
{% endfor %}
```

Target Template:

```
{{summary[2:]}}
```

### 1.11.4 SAMSUM

Dataset from Gliwa et al. (2019). Used in training.

**Data Example**

| Key | Value |
|---|---|
| dialogue | Amanda: I baked  cookies. Do you want some?Jerry... + |
| id | 13818513 |
| summary | Amanda baked cookies and will bring Jerry some tom... |

**Prompts**

Input Template:

```
Summarize this dialogue: {{dialogue}}
```

Target Template:

```
{{summary}}
```

Input Template:

```
{{dialogue}}
Given the above dialogue, write a summary.
```

Target Template:

```
{{summary}}
```

Input Template:

```
Summarize: {{dialogue}}
```

Target Template:

```
{{summary}}
```

Input Template:

```
{{dialogue}}
To sum up this dialog:
```

Target Template:

```
{{summary}}
```

Input Template:

```
Generate a summary for this dialogue:
{{dialogue}}
```

Target Template:

```
{{summary}}
```

Prompt not for the original task intended by the dataset authors
Input Template:

```
Write a dialogue that matches this summary: {{summary}}
```

Target Template:

```
{{dialogue}}
```

Input Template:

```
Sum up the following dialogue:
{{dialogue}}
```

Target Template:

```
{{summary}}
```

### 1.11.5  XSUM

Dataset from Narayan et al. (2018). Used in evaluation.

**Data Example**

| Key | Value |
|---|---|
| document | `Recent reports have linked some France-based playe...` |
| id | `29750031` |
| summary | `New Welsh Rugby Union chairman Gareth Davies belie...` |

**Prompts**

Input Template:

```
{{document}}

===

Write a summary of the text above :
```

Target Template:

```
{{summary}}
```

Input Template:

```
Article: {{document}}

Summary:
```

Target Template:

```
{{summary}}
```

Prompt from Brockman (2020)
Input Template:

```
{{document}}
How would you rephrase that in a few words?
```

Target Template:

```
{{summary}}
```

Prompt from Brockman (2020)
Input Template:

```
My college roommate asked me what this article means:

{{document}}

So I recapped it in layman's terms:
```

Target Template:

```
{{summary}}
```

Prompt from Brockman (2020)
Input Template:

```
{{document}}
This boils down to the simple idea that
```

Target Template:

```
{{summary}}
```

Input Template:

```
Summarize: {{document}}
```

Target Template:

```
{{summary}}
```

Input Template:

```
Summarize this document: {{document}}
Summary:
```

Target Template:

```
{{summary}}
```

Input Template:

```
{{document}}

===

Given the above document, write one sentence to summarize:
```

Target Template:

```
{{summary}}
```

Input Template:

```
First, please read the article below.

{{document}}

Now, can you write me an extremely short abstract for it?
```

Target Template:

```
{{summary}}
```

Prompt from Radford et al. (2019)
Input Template:

```
{{document}}

TL;DR:
```

Target Template:

```
{{summary}}
```

## 1.12 TOPIC CLASSIFICATION

### 1.12.1 AG_NEWS

Dataset from Zhang et al. (2015b). Used in training.

**Data Example**

| Key | Value |
|-----|-------|
| text | Wall St. Bears Claw Back Into the Black (Reuters) ... |
| label | 2 |

**Prompts**

Input Template:

```
What label best describes this news article?
{{text}}
```

Target Template:

```
{{answer_choices[label] }}
```

Answer Choices Template:

```
World politics ||| Sports ||| Business ||| Science and technology
```

Input Template:

```
Is this a piece of news regarding {{"world politics, sports, business,
or science and technology"}}?
{{text}}
```

Target Template:

```
{{answer_choices[label] }}
```

Answer Choices Template:

```
World politics ||| Sports ||| Business ||| Science and technology
```

Input Template:

```
Would you recommend the following article to a {{"politician"}}, an
{{"athlete"}}, a {{"business executive"}}, or a {{"scientist"}}?

{{ text }}
```

Target Template:

```
{{answer_choices[label]}}
```

Answer Choices Template:

```
Politician ||| Athlete ||| Business executive ||| Scientist
```

Input Template:

```
{{text}}

Which of the following sections of a newspaper would this article likely
appear in? {{"World News"}}, {{"Sports"}}, {{"Business"}}, or {{"Science
and Technology"}}?
```

**Target Template:**

```
{{answer_choices[label] }}
```

**Answer Choices Template:**

```
World News ||| Sports ||| Business ||| Science and Technology
```

**Input Template:**

```
{{text}}

Which section of a newspaper would this article likely appear in?
```

**Target Template:**

```
{{answer_choices[label] }}
```

**Answer Choices Template:**

```
World News ||| Sports ||| Business ||| Science and Technology
```

**Input Template:**

```
{{text}}
Is this a piece of news regarding {{"world politics, sports, business,
or science and technology"}}?
```

**Target Template:**

```
{{answer_choices[label] }}
```

**Answer Choices Template:**

```
World politics ||| Sports ||| Business ||| Science and technology
```

**Input Template:**

```
{{text}}
What label best describes this news article?
```

Target Template:

```
{{answer_choices[label] }}
```

Answer Choices Template:

```
World politics ||| Sports ||| Business ||| Science and technology
```

---

### 1.12.2   DBPEDIA_14

Dataset from Lehmann et al. (2015). Used in training.

**Data Example**

| Key | Value |
|---|---|
| content | Abbott of Farnham E D Abbott Limited was a Britis... |
| label | 0 |
| title | E. D. Abbott Ltd |

**Prompts**

Input Template:

```
{{content}} Given a list of categories: {{"company, educational
institution, artist, athlete, office holder, mean of transportation,
building, natural place, village, animal, plant, album, film or written
work"}}, what category does the paragraph belong to?
```

Target Template:

```
{{ answer_choices[label] }}
```

Answer Choices Template:

```
Company ||| Educational Institution ||| Artist ||| Athlete ||| Office
Holder ||| Mean Of Transportation ||| Building ||| Natural Place |||
Village ||| Animal ||| Plant ||| Album ||| Film ||| Written Work
```

Input Template:

```
Pick one category for the following text. The options are - {{"company,
educational institution, artist, athlete, office holder, mean of
transportation, building, natural place, village, animal, plant, album,
film or written work"}}. {{title}} - {{content}}
```

Target Template:

```
{{ answer_choices[label] }}
```

Answer Choices Template:

```
Company ||| Educational Institution ||| Artist ||| Athlete ||| Office
Holder ||| Mean Of Transportation ||| Building ||| Natural Place |||
Village ||| Animal ||| Plant ||| Album ||| Film ||| Written Work
```

Input Template:

```
{{title}} - {{content}} Given a choice of categories {{"company,
educational institution, artist, athlete, office holder, mean of
transportation, building, natural place, village, animal, plant, album,
film or written work"}}, the text refers to which one?
```

Target Template:

```
{{ answer_choices[label] }}
```

Answer Choices Template:

```
Company ||| Educational Institution ||| Artist ||| Athlete ||| Office
Holder ||| Mean Of Transportation ||| Building ||| Natural Place |||
Village ||| Animal ||| Plant ||| Album ||| Film ||| Written Work
```

Input Template:

```
"{{title}}", given a list of categories: {{"company, educational
institution, artist, athlete, office holder, mean of transportation,
building, natural place, village, animal, plant, album, film or written
work"}}, what category does the title belong to?
```

Target Template:

```
{{ answer_choices[label] }}
```

Answer Choices Template:

```
Company ||| Educational Institution ||| Artist ||| Athlete ||| Office
Holder ||| Mean Of Transportation ||| Building ||| Natural Place |||
Village ||| Animal ||| Plant ||| Album ||| Film ||| Written Work
```

### 1.12.3 TREC

Dataset from Li and Roth (2002). Used in training.

**Data Example**

| Key | Value |
|---|---|
| label-coarse | 0 |
| label-fine | 0 |
| text | How did serfdom develop in and then leave Russia ? |

**Prompts**

Input Template:

```
Categories: {{', '.join(answer_choices)}}

What category best describes: {{text}}
Answer:
```

Target Template:

```
{{ answer_choices [label_coarse] }}
```

Answer Choices Template:

```
Description ||| Entity ||| Abbreviation ||| Person ||| Quantity |||
Location
```

Prompt not for the original task intended by the dataset authors
Input Template:

```
{% set label_mapping = {21:0, 18:1, 24:2, 11:3, 14:4} %}
{% if label_coarse == 5 %}
Is this question asking for {{', '.join(answer_choices)}}?
{{text}}
```

Target Template:

```
{{ answer_choices [label_mapping[label_fine]] }}
{% endif %}
```

Answer Choices Template:

```
city ||| country ||| mountain ||| state ||| other location
```

Prompt not for the original task intended by the dataset authors
Input Template:

```
{% set label_mapping = {39:0, 13:1, 8:2, 40:3, 25:4, 43:5, 27:6, 38:7,
35:8, 41:9, 32:10, 45:11, 14:12} %}
{% if label_coarse == 4 %}
{{text}}

Is this question asking for {{', '.join(answer_choices)}}?
```

Target Template:

```
{{ answer_choices [label_mapping[label_fine]] }}
{% endif %}
```

Answer Choices Template:

```
code ||| count ||| date ||| distance ||| price ||| order ||| period of
time ||| percentage ||| speed ||| temperature ||| size ||| weight |||
other number
```

Prompt not for the original task intended by the dataset authors
Input Template:

```
{% set label_mapping = {2:0, 22:1, 19:2, 1:3, 46:3, 23:4, 10:5, 17:6,
33:7, 37:8, 15:9, 30:10, 26:11, 16:12, 28:13, 42:14, 31:15, 20:16,
44:17, 36:18, 14:19} %}
{% if label_coarse == 1 %}
Is this question asking for {{', '.join(answer_choices)}}?
{{text}}
```

Target Template:

```
{{ answer_choices [label_mapping[label_fine]] }}
{% endif %}
```

Answer Choices Template:

```
an animal ||| an organ of the body ||| a color ||| creative piece |||
currency ||| disease or medicine ||| event ||| food ||| musical
instrument ||| language ||| letter ||| plant ||| product ||| religion
||| sport ||| substance ||| symbol ||| technique ||| term ||| vehicle
||| word ||| other entity
```

Prompt not for the original task intended by the dataset authors
Input Template:

```
{% set label_mapping = {39:0, 13:1, 8:2, 40:3, 25:4, 43:5, 27:6, 38:7,
35:8, 41:9, 32:10, 45:11, 14:12} %}
{% if label_coarse == 4 %}
Is this question asking for {{', '.join(answer_choices)}}?
{{text}}
```

Target Template:

```
{{ answer_choices [label_mapping[label_fine]] }}
{% endif %}
```

Answer Choices Template:

```
code ||| count ||| date ||| distance ||| price ||| order ||| period of
time ||| percentage ||| speed ||| temperature ||| size ||| weight |||
other number
```

Input Template:

```
Question: {{text}}

Descriptors: {{', '.join(answer_choices)}}

Best Descriptor?
```

Target Template:

```
{{answer_choices[label_coarse]}}
```

Answer Choices Template:

```
Description ||| Entity ||| Abbreviation ||| Person ||| Quantity |||
Location
```

Input Template:

```
{{text}}

What is this question asking for?
```

Target Template:

```
{{answer_choices[label_fine] }}
```

Answer Choices Template:

```
Manner ||| Creative Piece ||| Animal ||| Expression abbreviated |||
Individual ||| Group ||| Title ||| Defintion ||| Date ||| Reason |||
Event ||| State ||| Description ||| Count ||| Other ||| Letter |||
Religion ||| Food ||| Country ||| Color ||| Term ||| City ||| Organ of
the body ||| Disease or medicine ||| Mountain ||| Price ||| Product |||
Period ||| Substance ||| Sport ||| Plant ||| Technique ||| Size |||
Instrument ||| Abbreviation ||| Speed ||| Word ||| Language |||
Percentage ||| Code ||| Distance ||| Temperature ||| Symbol ||| Order
||| Vehicle ||| Weight ||| Currency
```

Prompt not for the original task intended by the dataset authors
Input Template:

```
{% set label_mapping = {21:0, 18:1, 24:2, 11:3, 14:4} %}
{% if label_coarse == 5 %}
{{text}}

Is this question asking for {{', '.join(answer_choices)}}?
```

Target Template:

```
{{ answer_choices [label_mapping[label_fine]] }}
{% endif %}
```

Answer Choices Template:

```
city ||| country ||| mountain ||| state ||| other location
```

Input Template:

```
Which category best describes the following question: {{text}}

Choose from the following list:
{{', '.join(answer_choices)}}
```

Target Template:

```
{{ answer_choices [label_coarse] }}
```

Answer Choices Template:

```
Description ||| Entity ||| Abbreviation ||| Person ||| Quantity |||
Location
```

Prompt not for the original task intended by the dataset authors
Input Template:

```
{% set label_mapping={0:2, 7:1,  12:0, 9:3} %}
{% if label_coarse == 0 %}
Is this question asking for {{', '.join(answer_choices)}}?
{{text}}
```

Target Template:

```
{{ answer_choices[label_mapping[label_fine]] }}
{% endif %}
```

Answer Choices Template:

```
definition ||| description ||| manner of action ||| reason
```

Input Template:

```
{{text}}

Is this asking about {{(', ').join(answer_choices)}}?
```

Target Template:

```
{{ answer_choices [label_coarse] }}
```

Answer Choices Template:

```
Description ||| Entity ||| Abbreviation ||| Person ||| Quantity |||
Location
```

Prompt not for the original task intended by the dataset authors
Input Template:

```
{% set label_mapping={34:0, 3:1} %}
{% if label_coarse == 2 %}
Is this question asking for an {{', '.join(answer_choices)}}?
{{text}}
```

Target Template:

```
{{answer_choices[label_mapping[label_fine]] }}
{% endif %}
```

Answer Choices Template:

```
abbreviation ||| expression abbreviated
```

Prompt not for the original task intended by the dataset authors
Input Template:

```
{% set label_mapping = {34:0, 3:1} %}
{% if label_coarse == 2 %}
{{text}}

Is this question asking for an {{', '.join(answer_choices)}}?
```

Target Template:

```
{{ answer_choices [label_mapping[label_fine]] }}
{% endif %}
```

Answer Choices Template:

```
abbreviation ||| expression abbreviated
```

Input Template:

```
Is the following question asking about {{', '.join(answer_choices)}}?

{{text}}
```

Target Template:

```
{{ answer_choices [label_coarse] }}
```

Answer Choices Template:

```
Description ||| Entity ||| Abbreviation ||| Person ||| Quantity |||
Location
```

---

Prompt not for the original task intended by the dataset authors
Input Template:

```
{% set label_mapping = {5:0, 4:1, 6:2, 12:3} %}
{% if label_coarse == 3 %}
Is this question asking for {{', '.join(answer_choices)}}?
{{text}}
```

Target Template:

```
{{ answer_choices[label_mapping[label_fine]] }}
{% endif %}
```

Answer Choices Template:

```
group ||| individual ||| title ||| description
```

---

Input Template:

```
What is this question asking for?

{{text}}
```

Target Template:

```
{{ answer_choices[label_fine] }}
```

Answer Choices Template:

```
Manner ||| Creative Piece ||| Animal ||| Expression abbreviated |||
Individual ||| Group ||| Title ||| Defintion ||| Date ||| Reason |||
Event ||| State ||| Description ||| Count ||| Other ||| Letter |||
Religion ||| Food ||| Country ||| Color ||| Term ||| City ||| Organ of
the body ||| Disease or medicine ||| Mountain ||| Price ||| Product |||
Period ||| Substance ||| Sport ||| Plant ||| Technique ||| Size |||
Instrument ||| Abbreviation ||| Speed ||| Word ||| Language |||
Percentage ||| Code ||| Distance ||| Temperature ||| Symbol ||| Order
||| Vehicle ||| Weight ||| Currency
```

Prompt not for the original task intended by the dataset authors
Input Template:

```
{% set label_mapping = {5:0, 4:1, 6:2, 12:3} %}
{% if label_coarse == 3 %}
{{text}}

Is this question asking for {{', '.join(answer_choices)}}?
```

Target Template:

```
{{ answer_choices [label_mapping[label_fine]] }}{% endif %}
```

Answer Choices Template:

```
group ||| individual ||| title ||| description
```

Prompt not for the original task intended by the dataset authors
Input Template:

```
{% set label_mapping={0:2, 7:1,  12:0, 9:3} %}
{% if label_coarse == 0 %}
{{text}}

Is this question asking for {{', '.join(answer_choices)}}?
```

Target Template:

```
{{ answer_choices [label_mapping[label_fine]] }}
{% endif %}
```

Answer Choices Template:

```
definition ||| description ||| manner of action ||| reason
```

## 1.13 WORD SENSE DISAMBIGUATION

### 1.13.1 SUPER_GLUE WIC

Dataset from Pilehvar and os'e Camacho-Collados (2018). Used in evaluation.

**Data Example**

**Prompts**

Input Template:

| Key | Value |
|---|---|
| end1 | 36 |
| end2 | 32 |
| idx | 0 |
| label | 0 |
| sentence1 | Do you want to come over to my place later? |
| sentence2 | A political system with no place for the less prom... |
| start1 | 31 |
| start2 | 27 |
| word | place |

```
Does the word "{{word}}" have the same meaning in these two sentences?
Yes, No?
{{sentence1}}
{{sentence2}}
```

Target Template:

```
{% if label != -1%}
{{answer_choices[label]}}
{% endif %}
```

Answer Choices Template:

```
No ||| Yes
```

Input Template:

```
Does the word "{{word}}" have the same meaning in these two sentences?
{{sentence1}}
{{sentence2}}
```

Target Template:

```
{% if label != -1%}
{{answer_choices[label]}}
{% endif %}
```

Answer Choices Template:

```
No ||| Yes
```

Input Template:

```
Homework

Decide whether the word "{{word}}" is used with the same meaning in the
two following sentences. Answer by yes or no.
{{sentence1}}
{{sentence2}}
```

Target Template:

```
{% if label != -1%}
{{answer_choices[label]}}
{% endif %}
```

Answer Choices Template:

```
No ||| Yes
```

Input Template:

```
Sentence A: {{sentence1}}
Sentence B: {{sentence2}}

"{{word}}" has a similar meaning in sentences A and B. True or False?
```

Target Template:

```
{% if label != -1%}
{{answer_choices[label]}}
{% endif %}
```

Answer Choices Template:

```
False ||| True
```

Prompt from Brown et al. (2020)
Input Template:

```
{{sentence1}}
{{sentence2}}
Question: Is the word '{{word}}' used in the same sense in the two
sentences above?
```

Target Template:

```
{% if label != -1%}
{{answer_choices[label]}}
{% endif %}
```

Answer Choices Template:

```
No ||| Yes
```

Input Template:

```
Sentence 1: {{sentence1}}
Sentence 2: {{sentence2}}

Determine whether the word "{{word}}" is used in the same sense in both
sentences. Yes or no?
```

Target Template:

```
{% if label != -1%}
{{answer_choices[label]}}
{% endif %}
```

Answer Choices Template:

```
No ||| Yes
```

Input Template:

```
Determine if the word '{{word}}' is used in the same way in the two
sentences below.
{{sentence1}}
{{sentence2}}
```

Target Template:

```
{% if label != -1%}
{{answer_choices[label]}}
{% endif %}
```

Answer Choices Template:

```
No ||| Yes
```

Prompt from Brown et al. (2020)
Input Template:

```
{{sentence1}}
{{sentence2}}
Question: Is the word '{{word}}' used in the same sense in the two
sentences above? Yes, No?
```

Target Template:

```
{% if label != -1%}
{{answer_choices[label]}}
{% endif %}
```

Answer Choices Template:

```
No ||| Yes
```

Input Template:

```
The word "{{word}}" has multiple meanings. Does it have the same meaning
in sentences 1 and 2? Yes or no?

Sentence 1: {{sentence1}}
Sentence 2: {{sentence2}}
```

Target Template:

```
{% if label != -1%}
{{answer_choices[label]}}
{% endif %}
```

Answer Choices Template:

```
No ||| Yes
```

Prompt from **?**
Input Template:

```
{{sentence1}}
{{sentence2}}
Similar sense of {{word}}?
```

Target Template:

```
{% if label != -1%}
{{answer_choices[label]}}
{% endif %}
```

Answer Choices Template:

```
No ||| Yes
```

