# OpenReview forum: "Multitask Prompted Training Enables Zero-Shot Task Generalization"
_ICLR.cc/2022/Conference — ICLR 2022 Spotlight_

### Official Review · Reviewer_n8Ae · 2021-11-01

**Correctness:** 3
**Technical Novelty And Significance:** 3
**Empirical Novelty And Significance:** 4
**Recommendation:** 8
**Confidence:** 4

**Main Review:**

Strengths:

1. This paper asks two very important questions about applying LMs, and since LMs are so widely used right now, I believe investigating these questions is super important to the community right now.
2. The authors ran a lot of experiments, and while I did point out some missing potential experiments in the weaknesses section, I would like to state that overall the set of experiments ran by the authors here is very strong.
3. The paper is well-written and easy to follow.
4. The authors shine a light on the importance of creative and diverse prompts, some of which may seem too verbose or even silly at first. I really like this direction and hope this paper leads to this research direction being further explored!

Weaknesses:

1. I feel like your main baseline is the T5+LM model (T5 finetuned on the causal LMing task). Therefore I think it would make the paper much stronger if you were to show how that baseline performs on the tasks in Figure 5.  I understand that the baselines that you use in Figure 5 were built and trained by an external third party, so I trust that these baselines aren’t too weak or unoptimzed somehow, but it would still be very important to show that you also improve over T5 in this setting.  It would also be very interesting to see how well GPT-3 performs on those tasks, but I did not and will not penalize you for not having those numbers, as the GPT-3 API is not fully open and free.
2. This paper seems like an extension of ideas discussed in iPET/PET (“It’s Not Just Size That Matters: Small Language Models Are Also Few-Shot Learners”, Schick & Schutze), and so to me it feels like it would be quite important to compare to that model in at least some of the experiments that you run, for example, if Figure 4 compared to iPET I feel the paper would be much stronger. What do you think? Are T0 and iPET comparable?

Questions/comments:

1. Could you explain and/or cite relevant source “GPT-3 is behaviorally capable of zero-shot generalization to new tasks” (section 2) in the next draft?. I know that later on in the paper you do talk more about this, but I think it might also be important to explain this claim in section 2. I think that many people believe that while GPT-3 wasn’t explicitly trained on tasks such as translation or adding numbers,  GPT-3 can only do well on “zero-shot” tasks that it has implicitly observed during training
2. I’m using the chrome PDF reader and do not see an actual figure above the caption for figure 3.
3. “Unlike decoder-only language models such as GPT-3, is not trained to generate the input. This has computational benefits since in our prompted data, the target typically is much shorter than the input”. You might not need to generate the input but you still need to encode it and so you only save the time taken on the softmax layer which is insignificant. In addition, since you use an enc-dec model you need those extra encoder-attention sublayers so it’s not so clear that there’s an efficiency advantage here. I would like to ask you to consider adding some actual experiments that show that this claim is correct, or cite a relevant paper, or just remove that sentence.
4. “Baseline models are standard Transformer-based language models with varying size.” (Figure 5). What is a ‘standard transformer LM’? I don’t think such a thing really exists. I think it could be helpful in the paper if you would give more details, such as: position embedding, FF dimension, number of heads, head dimension, and number of layers.
5. Make sure Figure 5 is explicitly referenced in the text in the relevant paragraph.
6. Consider making the following statement slightly weaker: “These results suggest that T0 is more robust to prompt formulation than GPT-3”. It’s hard to draw such a strong conclusion from an experiment on one dataset.
7. In figure 6, what is the meaning of each dot? Is it the performance for a given prompt? If so, that should be explained in the caption.


**Summary Of The Paper:**

Today, most NLP applications use language models. These language models can be split into two different categories. The first is the GPT-3-style models, that are only trained on the unsupervised language modeling objective. In the second category, we have the models that use the finetuning method that was popularized by ELMO and BERT, where the model is first trained on an unsupervised objective and is then trained on supervised tasks.

This paper asks two questions about models in the second category. Specifically, the authors take the T5 model and test if it improves performance on unseen tasks when finetuned on multiple other tasks. The second question is whether training on more prompts improves performance.

The authors show that their model does indeed improve performance on multiple held-out datasets, when doing multitask training. In addition, the authors show that the median performance improves when training on multiple different prompts (and testing on unseen prompts).

**Summary Of The Review:**

The authors ask two very interesting research questions with regards to prompting and finetuning of large LMs. A wide array of experiments answers these questions, and I think this will lead to better usage of LMs in the future. I did point out that there are some baselines that the authors missed, but I don’t think that that’s a big deal. I recommend that this paper be accepted, and feel that it will be very valuable to the NLP community.

---

> ### Author Response · Authors · 2021-11-17
> **Answer to reviewer 4 - n8Ae**
>
> Thank you for your review! To address the weaknesses you point out: We regret the omission of the T5+LM baseline for Big Bench (Figure 5) and we have updated it in the new draft. As for the comparison to PET, although this is certainly scientifically valuable, the scope of our work is strictly about zero-shot, whereas PET is most effective in the few-shot setting. Further, it is not clear how to adapt our prompts for the encoder-only model of PET.
>
> To address your questions:
>
> > 1. Could you explain and/or cite relevant source “GPT-3 is behaviorally capable of zero-shot generalization to new tasks” (section 2) in the next draft? I know that later on in the paper you do talk more about this, but I think it might also be important to explain this claim in section 2. I think that many people believe that while GPT-3 wasn’t explicitly trained on tasks such as translation or adding numbers, GPT-3 can only do well on “zero-shot” tasks that it has implicitly observed during training
>
> We believe the baselines provided by the BIG-bench authors are a good data point suggesting that extremely large language models are capable of performing tasks not explicitly seen in training. For instance, on Novel Concepts, the tasks of abstracting and manipulating new words, a 68B parameters model gets a performance significantly higher than random chance (28 vs 18 multiple choice grade). Additional evidence of GPT-3’s zero-shot ability can be found in Figure 4.2 of “Language Models are Few Shot Learners.” we can provide a reference to [Brown et al., 2020](https://arxiv.org/abs/2005.14165) after this sentence.
>
> > 2. I’m using the chrome PDF reader and do not see an actual figure above the caption for figure 3.
>
> Thank you for being so charitable to us! It was just a LaTeX hiccup on our side. Sorry! We fixed it in the updated draft.
>
> > 3. “Unlike decoder-only language models such as GPT-3, T0 is not trained to generate the input. This has computational benefits since in our prompted data, the target typically is much shorter than the input”. You might not need to generate the input but you still need to encode it and so you only save the time taken on the softmax layer which is insignificant. In addition, since you use an enc-dec model you need those extra encoder-attention sublayers so it’s not so clear that there’s an efficiency advantage here. I would like to ask you to consider adding some actual experiments that show that this claim is correct, or cite a relevant paper, or just remove that sentence.
>
> Thanks for pointing this out. We've removed the sentence and provided further clarification of the difference in computational cost.
>
> > 4. “Baseline models are standard Transformer-based language models with varying size.” (Figure 5). What is a ‘standard transformer LM’?
>
> Unfortunately, the BIG-Bench maintainers were unable to share additional details about these models with us since they are not public. We assume they very closely match the original Transformer architecture, with the possible modification of placing the layer normalization outside of the residual path (which has become standard practice).
>
> > 5. Make sure Figure 5 is explicitly referenced in the text in the relevant paragraph.
>
> Fixed. Thank you!
>
> > 6. Consider making the following statement slightly weaker: “These results suggest that T0 is more robust to prompt formulation than GPT-3”. It’s hard to draw such a strong conclusion from an experiment on one dataset.
>
> This is fair. We have updated our writing.
>
> > 7. In figure 6, what is the meaning of each dot? Is it the performance for a given prompt? If so, that should be explained in the caption.
>
> Yes, each dot is the performance of a given prompt. Sorry for being unclear and we have updated the caption.

---

> > ### Comment · Reviewer_n8Ae · 2021-11-21
> > **response**
> >
> > Thank you! This answers my questions. I believe that this is a strong paper that should be accepted!

---

### Official Review · Reviewer_qyUt · 2021-11-01

**Correctness:** 3
**Technical Novelty And Significance:** 1
**Empirical Novelty And Significance:** 2
**Recommendation:** 3
**Confidence:** 5

**Main Review:**

There are some major comments which are presented here,

1- why MNLI and QNLI are not considered in NLI tasks?

2- Why yellow-colored tasks in figure 2 are not considered as unseen tasks too? In other words, why different held-out sets are not considered for unseen tasks?

3- The authors mentions that humans are not explicitly trained on NLI tasks, story completion, coreference resolution, or word sense disambiguation, and for that reason, these tasks are selected as held-out to test zero-shot generalization. For the same reason, question answering or paraphrasing can be selected, since humans are not explicitly trained on these tasks!

4- Section 5: the author mentioned training encoder-decoder model is computationally efficient since it is  trained to only generate the target as opposed to GPT3. However, isn't the  model use more supervision to generate the input as well, which can benefit the zero-shot performance? later, the author mentioned that since T5 is pretrained with masked language modeling, which is different than conditional text generation, they used T5LM model, the language model adapted T5. Isn't this contradict their previous statement about benefits of target-only generation?

5- how about T5 performance with different sizes?

6- it is mentioned that the best checkpoint was selected from step 12200 since it yields best validation metric. However, the training set of T0, T0+ and T0++ are different. are they using the same validation with same tasks?

7- Figure 4: why T0LM is not trained on the training mixture dataset? it is mostly underperforms T0

8- author mentions that they did not do prompt selection using best evaluation performance. Since zero-shot tasks are different than training, don't they have a completely different prompt template?

9- Since T0 model is the finetuned T5 on multi-tasked prompted dataset, I think the author should also add a finetuned encoder-decoder model which is not using T5 pretrained weights as initialization. This will helps to understand the benefits of encoder-decoder training for zero-shot learning. This way, the T0 can be evaluated on more BigBench tasks which has not in-vocabulary T5 tokens. Moreover, using pretrained T5 model for T0 means using implicit multi-tasking trained during T5 pretraining.

10- it would be interesting to see the T0 performance which trained with different number of mixed training tasks. In other words, what is the impact of each training task, to zero-shot learning

11- T0 outperforms 7 out of 11 zero-shot datasets, but it outperforms on 2 out of 4 zero-shot tasks. It underperforms on all datasets of  story completion and coreference resolution. what is the explanation for this?

12- what is the purpose of evaluating T5LM without training on multi-tasked training dataset in Figure 4 ? it underpreforms Gpt3 (6.7B) on all tasks except WiC!

13- the contribution is not clear. if it is about the benefit of encoder-decoder  model to decoder-only LM models for zero-shot learning, then a decoder-only LM model should be trained on the multi-tasked prompted training set. if the contribution is about the proposed multi-tasked prompted dataset

14- Figure 5: there is no constant improvement between T0, to T0+ and T0++. In other words, adding more dataset to multi-tasked trained does not always improve zero-shot learning on BigBench. what is the justification for this behaviour of encoder-decoder model? if using more training data sometimes reduce performance, e.g. Known Unknown and Logical Deduction tasks, perhaps due to model-scale constraint, it would be helpful to evaluate with a larger T0. In other words, what is the relation between T0 model scale and training size to the zero-shot performance

15- Section 6.2: the author compared T0 with GPT3 on RTE tasks with different prompts. what is the reason for just choosing RTE for this comparison only? shouldn't this comparison be done on all zero-shot tasks of Figure 4, to evaluate the robustness of GPT3 to prompts as well?

16- The performance of T0 can also be compared with FLAN model in Figure 4 for each zero-shot task, despite different training set.

17- Section 7: the author mentions a key difference between T0 and FLAN model is that T0 is an encoder-decoder model which is pretrained and finetuned with different objectives. However, no clear comparison with FLAN model is presented in the paper

18- Section 7: author mentioned that FLAN model with comparable size to T0 (8B) finds that increasing multi-task training reduce zero-shot performance, whereas they find the opposite. The only evaluation on the size of multi-task training dataset is presented on BigBench dataset in Figure 5, which is different than FLAN zero-shot evaluation.

**Summary Of The Paper:**

This paper proposed a new method for zero-shot generalization of NLP models. Since zero-shot generalization of the unsupervised pretrained language models are understood as an ability which is captured **implicitly** during unsupervised training on large natural text, this paper suggested a supervised learning method to understand zero-shot generalization through **explicit multi-task learning** in NLP models. To facilitate the multi-task supervised learning, each task is converted to multiple prompts, and an encoder-decoder model is trained (T5 model), to generate the answer for each tasks. The paper also provided a collection of crowdsourced prompt formats for each dataset, called P3.
The results indicate that a model which is trained with multiple prompts per dataset is less sensitive to the wording choice of prompts. The results also show that using encoder-decoder model which is trained to only generates the target, is more computationally efficient than standard language modeling training. It is also shown that the model is ~16x smaller than GPT3 and can attain similar performance in zero-shot learning.

**Summary Of The Review:**

This paper presents a new method for NLP zero-shot learning using supervised multi-task prompted training. They proposed to use encoder-decoder model which is already pretrained using masked language modeling to attain better zero-shot performance with smaller model compared to large-scale autoregressive language model (GPT3). However, there are some shortcomings,

1- The zero-shot evaluations are based on some selected tasks, including NLI, story completion, word sense disambiguation, and coreference resolution. In this setting, other tasks such as paraphrasing, QA, sentiment analysis, summarization, topic classification are not considered as zero-shot tasks, and only used for training, which makes an incomplete evaluation for zero-shot learning.

2- The author mentioned the benefits of encoder-decoder pretraining objective (target-only generation) to auto-regressive objective (input generation) as a key factor. However, there are no comparison with auto-regressive model with the same pretraining and zero-shot task evaluation. The only comparison is on the BigBench dataset, where Transformer-based language models are pretrained with different dataset, and also performance comparison on NLI, story completion, word sense disambiguation, and coreference resolution.

3- The paper is centered around understanding the multi-task pretraining, using explicit supervised learning of model. However, the evaluated model (T0) is using pretrained weights of T5 for training, which implicitly learned multi-tasking during unsupervised pretraining of T5 model. This makes the evaluation biased and an unfair comparison to GPT3 model.

---

> ### Author Response · Authors · 2021-11-17
> **Answer to reviewer 3 - qyUt (1/N)**
>
> Thank you for your extensive review! First, to address the general concerns raised in your summary of the review:
>
> > 1. The zero-shot evaluations are based on some selected tasks, including NLI, story completion, word sense disambiguation, and coreference resolution. In this setting, other tasks such as paraphrasing, QA, sentiment analysis, summarization, topic classification are not considered as zero-shot tasks, and only used for training, which makes an incomplete evaluation for zero-shot learning.
>
> We agree that additional evaluations on other tasks would be more ideal, but because we focus on task-level (not just dataset-level) generalizations, evaluation on more tasks would require holding them out and training a separate model, which is beyond our computational budget. Additionally, we chose NLI as a held-out task because humans also zero-shot generalize to NLI as an unseen task: most humans are never explicitly trained to classify whether a premise sentence entails or contradicts a hypothesis sentence, yet they ﬁnd it intuitive to perform this task without training ([Williams et al., 2020](https://arxiv.org/abs/2010.12729)). For the same reason, we also hold out coreference resolution and word sense disambiguation. We further hold out story completion because it is a task possibly too similar to NLI (Appendix C2 discusses this in detail). In contrast, sentiment and topic classification are now considered as relatively easy tasks in NLP, whereas QA as the literature defines it can be too broad and ambiguous (Appendix C1 discusses this in detail).
>
> > 2. The author mentioned the benefits of encoder-decoder pretraining objective (target-only generation) to auto-regressive objective (input generation) as a key factor. However, there are no comparison with auto-regressive model with the same pretraining and zero-shot task evaluation.
>
> [Related to Reviewer 4’s Question 3] Our main research question was not to distinguish between the autoregressive vs. encoder-decoder models. We selected encoder-decoder because it seems like a natural model for input target prompted multi-task learning. However, given the concurrent work by [Wei et al. 2021](https://arxiv.org/abs/2109.01652), we mention it as an additional variable. All of our comparisons are to both GPT-3 and to a control T5-LM.
>
> > 3.  The paper is centered around understanding the multi-task pretraining, using explicit supervised learning of model. However, the evaluated model (T0) is using pretrained weights of T5 for training, which implicitly learned multi-tasking during unsupervised pretraining of T5 model. This makes the evaluation biased and an unfair comparison to GPT-3 model.
>
> [Same as main review Q9 and Reviewer 1’s Question 2] Note that, as detailed in Section 1, our research question is precisely to compare the implicit multitask training in large LMs’ pretraining vs. explicit multitask prompted training with smaller models. We show that the latter is a more efficient method (as opposed to merely scaling to an ever larger model) to achieve better generalization to unseen tasks. For this research question, we wouldn’t consider it unfair as neither T0 nor GPT-3 was trained on any of the held-out tasks. Meanwhile, although it is true that T5 also went through implicit multitask training during its pretraining, we believe it is unproductive to devise a study that uses an un-pretrained transformer, especially given that prior work by [Raffel et al. 2020](https://arxiv.org/abs/1910.10683) shows that doing multitask training (without any unsupervised training) over a large task mixture results in a significant performance drop compared to unsupervised pretraining.

---

> > ### Author Response · Authors · 2021-11-17
> > **Answer to reviewer 3 - qyUt (2/N)**
> >
> > To address the specific questions raised in your main review:
> > > 1.  “Why MNLI and QNLI are not considered in NLI tasks?”
> >
> > We do consider them as NLI and thus held out from training, but they’re not in our eval mixture because they have no GPT-3 or FLAN baselines for comparison.
> >
> > > 2. Why yellow-colored tasks in figure 2 are not considered as unseen tasks too? In other words, why different held-out sets are not considered for unseen tasks?
> >
> > We used different colors to indicate training and held-out datasets: yellow for training and green for held-out. We have updated Figure 2 so that there is no confusion. Yellow boxes indicate the datasets in T0’s training mixture. The details of each model’s training and evaluation mixture are presented in Table 5.
> >
> > > 3. The authors mention that humans are not explicitly trained on NLI tasks, story completion, coreference resolution, or word sense disambiguation, and for that reason, these tasks are selected as held-out to test zero-shot generalization. For the same reason, question answering or paraphrasing can be selected, since humans are not explicitly trained on these tasks!
> >
> > We agree that paraphrasing could have also been a good target for generalization, but QA is too broad for that. Humans who receive formal education are trained on plenty of QA, but we do not expect them to zero-shot perform well on the science questions in ARC, the trivia questions in Natural Questions, or the US social norms in Social IQA, etc.
> >
> > > 4. Section 5: the author mentioned training encoder-decoder model is computationally efficient since it is trained to only generate the target as opposed to GPT3. However, isn't the model use more supervision to generate the input as well, which can benefit the zero-shot performance? later, the author mentioned that since T5 is pretrained with masked language modeling, which is different than conditional text generation, they used T5LM model, the language model adapted T5. Isn't this contradict their previous statement about benefits of target-only generation?
> >
> > Since GPT-3 is trained to predict every token (including both the context and the output/target), it receives a training signal for all tokens. In contrast, T0 is only trained to predict the target tokens. T5+LM is also only trained to predict the "target" tokens (i.e. decoder outputs). We have updated the wording in our paper to make this more clear.
> >
> > > 5. How about T5 performance with different sizes?
> >
> > We include T0's performance both for the 11B parameter and 3B parameter sizes (Appendix F Figure 8).
> >
> > > 6. it is mentioned that the best checkpoint was selected from step 12200 since it yields best validation metric. However, the training set of T0, T0+ and T0++ are different. are they using the same validation with same tasks?
> >
> > We performed checkpoint selection for only one model (T0) using the training splits of its training datasets. We found that 12’200 steps yield the highest performance on the training splits and for simplicity, we transferred this hyper-parameter to all the other models (in particular T0+, T0++). If we had the computational budget (and time budget), we would perform the same checkpoint selection for the other models.
> >
> > > 7. Figure 4: why T0LM is not trained on the training mixture dataset? it is mostly underperforms T0
> >
> > T5+LM isn't a proposed model. It is a comparison baseline as a non-multitask trained model.
> >
> > > 8. “author mentions that they did not do prompt selection using best evaluation performance. Since zero-shot tasks are different than training, don't they have a completely different prompt template?”
> >
> > That’s correct, held-out datasets have different prompts than training datasets. Furthermore, we did not use the other splits of the held-out datasets to select the best performing prompt (for instance evaluating on the validation split to report performance on the test split, or the train split for the validation split) but instead reported the median performance across prompts along with the interquartile. That way, we ensure that no information is leaked to the held-out datasets.

---

> > > ### Author Response · Authors · 2021-11-17
> > > **Answer to reviewer 3 - qyUt (3/N)**
> > >
> > > > 9. “Since T0 model is the finetuned T5 on multi-tasked prompted dataset, I think the author should also add a finetuned encoder-decoder model which is not using T5 pretrained weights as initialization. This will helps to understand the benefits of encoder-decoder training for zero-shot learning. This way, the T0 can be evaluated on more BigBench tasks which has not in-vocabulary T5 tokens. Moreover, using pretrained T5 model for T0 means using implicit multi-tasking trained during T5 pretraining.”
> > >
> > > We agree that including a baseline where we train a model with the same architecture/size on our multitask mixture only would be interesting. However, prior work by [Raffel et al. 2020](https://arxiv.org/abs/1910.10683) shows that doing multitask training (without any unsupervised training) over a large task mixture results in a significant performance drop compared to unsupervised pretraining. Furthermore, to do an equivalent amount of training to an equivalently-sized model would cost around $1M on Google Cloud TPU devices, and is therefore cost-prohibitive. However, we agree that this would be a very interesting avenue for future work.
> > >
> > > > 10. it would be interesting to see the T0 performance which trained with different number of mixed training tasks. In other words, what is the impact of each training task, to zero-shot learning
> > >
> > > We agree that measuring the performance as the number of training tasks varies is interesting. Our study contains one experiment (Sec 6.2) along these lines: T0++ is trained on more tasks than T0 since T0++ includes sentence completion, coreference resolution, and word sense disambiguation. Unsurprisingly, T0++'s performance is often best on Big Bench. We would be interested in running additional experiments ablating the impact of more training tasks, but lack the computational resources. We, therefore, encourage future work to explore this effect.
> > >
> > > > 11. T0 outperforms 7 out of 11 zero-shot datasets, but it outperforms on 2 out of 4 zero-shot tasks. It underperforms on all datasets of story completion and coreference resolution. what is the explanation for this?
> > >
> > > For story completion, T0 outperforms GPT-3 on StoryCloze and COPA. In the updated Appendix C3, we elaborate on the difficulty in replicating GPT-3’s evaluation setup for LAMBDA. As for HellaSwag and coreference, the authors of FLAN propose an interesting conjecture that prompts or instructions are largely redundant for tasks that are directly formulated as language modeling, e.g., coreference formatted as ﬁnishing an incomplete sentence. In fact, both FLAN and T0 underperform GPT-3 in those task categories, and we leave a full investigation of this conjecture to future work.
> > >
> > > > 12. what is the purpose of evaluating T5LM without training on multi-tasked training dataset in Figure 4 ? it underpreforms Gpt3 (6.7B) on all tasks except WiC!
> > >
> > > Please see our answer to Q7.
> > >
> > > > 13. the contribution is not clear. if it is about the benefit of encoder-decoder model to decoder-only LM models for zero-shot learning, then a decoder-only LM model should be trained on the multi-tasked prompted training set. if the contribution is about the proposed multi-tasked prompted dataset
> > >
> > > Our core hypothesis in this work is that massively multitask prompted fine-tuning enables strong zero-shot task generalization. We verified this hypothesis by prompting a large and diverse set of NLP datasets, fine-tuned a pretrained model on this mixture and observed strong zero-shot task generalization, and significant improvements over baselines that do “implicit” multitask learning (for instance from 51.81 for T5+LM to 81.23 for T0 on RTE).
> > >
> > > The second research question we ask is whether training on a wider range of prompts improves robustness to the wording of the prompts. We find that training on more prompts per dataset consistently improves the median and decreases the variability of performance on held-out tasks. Training on prompts from a wider range of datasets also generally improves the median but does not decrease the variability.
> > > While we used an encoder-decoder model in our experiments, we believe that our key insights transfer to other architectures (decoder-only for instance) as highlighted by concurrent work ([Wei et al, 2021](https://arxiv.org/abs/2109.01652)). We leave exhaustive comparisons between model architectures from the perspective of prompted fine-tuning to future work.

---

> > > > ### Author Response · Authors · 2021-11-17
> > > > **Answer to reviewer 3 - qyUt (4/4)**
> > > >
> > > > > 14. Figure 5: there is no constant improvement between T0, to T0+ and T0++. In other words, adding more dataset to multi-tasked trained does not always improve zero-shot learning on BigBench. what is the justification for this behaviour of encoder-decoder model? if using more training data sometimes reduce performance, e.g. Known Unknown and Logical Deduction tasks, perhaps due to model-scale constraint, it would be helpful to evaluate with a larger T0. In other words, what is the relation between T0 model scale and training size to the zero-shot performance
> > > >
> > > > As we increase the number of datasets in the training mixture, the median performance increases in most of the cases (Figure 5 and 7) though there are exceptions. We hypothesize that this is not a behavior specific to encoder-decoder architectures but answering this question requires proper experimentation that goes beyond the scope of our study. We agree that the model scale would be one of the key factors to examine (is the model capacity saturating as we add more datasets?).
> > > >
> > > > > 15. “Section 6.2: the author compared T0 with GPT3 on RTE tasks with different prompts. what is the reason for just choosing RTE for this comparison only? shouldn't this comparison be done on all zero-shot tasks of Figure 4, to evaluate the robustness of GPT3 to prompts as well?”
> > > >
> > > > We agree that proper comparison would require evaluating GPT3’s robustness on all zero-shot tasks of Figure 4. However, the cost of OpenAI's API would be quickly prohibitive. We hope that better-funded organizations will be able to report these numbers as valuable data points for the community.
> > > >
> > > > > 16. The performance of T0 can also be compared with FLAN model in Figure 4 for each zero-shot task, despite different training set.
> > > >
> > > > We have added the following comparison to Section 7: _”Compared to FLAN, T0’s zero-shot performance is better on CB and RTE, similar on Story Cloze and COPA, and worse on Winogrande, ANLI, and HellaSwag. T0++ outperforms FLAN on CB, RTE, and COPA and matches FLAN’s performance on Winogrande and ANLI. T0 and T0++ attain  this performance  despite being over 10x smaller  than  FLAN  (137B vs. 11B  parameters).”_ We do not include these comparisons in the results section because FLAN is concurrent work.
> > > >
> > > > > 17. Section 7: the author mentions a key difference between T0 and FLAN model is that T0 is an encoder-decoder model which is pretrained and finetuned with different objectives. However, no clear comparison with FLAN model is presented in the paper
> > > >
> > > > FLAN is concurrent work, appearing on ArXiv on Sept. 3. The ICLR Reviewer Guide notes that _“We consider papers contemporaneous if they are published (available in online proceedings) within the last four months. That means, since our full paper deadline is October 5, if a paper was published (i.e., at a peer-reviewed venue) on or after June 5, 2021, authors are not required to compare their own work to that paper.”_ Regardless, we discussed FLAN and its differences in Section 7 and Appendix D because it provides complementary results to our own.
> > > >
> > > > > 18. Section 7: author mentioned that FLAN model with comparable size to T0 (8B) finds that increasing multi-task training reduce zero-shot performance, whereas they find the opposite. The only evaluation on the size of multi-task training dataset is presented on BigBench dataset in Figure 5, which is different than FLAN zero-shot evaluation.
> > > >
> > > > Same as your Q5, please see T0 3B’s evaluation at Appendix F Figure 8.

---

> > > > > ### Comment · Reviewer_qyUt · 2021-11-22
> > > > > **Answer to Author**
> > > > >
> > > > > Thanks for providing responses. I have few more questions though,
> > > > >
> > > > > 1- Regarding your comment, Flan did the MNLI and QNLI, check page 23 of their paper
> > > > >
> > > > > 2- author mentions that multi-task training fails without unsupervised pertaining. Then what is the benefit of explicit mulit-task compared to implicit unsupervised pretraining.
> > > > >
> > > > > 3- if encoder-decoder T5 model performance drops a lot when pretraining using multi-task dataset, isn’t this the limitation of the chosen architecture and pretraining cost function for implicit multi-tasking? Especially given the pretraining cost of encoder-decoder model training
> > > > >
> > > > > 4- it might be intuitive that when training on prompt from a more diverse collections of datasets will results in higher variance. what about training on wider range of dataset, but the overall proportion of #prompt/#datast is kept fixed?
> > > > >
> > > > > 5- Regarding Q-18: is there a comparison on T0(3B) vs T0+(3B) and T0++(3B) models for zero-shot performance? Similar to Figure 7, since it is a small model!

---

> > > > > > ### Author Response · Authors · 2021-11-23
> > > > > > **Answers to follow up questions**
> > > > > >
> > > > > > Thank you for your follow up questions!
> > > > > >
> > > > > > 1) Thank you for pointing out these additional results. They were not in the original ArXiv version of FLAN, and included in the version posted the day of the ICLR deadline. Again, we emphasize that FLAN is concurrent work also under review at ICLR.
> > > > > >
> > > > > > 2) There are three scenarios:
> > > > > >   - a) Pre-train a model on a causal or masked language modeling task (or a mixture of both), then evaluate in a zero-shot setting.
> > > > > >   - b) Pre-train a model as in scenario (a), then fine tune on multiple supervised tasks, and finally evaluate in a zero-shot setting.
> > > > > >   - c) Train a model with randomly initialized weights on multiple supervised tasks, then evaluate in a zero-shot setting.
> > > > > >
> > > > > > Scenario (a) is current practice, and scenario (b) is the one we study. We took your original question to be asking about scenario (c), which does not work because there isn’t enough data to train a large language model. To answer your follow up question, Figures 4 and 5 highlight the benefits of explicit multitask training. The lightest green line (T5-LM) is an instance of scenario (a). The darker green lines (T0 and its variants) are instances of scenario (b), using the same pre-trained model. We find that T0 and its variants generally do much better on zero-shot tasks, which shows the benefit of explicit multitask fine-tuning.
> > > > > >
> > > > > > 3) Not necessarily. In scenario (c) above, it is likely a lack of supervised data. The available data for language modeling is orders of magnitude larger than the prompted datasets we create, so there is still a benefit from pre-training on language modeling then fine-tuning on multiple supervised tasks.
> > > > > >
> > > > > > 4) What you suggest is exactly what we show in Figure 7. We use all available prompts of every dataset, which fixes the ratio of prompts/dataset to be an average of 8.03, and we measure the effect of prompts from a wider range of datasets by increasing the number of datasets from 39 to 49 to 55 (T0, T0+, T0++, respectively).
> > > > > >
> > > > > > 5) We agree that evaluating the 3B architecture on multiple training mixtures is interesting. Our compute budget did not allow for this in our initial experiments, since even a 3B parameter architecture can be costly to train.

---

### Official Review · Reviewer_Ju6C · 2021-11-02

**Correctness:** 3
**Technical Novelty And Significance:** 3
**Empirical Novelty And Significance:** 3
**Recommendation:** 6
**Confidence:** 4

**Main Review:**

Large pre-trained language models demonstrate state-of-the-art performance with large amounts on labeled training data. This has inspired a line of research to demonstrate generalizability of the models when trained with few labeled training samples. Notably there is a huge performance gap between fully supervised SOTA models like BERT, and few-shot and zero-shot learning with models like GPT-3. In this work, the authors aim to use labeled training data from a large number of tasks to demonstrate the improvement of model performance on unseen tasks. To this end, the authors leverage prompting to convert the format of all the tasks and their labels to an unified format for multi-task learning.

The idea of converting all the tasks to an unified format is not new, and has been explored in earlier works like DecaNLP, UFO-Entail, T5 and more recently in the CLUES few-shot benchmark. One of the primary contributions of the work is in aggregating resources across 12 tasks, 54 datasets and a large number of prompts for converting these tasks to the unified format. This is a quite valuable resource for the community.

Now, coming back to the experiments, the authors partition the task types into training and hold-out a subset of the tasks, including all the datasets therein, for evaluation. The authors perform an important study to verify that data for those tasks is not leaked through the pretraining corpus which addresses memorization concerns. However, this analysis is missing for the training tasks. Since the objective of this work, is to evaluate "true" zero-shot performance of the models, it is necessary to perform this analysis to ensure that any text for the "unseen" tasks and datasets is not leaked from the training corpus. This is important since it is often difficult to explicitly partition tasks into disjoint sets. For instance, the authors evaluate their models on held-out NLI tasks, but train their models on paraphrase identification tasks that bear some similarity with the entailment tasks. Can the authors report the multi-task model performance on NLI tasks without using the paraphrase identification tasks?

The authors compare the performance of their model and demonstrate the multi-task version (T0) to perform better than the single-task version (T5-LM) and outperform GPT-3 in primarily NLI tasks. The performance in WiC seems a bit strange given that random choice should give an accuracy of 50%.

Finally, some analysis on transferability of the tasks, to study the impact of different source tasks on target tasks, will be an interesting contribution for the multi-task setting.


**Summary Of The Paper:**

The authors demonstrate that massive multi-task learning with prompting can improve generalizability of large language models for zero-shot inference on unseen tasks.


**Summary Of The Review:**

The authors demonstrate that massive multi-task learning with prompting can improve generalizability of large language models for zero-shot inference on unseen tasks. The authors develop a very useful resource for the community aggregating prompts across several tasks and datasets. The claim of "true" zero-shot generalization needs to be further supported by two more experiments as highlighted in the main review, namely, (i) analyzing the overlap in text from the MTL setup with that of the unseen tasks, and (ii) ablating paraphrase identification tasks to find the impact on NLI tasks.

---

> ### Author Response · Authors · 2021-11-17
> **Answer to reviewer 2 - Ju6C**
>
> Thank you for your review! To address your specific concerns:
>
> > 1. “The authors perform an important study to verify that data for those tasks is not leaked through the pretraining corpus which addresses memorization concerns. However, this analysis is missing for the training tasks.”
>
>
> We agree that our contamination analysis is incomplete. We will update it with the training set as soon as time and compute permit.
>
> > 2. “The authors evaluate their models on held-out NLI tasks, but train their models on paraphrase identification tasks that bear some similarity with the entailment tasks.”
>
>
> While we agree that paraphrase identification and entailment have some high-level similarities, it has been shown e.g. by [Pruksachatkun et al. 2020](https://arxiv.org/abs/2005.00628) that training on a paraphrase detection task (QQP) before training on an entailment task (RTE) actually _hurts_ performance compared to training on the entailment task only. We believe it is because the low-level specifics are quite different in these tasks and therefore, believe our choice to train on paraphrase and refer to our evaluation on entailment as "zero-shot" is justified. We also note that all NLP tasks have some degree of overlap in terms of the "skills" they require, and therefore any zero-shot tasks overlap to some extent with a training task (including unsupervised pretraining).
>
> That being said, we agree that it is difficult to partition tasks into disjoint sets. Please see Appendix C of the updated draft for a further discussion of this issue of categorizing tasks from a human perspective. As for categorizing tasks from a model’s perspective, your next suggestion is highly relevant:
>
> > 3. “Some analysis on transferability of the tasks, to study the impact of different source tasks on target tasks, will be an interesting contribution for the multi-task setting.”
>
> For sure! In fact, we did initially start with preliminary experiments with Task2Vec ([Achille et al, 2019](https://arxiv.org/abs/1902.03545); [Vu et al. 2020](https://arxiv.org/abs/2005.00770)), a method which identifies which tasks provide the most transfer gains to others. One of the biggest takeaways from our early experiments was that the target space matters the most. For example, SCAN ([Lake and Baroni, 2018](https://arxiv.org/abs/1711.00350)), a semantic parsing task that requires generating sequences of commands encoded as special symbols, occupies a task embedding cluster of its own, providing little generalization benefits to other tasks. Such results contributed to our decision to only include natural language (e.g., no programming language or complex math expressions) in our main train and eval mixtures. However, lacking a complete analysis of task embeddings with all of our models, we leave this promising direction to future work.

---

> > ### Comment · Reviewer_Ju6C · 2021-11-29
> > **Response**
> >
> > Thanks a lot for your response. I have read the feedback and other responses and decided to retain the original recommendation. In order to demonstrate "true" zero-shot generalization which is the main claim of the paper, the authors need to perform the experiments as mentioned in the review.

---

### Official Review · Reviewer_zBjM · 2021-11-03

**Correctness:** 4
**Technical Novelty And Significance:** 2
**Empirical Novelty And Significance:** 3
**Recommendation:** 8
**Confidence:** 4

**Main Review:**

Overall, this is a conceptually simple and clean paper with extensive results.

Strengths
The tasks considered are extensive, covering many areas of NLP tasks such as QA, NLI, sentiment classification, summarization etc. The zero-shot evaluation tasks are of different categories than in the training set, which helps support the zero-shot generalizability claim.


Weaknesses
- One could say that the paper lacks technical novelty, but to me, a simple paper with extensive results that work is a better than a technically rich paper that doesn't work as well.
- In a way, we can see the multi tasks as a single giant task with many domains. If we view this as question answering, the paper is essentially training a system that learns to answer well given various domains of questions/tasks. What would also be interesting is to see/what's missing in the paper, is to carefully study the difference in domains of the training tasks, versus zero-shot evaluation tasks. Repeated experiments with different subsets of training/eval tasks can help shed some light as well. However, I do understand that it would be an empirically expensive set of experiments.
- The conclusion of this paper is that multi-task supervised learning makes the model generalize well. However, we still rely on a vast amount of supervised data. In a way, the comparison with GPT-3 is not too fair.


Questions
Can authors provide intuition about the performance of T0 vs FLAN on the datasets that T0 does better, or worse? Do it mainly have to do with auto-regressive style versus encoder-decoder (that T0 does not predict the input)?

**Summary Of The Paper:**

This paper trains a sequence to sequence model in a large multi-task setting and tests the model's ability to generalize to unseen tasks (zero-shot). The crux of the contribution lies in the design of prompt setups, extensive experiment, and the evaluation. The conclusive of this paper supports a growing trend/consensus in the community that multi-task learning can be a good way for generalizability on unseen tasks.

**Summary Of The Review:**

This is a solid paper in terms of contribution to the language modeling line of work. It will steer the community in the right direction of doing multi-task learning as a way to meta-learn new task or adapt to unseen tasks.

---

> ### Author Response · Authors · 2021-11-17
> **Answer to reviewer 1 - zBjM**
>
> Thank you for your review! To address your specific concerns:
>
> > 1. “Can authors provide intuition about the performance of T0 vs FLAN on the datasets that T0 does better, or worse? Does it mainly have to do with auto-regressive style versus encoder-decoder (that T0 does not predict the input)?”
>
> We have highlighted key differences between FLAN and T0, namely, the different architecture (decoder-only vs. encoder-decoder), the different pretraining objective (auto-regressive language modeling vs. masked language modeling + language modeling adaptation), and the qualitative differences in our prompts. We observed noteworthy differences in the results: while the FLAN authors reported that the massively multitask fine-tuning did not help an 8B parameters model, we observed significant improvements after this multitask fine-tuning with models of 3B and 11B parameters. These are included in the paper and in Appendix D.
>
> Other factors which would be necessary to properly control include pretraining dataset and fine-tuning hyperparameters (number of steps for instance). It is not obvious why T0 performs worse than FLAN on some datasets and better on others, and we leave that question for future works.
>
> >  2. “However, we still rely on a vast amount of supervised data. In a way, the comparison with GPT-3 is not too fair.”
>
>
> [Same as Reviewer 3’s Question 9 and Summary 3] Note that, as detailed in Section 1, our research question is specifically to compare the implicit multitask training in large LM pretraining vs. explicit multitask prompted training with smaller models. We show that the latter is a more efficient method (as opposed to merely scaling to an ever larger model) to achieve better generalization to unseen tasks. For this research question, we wouldn’t consider it unfair as neither T0 nor GPT-3 was trained on any of the held-out tasks.

---

### Decision · Program_Chairs · 2022-01-20

**Decision:**

Accept (Spotlight)

**Comment:**

This paper studies constructing text2text transformer models that are good at zero-shot task generalization via multi-task learning over a diverse set of NLP tasks. One main contribution of the work is to create prompt templates for various NLP tasks (that are of different task formats) such that all tasks can be framed into text2text learning format and that is "natural" to the pretrained T6 model. The paper conducts extensive experiments to demonstrate the promising zero-shot generalization ability of such multi-task learner.

Strength:
- Important problem setup that has broad applications
- Extensive experiments to validate the claims
- Useful resources are developed for the problem

Weakness:
- Good to study the effect of using different combination of training tasks on the downstream zero-shot generalization, which can shed some light on the usefulness of upstream tasks
- Justification of "true zero shot learning" capability would require further experiments on analyzing the data overlap between MTL datasets (and also T5 pertaining task data) and the unseen task data.
- Some more discussion on the task split and categorization will be helpful.